# Zeroth-Order Policy Gradient for Reinforcement Learning from Human Feedback without Reward Inference

**Qining Zhang & Lei Ying**
Department of Electrical Engineering and Computer Science
University of Michigan – Ann Arbor
Ann Arbor, MI 48105, USA
`{qiningz, leiying}@umich.edu`

## Abstract

Reward inference (learning a reward model from human preferences) is a critical intermediate step in the *Reinforcement Learning from Human Feedback* (RLHF) pipeline for fine-tuning *Large Language Models* (LLMs). In practice, RLHF faces fundamental challenges such as distribution shift, reward model overfitting, and problem misspecification. An alternative approach is direct policy optimization without reward inference, such as *Direct Preference Optimization* (DPO), which provides a much simpler pipeline and has shown empirical success in LLM applications. However, DPO utilizes the closed-form expression between the optimal policy and the reward function, which is only suitable under the bandit setting or deterministic MDPs. This paper develops two RLHF algorithms without reward inference for *general* RL problems beyond bandits and deterministic MDPs, and *general* preference models beyond the Bradley-Terry model. The key idea is to estimate the local value function difference from human preferences and then approximate the policy gradient with a zeroth-order gradient approximator. For both algorithms, we establish polynomial convergence rates in terms of the number of policy gradient iterations, the number of trajectory samples, and human preference queries per iteration. Numerical experiments in stochastic environments validate the performance of our proposed algorithms, outperforming popular RLHF baselines such as DPO and PPO. Our paper shows that there exist provably efficient methods to solve general RLHF problems without reward inference.

## 1 Introduction

In the past decade, we have witnessed unprecedented success in applying *Reinforcement Learning* (RL) to many applications, such as video games (Knox & Stone, 2008; Warnell et al., 2018), recommendation and search (Zeng et al., 2016; Kohli et al., 2013), and autonomous driving (Kiran et al., 2022). RL studies the interaction between decision-making agents and an evolving dynamic environment. At each time step, the agent takes a certain decision (action) given the current state, and a reward signal to measure the quality of that decision is provided by the environment. The agent's goal is to learn a policy to maximize the cumulative reward, and the quality of the learned policy will depend on the per-step reward function. In classic RL, this reward function is usually handcrafted by domain experts to ensure it aligns with human interests. However, the problem of identifying a "good" reward function, also referred to as *Inverse Reinforcement Learning* (IRL), is non-trivial and one of the most fundamental problems in the history of RL (Ng & Russell, 2000). In recent years, *Reinforcement Learning from Human Feedback* (RLHF) that uses human preference feedback as a signal to recover a reward function has emerged to fine-tune *Large Language Models* (LLMs), which has delivered significant success (Christiano et al., 2017; Wu et al., 2021; Nakano et al., 2021; Ziegler et al., 2019; Stiennon et al., 2020; Ouyang et al., 2022). RLHF follows the diagram shown in Fig. 1, which includes three major steps (Ouyang et al., 2022): (i) pre-train a policy neural network, (ii) collect sets of trajectory pairs and query human evaluators for preferences over trajectory pairs to train a reward model using maximum likelihood to align with human feed-

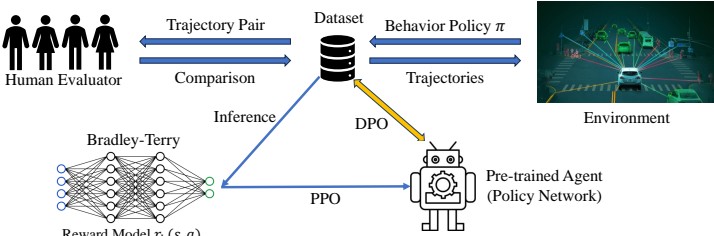

Figure 1: classic policy-based RLHF and DPO: classic RLHF, such as PPO, involves three steps: (i) policy pre-training, (ii) reward inference, and (iii) policy network training with reward model. DPO does not use a reward network but directly optimizes the policy network from human preferences.

back, and (iii) use policy-optimization-based RL algorithms such as PPO (Schulman et al., 2017) to fine-tune the policy network with the reward signals generated by the reward model. The reward inference intermediate step, which trains the reward network, is crucial to obtaining a high-quality policy through RL in the final step.

**Drawbacks of Reward Inference.** To train a good reward model, i.e., to infer the underlying per-step reward function from human feedback (Christiano et al., 2017; Wang et al., 2023), the most common approach is to assume the feedback is generated based on a preference model such as the *Bradley-Terry* model (Bradley & Terry, 1952), and then maximize the log-likelihood of the collected trajectory comparison dataset accordingly over all possible (parameterized) reward functions. This procedure is indeed analyzed in most theoretical RLHF papers for both offline (Zhu et al., 2023; Zhan et al., 2024a) and online settings (Saha et al., 2023; Zhan et al., 2024b; Wu & Sun, 2024; Wang et al., 2023; Du et al., 2024). However, several challenges occur in practice for reward model training, such as double problem misspecification, reward model evaluation without ground truth, distribution shift, and overfitting in joint reward model and policy training (Casper et al., 2023). These drawbacks are also reflected in the theoretical results, e.g., overfitting of the maximum likelihood estimator (MLE) in Zhu et al. (2024). Moreover, similar to the dilemma in IRL, the reward function that could explain human feedback is often not unique, especially when given limited training trajectories (Arora & Doshi, 2021; Ng & Russell, 2000). Some reward models may make it difficult for agents to learn a good RL policy.

**DPO.** To avoid the drawbacks of the reward inference in RLHF, Rafailov et al. (2023) proposed an algorithm called *Direct Preference Optimization* (DPO), which fine-tunes the LLM *directly* from human preferences. Based on the Bradley-Terry preference model and a closed-form expression of the optimal policy given a reference policy and the reward function, DPO constructs a loss function directly from human feedback for learning the optimal policy to avoid reward inference. This provides a much simpler pipeline and has great empirical performance (Rafailov et al., 2023; 2024a;b). However, the closed-form expression of the optimal policy that DPO builds on is only for non-parametric policies, and its theoretical justification only works for the bandit setting (Rafailov et al., 2023) or RL problems with deterministic transitions (Rafailov et al., 2024b). It remains an open question how to solve general RLHF problems without reward inference.

**RLHF without Reward Inference.** Recently, value-based RLHF algorithms without global reward inference have been theoretically developed and analyzed (Xu et al., 2020; Zhang et al., 2024a) based on a dueling bandit approach (Bengs et al., 2021). The results, however, only hold for MDPs in tabular settings with finite state and action spaces. Chen et al. (2022) studied the function approximation regime, but their algorithm requires both the true preference model and the transition kernel to belong to a known function class, which is also far from practice. The result also depends on the function class complexity, which is usually large for most function approximators in practice. So far, no provable policy-based algorithm in this category has been developed.

This paper addresses the following important question:

> *Does there exist a provably efficient RLHF approach that does not require a reward model and works for general RL problems such as stochastic MDPs or infinite state and action spaces?*

## 1.1 MAIN CONTRIBUTIONS

DPO (Rafailov et al., 2023) establishes a direct connection between human preferences and RL based on the Bradley-Terry model and the optimal policy in closed form:

$$\pi^*(a|x) \propto \pi_{\text{ref}}(a|x) \exp\left(\frac{1}{\beta} r(x,a)\right),\tag{1}$$

where $r(x,a)$ is the reward in state $x$ with action $a$, $\pi_{\text{ref}}$ is a reference policy and $\pi^*$ is the optimal policy. Based on the direction connection, the policy optimization can be formulated as a direct matching between human preference and the optimal policy with a log-likelihood loss function. In a recent paper (Rafailov et al., 2024a), it has been further shown that DPO solves a KL-divergence-constrained policy optimization problem for the *deterministic* token-level MDP for LLMs, where the next state is deterministic given the current state and action. For general RL problems with parameterized policies, equation 1 does not hold, and it is often hard, if not impossible, to obtain a "global" function like it that connects the optimal policy and the reward (hence human feedback).

This paper exploits the "local" relation between human feedback and policy optimization. In particular, given a policy $\pi_{\boldsymbol{\theta}}$ and a perturbed version of the policy $\pi_{\boldsymbol{\theta}+\boldsymbol{v}}$ where $\boldsymbol{v}$ is a small perturbation vector, we use human feedback over the trajectories generated from both policies to inform the direction of a more preferred policy. Intuitively, if one trajectory is preferred over the other, the policy that generates this trajectory is likely to have a higher value. Then given a preference model such as the Bradley-Terry model, we can further estimate the value function differences of the two policies, $V(\pi_{\boldsymbol{\theta}+\boldsymbol{v}}) - V(\pi_{\boldsymbol{\theta}})$, where $V(\pi)$ is the value function associated with policy $\pi$. Finally, the value difference can be used as an estimator of policy gradient, $\nabla_{\boldsymbol{\theta}} V(\pi_{\boldsymbol{\theta}})$, following the zeroth-order optimization approach (Nesterov & Spokoiny, 2017; Ghadimi & Lan, 2013) to improve the policy.

Based on this idea, this paper proposes two RLHF algorithms without reward inference: ***Z**eroth-**O**rder **P**olicy **G**radient* (ZPG) and ***Z**eroth-**O**rder **B**lock-**C**oordinate **P**olicy **G**radient* (ZBCPG), both from Human Feedback. ZBCPG differs from ZPG in its policy perturbation rule, which has lower computational complexity and allows parallel optimization since one can sample multiple perturbed policies to perform policy gradient and aggregate the estimated gradient. Under mild assumptions, both algorithms have the following rate of convergence to a stationary policy:

$$\mathcal{O}\left(\frac{Hd}{T} + \frac{d^2\sqrt{\log M}}{\sqrt{M}} + \frac{Hd\sqrt{d}}{\sqrt{N}}\right),$$

where $d$ is the dimension of policy network parameter $\boldsymbol{\theta}$, $H$ is the planning horizon, $T$ is the number of policy gradient steps, $N$ is the number of policy perturbations each step, and $M$ is the number of human queries for each pair of trajectories.

We remark that Tang et al. (2024b) proposes a similar approach towards utilizing human feedback and a zeroth-order gradient descent algorithm from ranking data. However, they assume an error-free ranking oracle over policies based on their value functions, which makes their problem a deterministic optimization problem and does not apply to trajectory preference data like in RLHF and DPO. This paper studies RLHF with trajectory preferences and quantifies the impacts of stochastic trajectories and human preferences on the rate of convergence of RLHF without reward inference.

## 2 PRELIMINARIES

**Episodic RL.** We consider an episodic RL problem $\mathcal{M} = (\mathbb{S}, \mathbb{A}, H, \boldsymbol{P}, \boldsymbol{\mu}_0)$, where $\mathbb{S}$ is the state space and $\mathbb{A}$ is the action space (both can be continuous), $H$ is the RL planning horizon, $\boldsymbol{P} = \{\boldsymbol{P}_h\}_{h=1}^H$ is the set of transition kernels, and $\boldsymbol{\mu}_0$ is the initial distribution. At the beginning of each episode, the agent will choose a policy $\pi$ represented by $H$ functions $\{\pi_h : \mathbb{S} \to \mathcal{P}(\mathbb{A})\}_{h=1}^H$, where $\mathcal{P}(\mathbb{A})$ denotes the set of all probability distributions over the action space. Then, an initial state $s_1$ is sampled from the initial distribution $\boldsymbol{\mu}_0$. At step $h$, the agent takes an action $a_h = \pi_h(s_h)$ after observing state $s_h$. The environment then moves to the next state $s_{h+1}$ sampled from the distribution $\boldsymbol{P}_h(\cdot|s_h, a_h)$ without revealing any reward feedback. We use $\tau$ to denote a trajectory with planing horizon $H$, i.e., $\tau = \{(s_h, a_h)\}_{h=1}^H$.

**Trajectory Reward.** we assume the expected reward of each trajectory $\tau$ is a general function $r(\tau)$ which maps any trajectory to a value in $[0, H]$ (Zhang et al., 2024a). Without loss of generality,

we scale the average per-step reward into $[0, 1]$, and the return of the trajectory does not necessarily need to be the sum of per-step rewards. For any given policy $\pi$, we can formulate the initial value function $V_1^\pi(s)$ as the expected reward of trajectories starting from $s$ with policy $\pi$:

$$V_1^\pi(s) = \mathbb{E}_\pi\left[r(\tau)|\, s_1 = s\right] = \mathbb{E}\left[r(\tau)|\, s_1 = s, \{a_1, \cdots, a_H\} \sim \pi\right].$$

The goal of the RL problem is to find a policy to maximize $V(\pi) = \mathbb{E}_{s \sim \boldsymbol{\mu}_0}[V_1^\pi(s)]$.

**Policy Parameterization.** to address the large state space $\mathbb{S}$ and action space $\mathbb{A}$ in most RL problems, we assume access to a parameterized policy network $\mathsf{N}_{\boldsymbol{\theta}} : \mathbb{S} \times [H] \to \mathcal{P}(\mathbb{A})$ which takes a state and a decision-making step as input, and then outputs the probability distribution of the next action. Here $\boldsymbol{\theta} \in \mathbb{R}^d$ is the policy network parameter vector. Each parameter $\boldsymbol{\theta}$ through the policy network will induce a policy, which we slightly abuse the notation and use $\pi_{\boldsymbol{\theta}}$ to denote.

**Human Feedback.** The agent has access to human feedback that provides a preference based on the rewards of two trajectories. In each episode, the agent can choose two trajectories $\tau_0$ and $\tau_1$ to query human preference: one-bit feedback $o \in \{0, 1\}$. We assume the preference $o$ is generated according to a known preference model where the probability of the outcome between two trajectories is determined by the difference in their rewards. Since the difference is not necessarily a value inside the unit interval, the preference model uses a *link* function $\sigma : \mathbb{R} \to [0, 1]$ to map these differences of rewards to actual probabilities, i.e.,

$$\mathbb{P}(\tau_1 \succ \tau_0) = \sigma(r(\tau_1) - r(\tau_0)), \tag{2}$$

where $\tau_1 \succ \tau_0$ is the event that the human feedback prefers $\tau_1$ over $\tau_0$. The human feedback $o$, therefore, is a random sample from a Bernoulli distribution with $\mathbb{P}(o = 1) = \mathbb{P}(\tau_1 \succ \tau_0)$. The notion of link function comes from the dueling bandit literature to model preference with latent utility between arms, e.g., see Bengs et al. (2021, Section 3.2). This preference model has been used in dueling bandits (Bengs et al., 2021; Yue & Joachims, 2009; Kumagai, 2017; Ailon et al., 2014) as well as RLHF (Wang et al., 2023). One can see that one specific link function $\sigma$ will define a specific preference model (Azari et al., 2012), i.e., replacing $\sigma(\cdot)$ with a logistic function, we recover the Bradley-Terry model (Bradley & Terry, 1952), which is commonly used in RLHF for both practical (Christiano et al., 2017; Ouyang et al., 2022; Rafailov et al., 2023) and theoretical works (Du et al., 2024; Zhan et al., 2024a;b). On the other hand, let $\sigma(\cdot)$ be the cumulative distribution function (CDF) of standard normal distribution, we obtain the *Probit* model (Thurstone, 1927) which has wide application in the study of social choice theory, economy, and psychology (Train, 2009; Greene, 2000). A detailed discussion on other preference models is provided in the appendix. The following assumption on the link function is standard in both dueling bandits (Bengs et al., 2021) and preference-based RL (Wang et al., 2023). One can easily verify that both the Bradley-Terry and Probit models satisfy this assumption.

**Assumption 1** *The link function $\sigma(\cdot)$ in the preference model in equation 2 is bounded within $[0, 1]$ and strictly monotonically increasing on $[-H, H]$ with $\sigma(0) = 1/2$.*

**Problem and Notations.** Our goal is to find parameter $\boldsymbol{\theta}$ that maximizes the value function, i.e., $\max_{\boldsymbol{\theta} \in \mathbb{R}^d} V(\pi_{\boldsymbol{\theta}})$. For a scalar $a$, we use $\mathrm{trim}[a|\Delta]$ to represent $\min\{\max\{a, \Delta\}, 1 - \Delta\}$. For a vector $\boldsymbol{v}$, $\mathrm{trim}[\boldsymbol{v}|\Delta]$ represents the vector after applying the $\mathrm{trim}$ operator to each element respectively. Let $\boldsymbol{e}_i \in \mathbb{R}^d$ represent the unit vector with all zero elements but 1 on the $i$-th coordinate.

## 3 ZEROTH-ORDER POLICY GRADIENT ALGORITHMS FOR RLHF

In this section, we propose two RLHF algorithms without reward inference, motivated by the relation between preference and zeroth-order gradient. We first present ZPG, a stochastic gradient descent algorithm, and then ZBCPG, a stochastic block-coordinate descent algorithm.

### 3.1 ZPG: ZEROTH-ORDER POLICY GRADIENT FROM HUMAN FEEDBACK

Our first algorithm ZPG, consists of the following five steps in each policy gradient iteration:

- From the current policy $\pi_{\boldsymbol{\theta}_t}$, it first obtained a perturbed policy $\pi_{\boldsymbol{\theta}_t + \mu \boldsymbol{v}_t}$ (line 2-3).
- Sample $N$ pairs of trajectories under the two policies $\pi_{\boldsymbol{\theta}_t}$ and $\pi_{\boldsymbol{\theta}_t + \mu \boldsymbol{v}_t}$ (lines 5-6).

---

**Algorithm 1:** Zeroth-Order Policy Gradient from Human Feedback

---

**Parameters:** initial parameter $\boldsymbol{\theta}_0$, learning rate $\alpha$, trim size $\Delta$, perturbation distance $\mu$.

1 **for** $t = 1 : T$ **do**
2      sample $\boldsymbol{v}_t$ uniformly from a unit sphere $\mathbb{S}^{d-1} = \left\{ \boldsymbol{v} \in \mathbb{R}^d \middle| \|\boldsymbol{v}\|_2 = 1 \right\}$;
3      obtain a perturbed policy $\pi_{\boldsymbol{\theta}_t + \mu \boldsymbol{v}_t}$;
4      **for** $n = 1 : N$ **do**
5          sample trajectory $\tau_{n,0} \sim \pi_{\boldsymbol{\theta}_t}$;
6          sample trajectory $\tau_{n,1} \sim \pi_{\boldsymbol{\theta}_t + \mu \boldsymbol{v}_t}$;
7          query $M$ human evaluators with $(\tau_{n,1}, \tau_{n,0})$ and obtain feedback $[o_{n,1}, \cdots, o_{n,M}]$;
8          estimate preference probability $p_{t,n} = \text{trim}\left[ \sum_{m=1}^M \frac{o_{n,m}}{M} \middle| \Delta \right]$;
9      estimate the policy gradient: $\hat{\boldsymbol{g}}_t = \frac{d}{\mu} \frac{\sum_{n=1}^N \sigma^{-1}(p_{t,n})}{N} \boldsymbol{v}_t$;
10      update $\boldsymbol{\theta}_{t+1} = \boldsymbol{\theta}_t + \alpha \hat{\boldsymbol{g}}_t$;

---

- For each trajectory pair, say $(\tau_{n,1}, \tau_{n,0})$, obtain $M$ independent human preferences (line 7) and estimate the probability that $\tau_{n,1}$ is preferred over $\tau_{n,0}$ (line 8), denoted by $p_{t,n}$.

- Use the $N$ estimates $\{p_{t,n}\}_n$ and link function $\sigma(\cdot)$ to estimate the gradient $\hat{\boldsymbol{g}}_t$ (line 9).

- Update the current policy to a new policy $\boldsymbol{\theta}_{t+1}$ using gradient ascent (line 10).

The pseudo-code is presented in Alg. 1. As we mentioned earlier, our approach uses human feedback in a way different from both the classic reward inference in RLHF and DPO. The reward inference uses human preferences to recover the *global* reward function, and DPO relates the human preference generation mechanism to the optimal policy. We view the human feedback as local information that points to the direction of a more preferred policy, i.e., the policy gradient direction. Some online RLHF algorithms, such as online DPO, also exploit similar local estimation viewpoints, i.e., using new trajectories of the current policy to locally improve the estimation of DPO loss and then proceed. However, we want to emphasize that the relation between the DPO loss and the optimal policy is still global, and it is limited to deterministic MDPs, which shows the novelty of our approach. The algorithm we propose has two key components: (i) a value function difference estimator from human preference, and (ii) a policy gradient estimator from value function difference.

**Policy Gradient Approximation.** At each iteration of the algorithm, it first samples a $d$-dimensional vector $\boldsymbol{v}_t$ from a unit sphere and then perturbs the policy network parameter $\boldsymbol{\theta}_t$ along the direction of this sampled vector. Then, it uses the inner for loop to construct an estimation of the value function difference between the original policy and the perturbed policy, i.e., $V(\pi_{\boldsymbol{\theta}_t + \mu \boldsymbol{v}_t}) - V(\pi_{\boldsymbol{\theta}_t})$. We then plug it into the zeroth-order stochastic gradient descent (SGD) algorithm proposed in (Ghadimi & Lan, 2013) to construct a zeroth-order approximation to the policy gradient, i.e.,

$$\nabla_{\boldsymbol{\theta}} V(\pi_{\boldsymbol{\theta}_t}) \approx \mathbb{E}_{\boldsymbol{v}_t}\left[ \frac{d}{\mu}\left( V(\pi_{\boldsymbol{\theta}_t + \mu \boldsymbol{v}_t}) - V(\pi_{\boldsymbol{\theta}_t}) \right) \boldsymbol{v}_t \right].$$

We remark that the random vector $\boldsymbol{v}_t$ for each iteration can also be drawn from a normal distribution (Nesterov & Spokoiny, 2017), but the unit sphere is more numerically stable.

**Value Function Inference.** The inner for loop of Alg. 1 aims to estimate the value function difference between the perturbed policy $\pi_{\boldsymbol{\theta}_t + \mu \boldsymbol{v}_t}$ and current policy $\pi_{\boldsymbol{\theta}_t}$. The algorithm samples multiple trajectory pairs with both policies and for each pair, it queries humans multiple times to obtain pairwise preferences $[o_{n,1}, \cdots, o_{n,M}]$. It then uses the preferences to construct a robust estimator $p_{t,n}$ to approximate the probability of comparison $\mathbb{P}(\tau_{n,1} \succ \tau_{n,0})$, which is further converted to an estimate of the value function difference based on the preference model in equation 2 as follows:

$$V(\pi_{\boldsymbol{\theta}_t + \mu \boldsymbol{v}_t}) - V(\pi_{\boldsymbol{\theta}_t}) \approx \frac{1}{N} \sum_{n=1}^N \sigma^{-1}(p_{t,n}). \tag{3}$$

Querying humans multiple times ensures an accurate estimation of the reward gap between two trajectories. The reward gap of two trajectories is a random sample of the value function difference, so we sample multiple trajectories to ensure the average trajectory reward gap converges to the

---

**Algorithm 2:** Zeroth-Order Block-Coordinate Policy Gradient from Human Feedback

---

**Parameters:** initial parameter $\boldsymbol{\theta}_0$, learning rate $\alpha$, trim size $\Delta$, perturbation distance $\mu$, coordinate batch size $K$.

1 **for** $t = 1 : T$ **do**
2     sample a set of $K$ coordinates $\boldsymbol{i}_t = [i_{t,1}, i_{t,2}, \cdots, i_{t,K}]$ from $\{1, 2, \cdots, d\}$;
3     sample a set $\boldsymbol{\lambda}_t = [\lambda_{t,1}, \lambda_{t,2}, \cdots, \lambda_{t,K}]$ where each $\lambda_{t,j}$ is uniformly sampled from $\{-1, 1\}$;
4     construct the perturbation vector: $\boldsymbol{v}_t = \frac{1}{\sqrt{K}} \sum_{j=1}^{K} \lambda_{t,j} \boldsymbol{e}_{i_{t,j}}$;
5     **for** $n = 1 : N$ **do**
6        sample trajectory $\tau_{n,0} \sim \pi_{\boldsymbol{\theta}_t}$;
7        sample trajectory $\tau_{n,1} \sim \pi_{\boldsymbol{\theta}_t + \mu \boldsymbol{v}_t}$;
8        query $M$ human evaluators with $(\tau_{n,1}, \tau_{n,0})$ and obtain feedback $[o_{n,1}, \cdots, o_{n,M}]$;
9        estimate preference probability $p_{t,n} = \text{trim}\left[\sum_{m=1}^{M} \frac{o_{n,m}}{M} \Big| \Delta\right]$;
10    estimate the policy gradient: $\hat{\boldsymbol{g}}_t = \frac{d}{\mu} \frac{\sum_{n=1}^{N} \sigma^{-1}(p_{t,n})}{N} \boldsymbol{v}_t$;
11    update $\boldsymbol{\theta}_{t+1} = \boldsymbol{\theta}_t + \alpha \hat{\boldsymbol{g}}_t$;

---

value function difference. To ensure finite variance after applying $\sigma^{-1}(\cdot)$ function, we trim $p_{t,n}$ with a small constant which can be set to $\min\{\sigma(-H), 1 - \sigma(H)\}$ in this case.

## 3.2 ZBCPG: ZEROTH-ORDER BLOCK-COORDINATE POLICY GRADIENT FROM HUMAN FEEDBACK

In high-dimensional optimization problems, it is usually memory and operation-inefficient to approximate the full gradient and update all the parameters in the policy network at the same iteration step (Malladi et al., 2023; Zhang et al., 2024b), which motivates parameter-efficient fine-tuning (PEFT). The stochastic (block) coordinate descent approach naturally arises because of its ease of implementation, low memory requirements, and adaptability to distributed settings (Nesterov, 2012; Lu & Xiao, 2015). The same advantage also applies to RLHF when the number of parameters in the policy network is too large. Therefore, we propose ZBCPG, a block coordinate version of ZPG, which is summarized in Alg. 2. The key difference between ZBCPG and ZPG is the choice of the perturbation direction, where we use Rademacher noise instead of the normal perturbation in ZPG.

**Zeroth-Order Block Coordinate Gradient Approximation.** Instead of sampling from a sphere, which perturbs all parameters of the policy network, ZBCPG separates the sampling procedure into two simple parts: first, sample a minibatch of coordinates and then sample a zero-centered Bernoulli random variable for each coordinate, which still results in a valid gradient estimator.

$$\nabla_{\boldsymbol{\theta}} V(\pi_{\boldsymbol{\theta}_t}) \approx \mathbb{E}_{\boldsymbol{i}_t, \boldsymbol{\lambda}_t} \left[ \frac{d}{K\mu} \left( V(\pi_{\boldsymbol{\theta}_t + \mu \boldsymbol{v}_t}) - V(\pi_{\boldsymbol{\theta}_t}) \right) \boldsymbol{v}_t \right].$$

The block-coordinate approach allows us to (i) perturb a subset of parameters at each iteration, e.g., a specific layer of the policy network for fine-tuning, and (ii) have a parallel implementation where we have multiple gradient estimators $\hat{\boldsymbol{g}}_t$ when updating the policy. We will later show that both algorithms have similar provable convergence guarantees, but the analysis of ZBCPG is more challenging due to the perturbation mechanism.

## 4 THEORETICAL ANALYSIS: RATE OF CONVERGENCE

In this section, we provide theoretical performance guarantees for both ZPG and ZBCPG. We first provide technical assumptions on the preference generation model, the policy network, and the value function landscape, which are necessary for deriving theoretical insights.

## 4.1 ASSUMPTIONS

To infer the local reward difference from human preference probability through the link function $\sigma(\cdot)$, we impose the following assumption, which is satisfied by the Bradley-Terry model. A slightly weaker assumption is also adopted by Wang et al. (2023) and justified as a minimal requirement to learn the optimal policy. We use $\Delta = \min\{\sigma(-H), 1 - \sigma(H)\}$ as the trim constant.

**Assumption 2** *The inverse link function $\sigma^{-1}(\cdot)$ is L-Lipchitz continuous on $[\Delta, 1 - \Delta]$.*

We further require the landscape of the value function and the policy network to be "regular", and impose the following assumption, which is a standard assumption used in nonconvex optimization literature (Liu et al., 2019; Bernstein et al., 2018; Reddi et al., 2018).

**Assumption 3** *The value function $V(\pi_{\boldsymbol{\theta}})$ for the policy network parameters $\boldsymbol{\theta}$ is L-smooth on $\mathbb{R}^d$.*

Since a trajectory reward is bounded in $[0, H]$, $V(\pi_{\boldsymbol{\theta}^*}) < \infty$, where $\boldsymbol{\theta}^*$ is the global optimal solution. For simplicity, we assume $L$ is the constant upper bound for both assumptions.

## 4.2 CONVERGENCE RATE AND SAMPLE COMPLEXITY

In this section, we present the theoretical guarantees for both ZPG and ZBCPG under all three assumptions mentioned in the previous sections. We aim to learn an $\epsilon$-stationary policy $\pi_{\boldsymbol{\theta}}$ with $\|\nabla_{\boldsymbol{\theta}} V(\pi_{\boldsymbol{\theta}})\|_2^2 \le \epsilon$, and study the convergence rate and sample complexity.

**Theorem 1** *Choose the perturbation distance to be $\mu^2 = \Theta\left(\max\left\{\frac{1}{\sqrt{M}}, \frac{H}{\sqrt{dN}}\right\}\right)$ and learning rate to be $\alpha = \Theta(d^{-1})$. If $M = \Omega(H^2)$ and we randomly pick $\boldsymbol{\theta}_R$ uniformly from the trajectory $\{\boldsymbol{\theta}_0, \boldsymbol{\theta}_1, \cdots, \boldsymbol{\theta}_{T-1}\}$, then the convergence rate of ZPG satisfies:*

$$\mathbb{E}\left[\|\nabla_{\boldsymbol{\theta}} V(\pi_{\boldsymbol{\theta}_R})\|_2^2\right] = \mathcal{O}\left(\frac{Hd}{T} + \frac{d^2 \sqrt{\log M}}{\sqrt{M}} + \frac{Hd\sqrt{d}}{\sqrt{N}}\right).$$

**Theorem 2** *Choose the perturbation distance to be $\mu^2 = \Theta\left(\max\left\{\frac{1}{\sqrt{M}}, \frac{H}{\sqrt{dN}}\right\}\right)$ and learning rate to be $\alpha = \Theta(d^{-1})$. If $M = \Omega(H^2)$ and we randomly pick $\boldsymbol{\theta}_R$ uniformly from the trajectory $\{\boldsymbol{\theta}_0, \boldsymbol{\theta}_1, \cdots, \boldsymbol{\theta}_{T-1}\}$, then the convergence rate of ZBCPG satisfies:*

$$\mathbb{E}\left[\|\nabla_{\boldsymbol{\theta}} V(\pi_{\boldsymbol{\theta}_R})\|_2^2\right] = \mathcal{O}\left(\frac{Hd}{T} + \frac{d^2 \sqrt{\log M}}{\sqrt{M}} + \frac{Hd\sqrt{d}}{\sqrt{N}}\right).$$

The complete proof of both theorems is presented in the appendix. Here we first provide insights into the choice of hyper-parameters and convergence rate results in both theorems, and then we discuss the challenges and technical novelties of our proof.

**Insights behind the Convergence Rate.** Both ZPG and ZBCPG have the same rate of convergence, which consists of three components: the zeroth-order gradient descent rate, the preference estimation error, and the value function approximation error

$$\underbrace{\frac{Hd}{T}}_{\text{Zeroth-Order Gradient Descent}} + \underbrace{\frac{d^2 \sqrt{\log M}}{\sqrt{M}}}_{\text{Preference Estimation}} + \underbrace{\frac{Hd\sqrt{d}}{\sqrt{N}}}_{\text{Value Function Approximation}}.$$

The second represents the error that occurs when using multiple human preferences $[o_{n,1}, \cdots, o_{n,M}]$ to approximate the population-level human preference probability for given two trajectories, i.e., $\mathbb{P}(\tau_{n,1} \succ \tau_{n,0})$. This error will further result in a bias term after being plugged into the inverse link function $\sigma^{-1}(\cdot)$ to construct an estimation of the value function difference. The third term comes from the variance of using multiple trajectory rewards to approximate the value function of a policy. The first term represents the error resulting from zeroth-order stochastic gradient descent or blocked coordinate descent, which matches the state-of-the-art analysis result $\mathcal{O}(d/T)$ for

non-convex smooth function optimization (Nesterov & Spokoiny, 2017). Even though the final convergence rates are the same and we both use constant learning rates, how we choose the perturbation distance to obtain the rate differs from (Ghadimi & Lan, 2013). Specifically, they chose a small perturbation distance with $\mu^2 = \mathcal{O}(d/T)$ to make sure the zeroth-order approximation error is of lower order. However, this choice will not work for us, because our gradient estimate is biased due to the non-linear nature of the link function in preference estimation. If we choose the perturbation distance $\mu$ to be too small, the preference estimation error will be amplified by $d/\mu$ due to the formula of zeroth-order approximation $\hat{\boldsymbol{g}}_t$. This phenomenon adds complication to our theoretical analysis. Our method is to use a moderate perturbation distance $\mu$. Moreover, this moderate perturbation distance also balances the preference estimation and the value function approximation errors.

Based on the theorems, we have the following corollary that characterizes the sample complexity.

**Corollary 1** *To learn an $\epsilon$-stationary policy, the required number of human preference queries of ZPG and ZBCPG with proper hyperparameters satisfies*

$$TMN = \mathcal{O}\left(\frac{d^8 H^3}{\epsilon^5} \log\left(\frac{d}{\epsilon}\right)\right).$$

### 4.3 TECHNICAL CHALLENGES AND PROOF NOVELTIES

This section provides an overview of the proof of zeroth-order stochastic gradient descent used in (Ghadimi & Lan, 2013; Nesterov & Spokoiny, 2017; Gao et al., 2018; Liu et al., 2019) from a Lyapunov drift optimization perspective. We then show the major technical difficulties in applying such a framework to analyze both ZPG and ZBCPG, i.e., the gradient estimator is biased due to stochastic human preference. Then, we demonstrate our novel analysis techniques to resolve them.

**Classic Proof of Zeroth-Order Optimization.** To illustrate the procedure of the analysis of zeroth-order gradient estimate, we suppose we can query $V(\pi_{\boldsymbol{\theta}})$ for any $\boldsymbol{\theta}$. This procedure makes use of the randomized smoothing function $V_\mu(\boldsymbol{\theta})$ (Ghadimi & Lan, 2013; Gao et al., 2018) as $V_\mu(\pi_{\boldsymbol{\theta}}) = \mathbb{E}_{\boldsymbol{v}'}\left[V(\pi_{\boldsymbol{\theta}+\mu\boldsymbol{v}'})\right]$, where the random vector $\boldsymbol{v}'$ follows a uniform distribution over the unit Euclidean ball. It is shown in (Gao et al., 2018) that the zeroth-order gradient estimator used in ZPG, constructed from sampling $\boldsymbol{v}_t$ uniformly over a sphere, is an unbiased estimator of the smoothing function gradient, i.e.,

$$\nabla_{\boldsymbol{\theta}} V_\mu(\pi_{\boldsymbol{\theta}}) = \mathbb{E}_{\boldsymbol{v}}\left[\frac{d}{\mu}\left(V(\pi_{\boldsymbol{\theta}+\mu\boldsymbol{v}}) - V(\pi_{\boldsymbol{\theta}_t})\right)\boldsymbol{v}\right],$$

where $\boldsymbol{v}$ is sampled from a unit sphere. The value and gradient of the smoothing function are different from the value function, but they are close as long as $\mu$ is small (Liu et al., 2018b):

$$|V_\mu(\pi_{\boldsymbol{\theta}}) - V(\pi_{\boldsymbol{\theta}})| = \mathcal{O}\left(\mu^2\right); \quad \|\nabla_{\boldsymbol{\theta}} V_\mu(\pi_{\boldsymbol{\theta}}) - \nabla_{\boldsymbol{\theta}} V(\pi_{\boldsymbol{\theta}})\|_2 = \mathcal{O}\left(\mu d\right). \quad (4)$$

The standard proof uses the randomized smoothing function $V_\mu(\pi_{\boldsymbol{\theta}})$ as the Lyapunov function and then bounds the drift given the stochastic gradient descent update rule when $\alpha = \Theta(1/d)$. Neglecting problem-independent constants, we have:

$$V_\mu(\pi_{\boldsymbol{\theta}_t}) - V_\mu(\pi_{\boldsymbol{\theta}_{t+1}})$$
$$\leq -\alpha \underbrace{\|\nabla_{\boldsymbol{\theta}} V_\mu(\pi_{\boldsymbol{\theta}_t})\|_2^2}_{\text{Drift}} + \alpha \underbrace{\langle \nabla_{\boldsymbol{\theta}} V_\mu(\pi_{\boldsymbol{\theta}_t}), \nabla_{\boldsymbol{\theta}} V_\mu(\pi_{\boldsymbol{\theta}_t}) - \hat{\boldsymbol{g}}_t \rangle}_{\text{1st Order: GradBias}} + \alpha^2 \underbrace{\|\hat{\boldsymbol{g}}_t - \nabla_{\boldsymbol{\theta}} V_\mu(\pi_{\boldsymbol{\theta}_t})\|_2^2}_{\text{2nd Order: GradVar} \approx \mu^2 d^2}.$$

Note the gradient estimator $\hat{\boldsymbol{g}}_t$ is unbiased and bounded, and the gradient of $V_\mu(\pi_{\boldsymbol{\theta}})$ is close to $V(\pi_{\boldsymbol{\theta}})$, taking a conditional expectation over the filtration before time $t$ will result in:

$$\mathbb{E}[V_\mu(\pi_{\boldsymbol{\theta}_t})|\mathcal{F}_t] - \mathbb{E}[V_\mu(\pi_{\boldsymbol{\theta}_{t+1}})|\mathcal{F}_t]$$
$$\leq -\alpha\|\nabla_{\boldsymbol{\theta}} V_\mu(\pi_{\boldsymbol{\theta}_t})\|_2^2 + \alpha\langle \nabla_{\boldsymbol{\theta}} V_\mu(\pi_{\boldsymbol{\theta}_t}), \mathbb{E}[\nabla_{\boldsymbol{\theta}} V_\mu(\pi_{\boldsymbol{\theta}_t}) - \hat{\boldsymbol{g}}_t] \rangle + \alpha^2 \mu^2 d^2$$
$$\leq -\alpha\|\nabla_{\boldsymbol{\theta}} V(\pi_{\boldsymbol{\theta}_t})\|_2^2 + \alpha\mu^2 d^2 + \alpha^2 \mu^2 d^2,$$

where the last step uses equation 4 and the fact that the gradient is unbiased. Let us choose a small learning rate $\alpha = \Theta(1/d)$ and take an expectation with a telescoping sum to obtain:

$$\underbrace{\frac{\mathbb{E}[V_\mu(\pi_{\boldsymbol{\theta}_0})] - \mathbb{E}[V_\mu(\pi_{\boldsymbol{\theta}_T})]}{T}}_{\mathcal{O}(H/T)} \lesssim -\alpha \underbrace{\mathbb{E}\left[\frac{\sum_{t=1}^T \|\nabla_{\boldsymbol{\theta}} V(\pi_{\boldsymbol{\theta}_t})\|_2^2}{T}\right]}_{\text{Target}} + \alpha\mu^2 d^2.$$

A little manipulation will lead to the following bound, which can be made small when $\mu \approx \sqrt{1/dT}$.

$$\mathsf{Target} = \mathcal{O}\left(\frac{H}{T\alpha} + \mu^2 d^2\right) = \mathcal{O}\left(\frac{Hd}{T} + \mu^2 d^2\right) = \mathcal{O}\left(\frac{Hd}{T}\right).$$

**Amplified Gradient Biases for ZPG.** If we directly apply the steps above to ZPG, we immediately run into the issue that our gradient estimator $\hat{g}_t$ in expectation is biased even compared to the smoothing function gradient due to preference estimation. Moreover, the second-order gradient variance will be larger since we used the trajectory reward to estimate the value function. From concentration, we obtain an error bound of using preference to estimate the value function difference:

$$\left|\mathbb{E}\left[\sigma^{-1}(p_{t,n})\right] - (V(\pi_{\boldsymbol{\theta}_t + \mu\boldsymbol{v}_t}) - V(\pi_{\boldsymbol{\theta}_t}))\right| \leq \tilde{\mathcal{O}}\left(\frac{1}{\sqrt{M}}\right),$$

where $\tilde{\mathcal{O}}$ hides logarithmic terms. This bias term will be amplified by $d/\mu$ and then added to the gradient estimation bias in the first-order drift term if plugged into the analysis:

$$\mathbb{E}[V_\mu(\pi_{\boldsymbol{\theta}_t})|\mathcal{F}_t] - \mathbb{E}[V_\mu(\pi_{\boldsymbol{\theta}_{t+1}})|\mathcal{F}_t] \leq \underbrace{-\alpha\|\nabla_{\boldsymbol{\theta}} V(\pi_{\boldsymbol{\theta}_t})\|_2^2 + \alpha\mu^2 d^2}_{\text{Same Drift as Before}} + \alpha \underbrace{\frac{d\|\nabla_{\boldsymbol{\theta}} V_\mu(\pi_{\boldsymbol{\theta}_t})\|_2}{\mu\sqrt{M}}}_{\text{Additional Bias}}.$$

Using the same perturbation distance as before, the additional bias will lead to an $\tilde{\mathcal{O}}(\sqrt{T/M})$ term in the final bound, which is small only when $M$ is much larger than $T$ and is much looser compared with ours. For example, letting $M = T^2$, the above bound is $\tilde{\mathcal{O}}(1/\sqrt{T})$ while ours is $\tilde{\mathcal{O}}(1/T)$.

Our approach to avoid this term in the final result is to make use of the gradient value $\nabla_{\boldsymbol{\theta}} V(\pi_{\boldsymbol{\theta}_t})$ in the first-order term to cancel out the additional bias on certain occasions. Specifically, we divide the trajectory of $\boldsymbol{\theta}_t$ into two sets, one with a relatively large gradient and one with a relatively small gradient. For $\boldsymbol{\theta}_t$ with a large gradient, we use a part of the negative drift to cancel out the additional bias, since the negative drift is the square of the gradient $\nabla_{\boldsymbol{\theta}} V(\pi_{\boldsymbol{\theta}_t})$, which is even larger. For $\boldsymbol{\theta}_t$ with a small gradient, we know the bias term will be small and thus can provide a refined drift bound. Combining this analysis with a slightly larger perturbation distance $\mu$, we will be able to balance the additional bias with gradient variance to cancel out the $\tilde{\mathcal{O}}(\sqrt{T/M})$ term and obtain the final result.

**Implicit Smoothing Function for ZBCPG.** Due to the choice of blocked perturbation vector sampling procedure, it is difficult to obtain the exact analytical expression of the smoothing function $V_\mu(\pi_{\boldsymbol{\theta}})$ whose gradient is the expectation of gradient estimation $\hat{g}_t$ for ZBCPG. This prohibits us from continuing to use $V_\mu(\pi_{\boldsymbol{\theta}})$ as the Lyapunov function, as it is hard to analyze the gradient bias and the variance without an explicit target format. However, if we rethink the reason for introducing the smoothed function in zeroth-order optimization, we hope the gradient of the smoothed function will be unbiased to cancel out the first-order positive drift. However, this is already not true in the analysis of ZPG since we have gradient estimation bias from human feedback, but it is small enough on average to be controlled. If the gradient difference between the smoothed function $V_\mu(\pi_{\boldsymbol{\theta}})$ and the vanilla value function $V(\pi_{\boldsymbol{\theta}})$ is smaller than this additional bias, then we can use the original value function $V(\pi_{\boldsymbol{\theta}})$ as the Lyapunov function at the cost of an additional bias besides preference estimation. Fortunately, this can be achieved through a carefully chosen perturbation distance $\mu$ to balance these two types of errors.

## 5 EXPERIMENTS

| Algorithm | ZPG (Ours) | ZBCPG (Ours) | RM+PPO | DPO | Online DPO |
|-----------|------------|--------------|--------|-----|------------|
| Return | $\mathbf{1.94} \pm 0.09$ | $\mathbf{1.91} \pm 0.09$ | $1.80 \pm 0.09$ | $-4.13 \pm 0.09$ | $1.71 \pm 0.09$ |

Table 1: Last Iterate Policy Average Return with Bradley-Terry Feedback.

We study the empirical performance of ZPG and ZBCPG in a stochastic GridWorld environment, where details can be found in Appendix C. In our environment, the actions chosen by the agent may be reversed with certain probabilities due to imperfect control or environmental disturbances

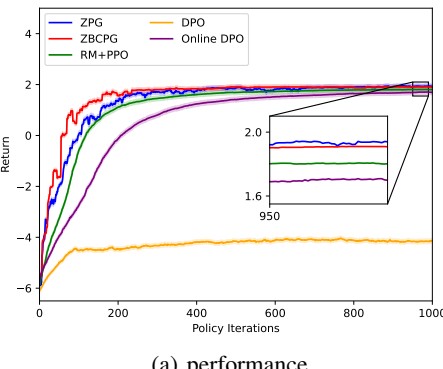 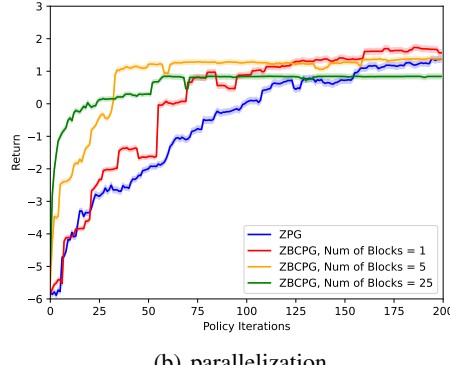

(a) performance (b) parallelization

Figure 2: GridWorld with Bradley-Terry Feedback: (a) the trajectory return of ZPG, ZBCPG, and RLHF baselines, and (b) the return of ZBCPG with different parallelization levels. All results are averaged over $10^4$ repetitions of policy evaluation, and shaded areas indicate confidence intervals.

such as wind or turbulence. The agent can query human evaluators for preference over two trajectories, and the human feedback is generated from the Bradley-Terry model with a logistic link function. We consider three baselines: (1) RM+PPO (Ouyang et al., 2022), (2) DPO for token-level MDP (Rafailov et al., 2024b), and (3) Online DPO for token-level MDP (Dong et al., 2024; Guo et al., 2024). All algorithms collect $N = 1000$ trajectory pairs between policy updates, and $M = 1000$ human experts evaluate each pair. The trajectory return for each iteration is compared in Fig. 2(a), and the return of the final policy is reported in Tab. 1. Both ZPG and ZBCPG perform better than the three baselines in both convergence rates and the quality of the last iterate policy. Compared to PPO, our algorithms converge to a better policy, partially because the reward model is inaccurate and the agent is not able to learn the optimal policy from it. It is also observed that vanilla DPO has a much worse performance than our proposed algorithms. This may result from two reasons: first, the DPO loss is valid only in a deterministic MDP, and second, DPO is constrained to the neighborhood of the sub-optimal reference policy. The online DPO algorithm improves over vanilla DPO but still has inferior performance due to the inherent model error of the DPO loss. This also shows the fundamental difference between stochastic and deterministic MDPs and the need to design RLHF algorithms for general RL problems. Moreover, online DPO converges much slower, partly because the DPO loss landscape becomes flat and hard to optimize when the weight of the KL constraint is small for better exploration. In Fig. 2(b), we also compare ZPG to distributed implementations of ZBCPG, where the panel of human evaluators is also separated into small groups for parallelization. It is shown that as the number of blocks increases, ZBCPG converges faster to a stationary policy. However, the number of human queries per pair of trajectories in each parallelization also decreases, which introduces a larger gradient bias and leads to a sub-optimal policy. Therefore, the trade-off between computation parallelization and accuracy should be taken carefully.

## 6 CONCLUSION

In this paper, we proposed two RLHF algorithms without reward inference based on a zeroth-order policy gradient called ZPG and ZBCPG, which train the policy network directly from human preferences without a global reward model. Both algorithms are shown to have a provable polynomial sample complexity to learn a stationary policy under mild conditions and exhibit nice empirical performances in environments with stochastic transitions, outperforming popular RLHF baselines.

### ACKNOWLEDGMENTS

The work of Qining Zhang and Lei Ying is supported in part by NSF under grants 2112471, 2134081, 2207548, 2240981, and 2331780.

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
