# A   RELATED WORK

We review recent developments in RLHF and zeroth-order optimization for both empirical and theoretical results. A thorough survey on RLHF can be found in Kaufmann et al. (2023) and Casper et al. (2023)

**Empirical Studies of Direct RLHF in LLMs.** DPO (Rafailov et al., 2024a) and SLiC-HF (Zhao et al., 2023) have empirically shown it is possible to directly learn an RL policy from human preference. In DPO, the authors solve a KL-divergence-constrained reward maximization problem for a prompt response generation problem similar to the contextual bandit. The optimal policy for this problem has a closed-form expression, and the reward of each prompt-response pair can be computed knowing the optimal policy and the reference policy. The authors then minimize a log-likelihood loss by plugging the reward expression into the Bradley-Terry model to measure the alignment between the policy and the human preference without training a reward network. Rafailov et al. (2024b) extends this loss to token-level MDP on the condition that the MDP transition is deterministic, i.e., the next state is a concatenation of the current state and action. This limits the DPO loss minimization approach to LLM problems. In general, MDPs with stochastic transitions, the DPO loss cannot be computed following the derivation in (Rafailov et al., 2024b). Azar et al. (2024) extends DPO to a wider class of RL problems, even avoiding the notion of an underlying reward function. Instead of maximizing the reward in a KL-constrained problem like DPO, the authors proposed to optimize a general non-decreasing function of the ground-truth population-level preference probability. Other variants are also studied. Ethayarajh et al. (2024) considers aligning policy with humans and designs loss from a prospect theory perspective, and Tang et al. (2024a) considers optimizing a general loss of the preference loss instead of the log-likelihood. Dong et al. (2024) and Xiong et al. (2024) proposed to obtain human feedback in an online fashion to mitigate the distribution-shift and over-parameterization phenomenon. Attempts to understand the theoretical performance of RLDHF algorithms such as DPO are made in (Azar et al., 2024), but the authors only showed the existence of optima of the loss function, without any policy optimality and sample complexity guarantees.

**RLHF with Provable Sample Complexity.** Two major approaches have been studied to learn the optimal policy from human preference data. The first is similar to the traditional RLHF paradigm, such as PPO used in empirical studies, which infers a reward function, or sometimes a utility function, and then trains an RL agent. This reward inference step is also called surrogate learning in the preference-based RL literature, i.e., see (Wirth et al., 2017, Sec. 2.4). The second approach directly optimizes the policy from human preferences. Empirical algorithms for both approaches have been developed without theoretical guarantees for a few years. For example, reward function or utility function estimation followed by an RL policy search algorithm has been proposed and studied in (Schoenauer et al., 2014; Wirth et al., 2016; Christiano et al., 2017), while direct policy learning from humans through trajectory sampling also has been proposed in using a heuristic evolutionary algorithm (Busa-Fekete et al., 2014; Akrour et al., 2011), or from a Bayesian Markov chain Monte Carlo perspective (Wilson et al., 2012). However, it was not until recently that algorithms with provable theoretical guarantees were proposed. From the reward inference approach, Novoseller et al. (2020) took a Bayesian perspective and maintained a posterior distribution over the reward function and the transition kernel, which is computationally costly. Wang et al. (2023) assumes the reward function can be linearly parameterized with known feature embedding and proposes a preference-to-reward interface (a reward model) using an online policy update and a baseline trajectory for comparison. The authors theoretically showed that for a general known preference model setting, RLHF is no harder than RL. A similar analysis framework is adopted in contemporary theoretical works, e.g., (Saha et al., 2023; Zhan et al., 2024a;b; Zhu et al., 2023; Kong & Yang, 2022; Wu & Sun, 2024), for both online, offline, and hybrid RL problems, where the algorithms first learn the linear parameter of the reward function and perform value-based RL on the learned reward function. The analysis first characterizes the error of the reward parameter using the concentration of the MLE estimator and then propagates this error to the value-based RL algorithm. Specifically, Saha et al. (2023) considers the tabular RL setting and assumes a known feature embedding for each trajectory with a known transition kernel. Then, the agent is directed by RL to explore the feature direction where the uncertainty in the reward function is large. Wu & Sun (2024) extends the scenario to linear MDPs with unknown transition and uses least square value iteration to solve the RL problem. Zhu et al. (2023) extends the work (Saha et al., 2023) to the offline RL scenario where a pessimistic estimator of the reward parameter is used to combat the insufficient data coverage in offline settings.

Zhan et al. (2024b) replaces the linear reward parameterization with a general function class under the realizability condition. This also enables them to solve the unknown preference model as long as it is in the known function class. Zhan et al. (2024b) and Kong & Yang (2022) study the hybrid RL problem with human preference, where they first use an exploratory policy from an optimal design perspective to improve the coverage of the offline dataset and to extract more information useful for reward inference. Then, the problem will be solved under the general RL framework. All papers above took a value-based approach in RL and Du et al. (2024) analyzed natural policy gradient with reward inference. Direct policy learning from human preference has been less analyzed compared to reward inference approaches. In tabular MDPs, Xu et al. (2020) and Zhang et al. (2024a) reduce the RL problem to a sequence of dueling bandit problems. However, the approach is only suitable for MDPs with finite state and action spaces. For general MDPs, Chen et al. (2022) first uses function approximation for the mapping from trajectory pair to human preference, and learns the RL transition kernel from a least square estimator with this preference approximator. The optimal policy can then be learned using a dueling bandit approach. However, their results assume the true preference model and transition kernel are inside the known function class with small complexity, which is strong in real applications. RLHF has also been studied in other aspects. Li et al. (2023) studies RL from the human behavior dataset from a dynamic choice perspective. Chakraborty et al. (2024) formulated the reward inference and the policy optimization as a bilevel optimization problem, and Kausik et al. (2024) studies RLHF with partially observed rewards and states. Zhu et al. (2024) studies the overfitting issue in reward model training. Recently, Xie et al. (2024) combined DPO with optimistic exploration to design XPO in the function approximation regime with provable convergence. However, their algorithm and results are still dependent on the DPO loss, which only works for deterministic MDPs.

**Zeroth-Order Optimization and Evolutionary Strategies.** The zeroth-order optimization problem has been studied in the convex and non-convex optimization literature for more than a decade (Ghadimi & Lan, 2013; Nesterov & Spokoiny, 2017), where the stochastic gradient descent algorithm with two-point gradient estimator is most widely used. The convergence rate in smooth functions is first studied in (Ghadimi & Lan, 2013) in both convex and non-convex settings. In the convex setting, the algorithm finds the optimal point, while in the non-convex setting, the algorithm finds a stationary point. The rate of convergence for non-convex functions is improved in Nesterov & Spokoiny (2017). Variants of stochastic gradient descent, such as variance reduction techniques (Liu et al., 2018b) and ADMM (Liu et al., 2018a; Gao et al., 2018) have also been studied in the zeroth-order literature. (Cai et al., 2021) extends the zeroth-order method to blocked coordinate descent for computational efficiency and Liu et al. (2019) extends the method to analyze a signed version of SGD, which is more memory efficient in federated settings since each element of the gradient takes only one bit. In recent years, the study of evolutionary strategies (Rechenberg, 1973) brought zeroth-order optimization methods to optimize RL agents, which has achieved empirical success as a scalable alternative to classic RL algorithms such as Q-learning or policy gradient (Salimans et al., 2017; Conti et al., 2018). This strategy has also been applied in preference-based RL as well (Busa-Fekete et al., 2014; Akrour et al., 2011). However, the theoretical guarantees and provable algorithms remain under-explored. The zeroth-order optimization technique has been proposed in optimizing LLMs (Malladi et al., 2023; Zhang et al., 2024b), but they only implement it in the procedure of policy optimization from the reward network to avoid the heavy memory burden in back-propagation. In their studied problem, the loss function can be explicitly estimated or calculated, and thus can be queried to construct the zeroth-order gradient, which is more similar to classic evolutionary strategies. Our work is different in that the loss function, i.e., the value function, cannot be directly queried without reward feedback, so the zeroth-order algorithms cannot be directly used. However, we view human feedback as a natural source of zeroth-order information and apply the method directly from human preference.

**Preference Models.** The study of modeling the rationale of human decision-making has been pertinent for almost a century. The most popular and widely studied model in the literature is the random utility model (RUM) in social choice theory (Azari et al., 2012), which was first developed as early as 1920s (Thurstone, 1927). In RUM, for each decision-maker, each choice is associated with a utility that consists of two portions: a shared public utility, which is the same for all people, and a private utility (noise) unique to each person. People are modeled as utility maximizers and therefore will choose the choice that has the largest total utility. The private utility is assumed to follow a certain distribution among the population of decision-makers. It is not hard to see that the public utility will determine which choice is more preferred among the population, and the distribution of

the private utility will determine how much it is more preferred, i.e., the population preference probability. Different distribution assumptions give rise to different preference models with different link functions. If the private utility follows a normal distribution, the preference model is called the Probit model, where the link function is the cumulative distribution function of a normal distribution. The Probit model is the first proposed and studied preference model in the literature (Thurstone, 1927). Suppose the private utility follows the Gumbel distribution. We recover the Bradley-Terry model, also known as the logit model, with the logistic link function. Other preference models include the linear model with linear link function, the Weibull model, the Cauchy model, and the complementary log-log model, which have been studied in the literature for specific applications, e.g., see (Train, 2009; Greene, 2000; 2010) for a comprehensive review. In general, when the utility is close among choices, the Bradley-Terry model and the Probit model are often more accurate and easier to implement.

## B  DISCUSSION

**Local and Global Convergence.** Our results only establish the local convergence of both proposed algorithms. In general, considering the generality of stochastic MDPs and policy parameterization studied in our paper, it is extremely difficult to establish global convergence results without further assumptions beyond the ones that have already been made in the paper, such as smoothness. We conjecture that our proposed algorithms will have global convergence under additional assumptions on the value function, such as convexity or the Polyak-Lojasiewicz (PL) condition (Karimi et al., 2016). Again, these assumptions are often questionable in practice. Some contemporary works, such as Xie et al. (2024), established global convergence in the deterministic KL-constrained MDPs with additional coverability assumptions, a much easier and narrower model where the optimal policy can be explicitly expressed. Considering the difficulty and generality of our setting, we think local convergence is the best result that can be obtained, and a meaningful result showing the effectiveness of the proposed algorithm to a wider range of RL problems beyond LLMs. It remains an open question of how to design algorithms that achieve global optimality under minimal and practical assumptions.

**Local and Global Reward Estimation.** The main philosophical difference between our proposed algorithms and classic RLHF paradigms is the use of local reward information around the neighborhood of the current policy. This is completely different from RLHF with reward inference since reward models intend to approximate the global reward landscape of the MDP. Some recently developed online RLHF algorithms, such as online DPO (Guo et al., 2024) and XPO (Xie et al., 2024), also employ certain local reward information to estimate the DPO loss in each iteration. Specifically, the algorithms will explore the trajectories in the neighborhood of the current policy to construct a local estimation of the DPO loss. However, on the other hand, the relation between the DPO loss and the best policy is still global in deterministic MDPs studied in both papers. Our paper's approach is purely local and does not rely on such global relations, which is incorrect in stochastic MDPs. It estimates the local policy gradient directly from human feedback and proceeds with gradient descent for policy updates. The use of local information in our approach is more general and may apply to a wider range of RL problems.

**KL Regularization in RLHF.** The KL-regularization is commonly used in RLHF for both empirical and theoretical studies, e.g., PPO (Ouyang et al., 2022), DPO (Rafailov et al., 2024a;b; Guo et al., 2024; Dong et al., 2024), and more recent algorithms (Xie et al., 2024; Xiong et al., 2024). On the positive side, the KL-regularization improves the training stability in RLHF tasks. For example, in LLM applications, the model pre-training procedure will provide powerful capabilities such as reasoning, and the regularization ensures that the updated policy is unlikely to deviate from the pre-training model and lose these capabilities. It also ensures the global relation between the optimal policy and the reward function in deterministic MDPs for LLM studies and motivates easier objectives, such as the DPO loss. On the other hand, in general RL problems, if the regularization is too large, it also limits the RL agent from the possibility of learning the best policy, which may be very different from the reference policy of the pre-trained model. It is also worth noting that KL-regularized MDPs may be easier to explore compared to their unregularized counterparts, especially for MDPs with sparse rewards. This is because the best policies for regularized MDPs are likely to be stochastic policies that exhibit certain exploration power due to stochasticity. In contrast, the optimal policy for unregularized counterparts is deterministic, which does not explore. It seems

there is no golden rule on whether KL-regularization is indeed needed in RLHF applications beyond LLMs. The answer is likely to depend on specific applications. Our algorithm can also be applied to KL-regularized MDPs after a slight modification in gradient estimation. In this case, the zeroth-order gradient of the policy will include the sum of two components, the value function difference estimation, and the KL regularization difference, i.e.,

$$\left(V(\pi_{\boldsymbol{\theta}_t + \mu \boldsymbol{v}_t}) - \beta \mathsf{KL}(\pi_{\boldsymbol{\theta}_t + \mu \boldsymbol{v}_t} \| \pi_{\boldsymbol{\theta}_0})\right) - \left(V(\pi_{\boldsymbol{\theta}_t}) - \beta \mathsf{KL}(\pi_{\boldsymbol{\theta}_t} \| \pi_{\boldsymbol{\theta}_0})\right),$$

since the objective is now the KL-regularized value function. Here, $\pi_{\boldsymbol{\theta}_0}$ is the reference policy, and $\mathsf{KL}(\pi_{\boldsymbol{\theta}} \| \pi_{\boldsymbol{\theta}_0})$ is the KL divergence between a policy parameterized by $\boldsymbol{\theta}$ and the reference policy, i.e.,

$$\mathsf{KL}(\pi_{\boldsymbol{\theta}} \| \pi_{\boldsymbol{\theta}_0}) = \mathbb{E}_{s_h, a_h \sim \pi_{\boldsymbol{\theta}}} \left[ \sum_{h=1}^{H} \log \frac{\pi_{\boldsymbol{\theta}}(a_h | s_h)}{\pi_{\boldsymbol{\theta}_0}(a_h | s_h)} \right].$$

Remark that human evaluators will only provide preference based on the trajectory returns, not the KL regularization. Therefore, the value function difference component can be estimated in the same way as the current algorithm, i.e., line 9 in Alg. 1. Then, the rest is to estimate the KL regularization difference between the perturbed policy $\pi_{\boldsymbol{\theta}_t + \mu \boldsymbol{v}_t}$ and the current policy $\pi_{\boldsymbol{\theta}_t}$, concerning the reference policy $\pi_{\boldsymbol{\theta}_0}$, i.e.,

$$\mathsf{KL}(\pi_{\boldsymbol{\theta}_t + \mu \boldsymbol{v}_t} \| \pi_{\boldsymbol{\theta}_0}) - \mathsf{KL}(\pi_{\boldsymbol{\theta}_t} \| \pi_{\boldsymbol{\theta}_0})$$

$$= \mathbb{E}_{s_h, a_h \sim \pi_{\boldsymbol{\theta}_t + \mu \boldsymbol{v}_t}} \left[ \sum_{h=1}^{H} \log \frac{\pi_{\boldsymbol{\theta}_t + \mu \boldsymbol{v}_t}(a_h | s_h)}{\pi_{\boldsymbol{\theta}_0}(a_h | s_h)} \right] - \mathbb{E}_{s_h, a_h \sim \pi_{\boldsymbol{\theta}_t}} \left[ \sum_{h=1}^{H} \log \frac{\pi_{\boldsymbol{\theta}_t}(a_h | s_h)}{\pi_{\boldsymbol{\theta}_0}(a_h | s_h)} \right].$$

This can be achieved by estimating the KL regularization term of both policies, which is conducted by evaluating the action probability of the reference policy $\pi_{\boldsymbol{\theta}_0}$ on the trajectories generated from both policies. With both components, we can construct the zeroth-order gradient for the regularized MDP and conduct gradient descent similar to ZPG and ZBCPG.

**Limitations.** Both proposed algorithms require a large number of online human queries for accurate gradient estimation to ensure fast convergence to a good policy, both in theory and in experiments, which may limit their practicality. Such limitation is common in all online RLHF algorithms, including online DPO (Guo et al., 2024). On the other hand, online RLHF algorithms are shown to exhibit better performance, which is also observed in our experiments. It converges faster and requires fewer iterations. One possible approach to avoid this limitation is to replace human feedback with AI feedback similar to Guo et al. (2024). Another possible method is to replace the SGD-style update in the current algorithm with a momentum-based optimization scheme to reuse the human queries from previous policy iterations. The necessity of the current level of human queries will be further explored. Another limitation of our algorithms is the lack of strategic exploration, which may either fail to explore hard-to-reach trajectories or produce similar trajectories that are hard to provide preferences for. One possible way to incorporate better exploration practice into our proposed algorithms is to replace the SGD type of policy update with projected SGD, which forces the agent to explore before convergence. Solving both limitations will be our future direction.

## C  DETAILS OF THE NUMERICAL EXPERIMENT IN SEC. 5

In this section, we describe the GridWorld experiment environment used in our numerical evaluation and the details of the algorithm implementations.

**Environment.** Our GridWorld environment consists of $5 \times 5$ blocks, where the location of each block, i.e., the state space $\mathbb{S}$, is denoted as $(1, 1)$ to $(5, 5)$. For each block, we flip a fair coin. If the head shows, we will place a random reward on top, and if the tail shows, there will be no reward on this block. The reward, if placed in a block, is randomly sampled from a standard normal distribution, which means we allow a negative reward as punishment. Each episode consists of $H = 20$ steps, and at the start of each episode, an agent is positioned in block $(3, 3)$, i.e., the center of the GridWorld environment. At each step, the agent can choose from four actions that constitute the action space $\mathbb{A}$: going up, going down, going left, and going right. However, unlike classic GridWorld environments where the agent can perfectly control its movement, in our experiment, we assume the agent is subject to environment disturbances or imperfect control, to model the stochastic

transition and demonstrate the weaknesses of predominant RLHF frameworks such as DPO. We assume at each step, the action taken by the agent may be reversed with probability $0.4$. For example, if the agent is in the center and chooses to go up, it will land on the block above the current block with a probability of $0.6$ and on the block beneath it with a probability of $0.4$. Similarly, if the agent chooses to go left, it will land on the block left next to it with a probability of $0.6$ and on the block right next to it with a probability of $0.4$. The motivation for imperfect control arises naturally from designing agents for turbulent environments, e.g., for robots operating on airplanes, cars, or ships, where continual wind or bumps may shift the agent's action. The goal of the agent is to maximize the cumulative reward over the $H = 20$ time horizon, and the interaction is conducted episodically.

**Policy Parameterization.** For all algorithms implemented in our experiment, we used tabular policy softmax parameterization, i.e., each state-action pair $(s, a)$ is equipped with a parameter $\xi_{s,a}$ and the policy $\pi(a|s)$ of taking action $a$ at state $s$ would follow:

$$\pi(a|s) = \frac{\exp(\xi_{s,a})}{\sum_a \exp(\xi_{s,a})}.$$

**Human Feedback.** In this experiment, we assume access to a human panel of size $M = 1000$, i.e., a group of human experts, can be recruited and queried at the same time for two trajectories, except for the distributed implementation of ZBCPG, which we will discuss in more detail in the next section. Each human expert, whom we refer to as a panelist, will generate a preference based on the standard Bradley-Terry model over the trajectory rewards. For example, for two trajectories $\tau_0$ and $\tau_1$, each panelist would provide feedback following the distribution as follows:

$$\mathbb{P}(\tau_1 \succ \tau_0) = \frac{\exp(r(\tau_1))}{\exp(r(\tau_1)) + \exp(r(\tau_0))}, \quad \mathbb{P}(\tau_0 \succ \tau_1) = \frac{\exp(r(\tau_0))}{\exp(r(\tau_1)) + \exp(r(\tau_0))}.$$

### C.1 EXPERIMENT COMPARING ZPG AND ZPBCPG TO BASELINES

In this section, we describe the implementation of our proposed algorithms and the implementation of baseline algorithms such as DPO, online DPO, and PPO in Fig. 2(a). For all algorithms, we tune the learning rates and perturbation distances. We also ensure that the total number of trajectory samples and the total number of human queries are the same for fair comparison.

**ZPG.** We implemented ZPG according to Alg. 1 with trim size $\Delta = 0.001$ and batch size $N = 1000$ for $T = 1000$ iterations. After collecting $N$ trajectory pairs for the perturbed policy and the current policy, all panelists were queried to obtain the preference of all trajectory pairs.

**ZBCPG.** We implemented ZBCPG according to Alg. 2 with the same setup as ZPG with trim size $\Delta = 0.001$ and batch size $N = 1000$ for $T = 1000$ iterations. The coordinate batch size $K$ is chosen to be $5 \times 4 = 20$, i.e., at each time we perturb the parameter $\xi_{s,a}$ associated with all actions for $5$ states. After collecting $N$ trajectory pairs for the perturbed policy and the current policy, all panelists were queried to obtain the preference of all trajectory pairs.

**RM+PPO.** The PPO baseline contains two components: reward model training and policy training. The reward model we consider uses tabular parameterization. To train the reward model, we first used the randomly initialized policy to collect $5 \times 10^5$ trajectory pairs, which is half the number of trajectories used in ZPG and ZBCPG. Then, we queried all panelists for the preferences of all trajectory pairs and obtained the population preference probability for all pairs. After that, we used the SGD optimizer to train the reward model for $5$ epochs under the loss in Christiano et al. (2017), which also used multiple human queries for each pair of trajectories. For the policy training, we used a standard online PPO with a KL regularization weight being $0.1$. We also conducted $T = 1000$ policy iterations, and the agent collects $N = 1000$ trajectories between policy updates for improved training stability. The PPO loss in each iteration is optimized using SGD for $5$ epochs.

**DPO.** To train the DPO agent, we also conducted $T = 1000$ policy iterations, and the agent collects $N = 1000$ trajectory pairs between policy updates. The KL regularization weight in the DPO loss is $0.1$. For each pair of trajectories, the agent queried all panelists for preferences and then used the averaged preference probability to replace the single human preference in the vanilla DPO loss, similar to the loss in Christiano et al. (2017), to ensure fair comparison among algorithms. The DPO loss at each iteration is optimized using SGD for $5$ epochs.

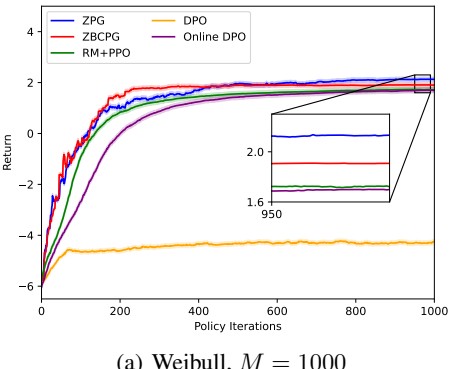 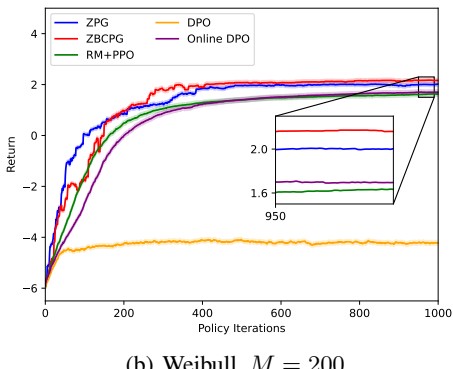

(a) Weibull, $M = 1000$        (b) Weibull, $M = 200$

Figure 3: GridWorld with Weibull Feedback: (a) the return of ZPG, ZBCPG, and RLHF baselines under Weibull human feedback with panel size $M = 1000$, and (b) the trajectory return of ZPG, ZBCPG, and RLHF baselines under Weibull human feedback with panel size $M = 200$. All results are averaged over $10^4$ repetitions of policy evaluation, and shaded areas indicate confidence intervals.

**Online DPO.** The implementation of online DPO is almost the same as offline DPO, except that we replaced the reference policy in the KL regularization with the current policy.

### C.2 EXPERIMENT COMPARING ZPG TO DISTRIBUTED ZBCPG IMPLEMENTATION

In this section, we describe the distributed implementation of ZBCPG, which produced the results in Fig. 2(b). For distributed ZBCPG, we assume multiple parallel agents can collect trajectories and query humans at the same time, and the level of parallelization is controlled by the number of blocks in the legend of Fig. 2(b). For example, when the number of blocks is 5, we divide the parameters in the policy network into 5 blocks. Then, five perturbed policies are created by perturbing each block individually. These policies are distributed to 5 agents, and each agent will collect $N = 1000$ trajectories simultaneously. However, to ensure fairness in the human panel query, we also divide the complete 1000 people panel into 5 sub-panels that could work in parallel, where each sub-panel includes 200 panelists and will assist one agent with human preference queries. Therefore, the parallelized version of ZBCPG only has the advantage of sampling multiple trajectories at the same time. The human panel is also divided, so the overall number of human queries is the same. When the number of blocks is 5, we perturbed the parameters associated with all actions for 5 states at the same time, so each block contains $5 \times 4 = 20$ parameters. When the number of blocks is 25, we perturbed parameters associated with all actions for a single state at the same time, and each block contains 4 parameters. When the number of blocks is 1, we perturbed all parameters with Rademacher noise. The implementation of ZPG and other parameters is the same as in the previous section.

## D ADDITIONAL NUMERICAL EXPERIMENT RESULTS

In this section, we conduct additional numerical experiments to evaluate the impact of the preference model link functions on our proposed algorithms and the baselines, as well as the influence of the size $M$ of the human evaluator panel.

### D.1 WEIBULL HUMAN FEEDBACK

For the first aspect, we consider a Weibull preference feedback mechanism (Train, 2009) which is anti-asymmetric between the two trajectories being compared, i.e., $\sigma(x) \neq 1 - \sigma(-x)$. For example, if two trajectories $(\tau_0, \tau_1)$ are compared by a panelist with the Weibull preference model, it would

| Algorithm | ZPG (Ours) | ZBCPG (Ours) | RM+PPO | DPO | Online DPO |
|-----------|-----------|--------------|--------|-----|-----------|
| Return | $\mathbf{2.13} \pm 0.09$ | $\mathbf{1.90} \pm 0.09$ | $1.72 \pm 0.09$ | $-4.25 \pm 0.09$ | $1.69 \pm 0.09$ |

Table 2: Last Iterate Policy Average Return under Weibull Feedback with $M = 1000$.

prefer the trajectory $\tau_1$ with probability:

$$\mathbb{P}(\tau_1 \succ \tau_0) = \exp_2(-\exp_2(r(\tau_0) - r(\tau_1))).$$

This preference model would capture the bias effect presented in the revealing sequence to the human evaluators. Namely, human experts would unintentionally find it easier to distinguish if the better trajectory is presented first. The implementation of all algorithms is the same as the implementation in Sec. 5 except that we use the Weibull link function in both ZPG and ZBCPG. The results are shown in Fig. 3(a) and Tab. 2. It can be seen that both our algorithms perform much better than the baselines in terms of convergence rate and the quality of the last iterate policy. Even though the trend is similar to the Bradley-Terry feedback results shown in Sec. 5, the performance gap is slightly larger due to the additional model mismatch factor. Both DPO and Online DPO are designed explicitly under the Bradley-Terry model, which is inaccurate for the ground-truth Weibull feedback, and the same applies to the reward model training step in the PPO baseline. However, the performance is not significantly degrading, which implies the Weibull link function is not fundamentally different than the Bradley-Terry model and allows these algorithms to have a compelling performance still. It is also shown that our algorithms are more flexible with the different human preference models and deliver better quality.

## D.2    INFLUENCE OF PANEL SIZE

| Algorithm | ZPG (Ours) | ZBCPG (Ours) | RM+PPO | DPO | Online DPO |
|-----------|-----------|--------------|--------|-----|-----------|
| Return | $\mathbf{2.02} \pm 0.08$ | $\mathbf{2.16} \pm 0.09$ | $1.63 \pm 0.09$ | $-4.22 \pm 0.10$ | $1.69 \pm 0.09$ |

Table 3: Last Iterate Policy Average Return under Weibull Feedback with $M = 200$.

For the second aspect, i.e., to study the influence of panel size $M$ on the quality of the learned policy, and to compare with our theoretical results presented in Theorem. 1 and Theorem 2, we conduct the same experiment with Weibull preference feedback with the same setup as the previous subsection where we only use $M = 200$ human evaluators for each trajectory pair, which is smaller than the $\mathcal{O}(H^2)$ guarantee required by theory. The results are shown in Fig. 3(b) and Tab. 3. It can be seen that even with a much smaller human panel, our proposed algorithms can still converge to a local optimal policy which has a better quality than all benchmarks. Comparing the quality of the last iterate policy in Tab. 3 to Tab. 2, it can also be deduced that the final policy found by our proposed algorithm is robust to the size of the human panel $M$, and the performance with less human evaluators does not significantly degenerate. However, it can also be seen from Fig. 3 that both ZPG and ZBCPG have relatively larger policy quality fluctuation over the training trajectory compared to the three baselines, and this fluctuation increases as the number of human experts $M$ decreases, which poses stability concerns. This is partially because our proposed algorithms do not use KL regularization over the policies to constrain each gradient ascent step, which results in large policy alterations between iterations. This may be mitigated if we apply the KL regularization to our algorithms as discussed in Sec. B.

## E    PROOF BACKBONE FOR THEOREMS: PREFERENCE ESTIMATION ERROR

In the following sections, we provide proofs for both our main theorems, i.e., Theorem. 1 and Theorem. 2 for ZPG (Algorithm. 1) and ZBCPG (Algorithm. 2) respectively. Both algorithms follow the Lyapunov drift analysis framework, where the key to the final result relies on a tight bound on the drift. At each iteration $t$ and for each trajectory pair, both algorithms estimate the population-level human preference probability $p_{t,n}$ and use it with the preference model in equation 2 to estimate the reward difference between trajectories, which is later averaged to estimate the value function

difference as in equation 3. Before presenting the proofs of the theorems, we first characterize the error incurred by preference estimation in the following Lemma:

**Lemma 1 (Concentration of Reward Difference)** *Suppose* $\Delta = \min\{\sigma(-H), 1-\sigma(H)\}$, *for any trajectory pairs* $(\tau_{n,0}, \tau_{n,1})$ *that is queried from $M$ human evaluators. We have:*

$$\mathbb{E}\left|\sigma^{-1}(p_{t,n}) - [r(\tau_{n,1}) - r(\tau_{n,0})]\right| \leq L\sqrt{\frac{2\log M}{M}} + \frac{2H}{M^2}; \tag{5}$$

$$\mathbb{E}\left|\sigma^{-1}(p_{t,n}) - [r(\tau_{n,1}) - r(\tau_{n,0})]\right|^2 \leq \frac{2L^2\log M}{M} + \frac{4H^2}{M^2}. \tag{6}$$

This lemma shows that our reward difference estimation from human preference is accurate as long as the number of human evaluators $M$ is large. The proof of this lemma is provided in Sec. E.1

### E.1 PROOF OF LEMMA. 1

Consider any iteration $t$ and any trajectory pairs $(\tau_{n,0}, \tau_{n,1})$ generated by the original and perturbed policy in both ZPG and ZBCPG. We first bound the estimation error of population-level human preference $\mathbb{P}(\tau_{n,1} \succ \tau_{n,0})$ given any arbitrary trajectory pair in the following Lemma. 2. To formalize, let $\bar{o}_n$ be the empirical average over the human feedback $o_{n,m}$, i.e.,

$$\bar{o}_n = \sum_{m=1}^{M} \frac{o_{n,m}}{M}.$$

Naturally, $\mathbb{E}[\bar{o}_n] = \mathbb{E}[o_{n,m}] = \mathbb{P}(\tau_{n,1} \succ \tau_{n,0})$. And we need to characterize this estimation error first before we prove Lemma. 1.

**Lemma 2 (Concentration of Preference Probability)** *For any* $\delta < \frac{1}{4}$ *and suppose* $\Delta = \min\{\sigma(-H), 1-\sigma(H)\}$, *for any trajectory pairs* $(\tau_{n,0}, \tau_{n,1})$ *that is queried from $M$ human evaluators, with probability at least* $1-\delta$, *we have:*

$$|p_{t,n} - \mathbb{P}(\tau_{n,1} \succ \tau_{n,0})| = |\mathrm{trim}\left[\bar{o}_n|\Delta\right] - \mathbb{P}(\tau_{n,1} \succ \tau_{n,0})| \leq \sqrt{\frac{\log\frac{1}{\delta}}{M}},$$

*where $o_{n,m}$ is the feedback for the $m$-th human evaluator for this trajectory pair.*

**Proof.** First, we notice that for two trajectories $\tau_{n,1}$ and $\tau_{n,0}$, the reward difference of the two trajectories is bounded, i.e., $|r(\tau_{n,1}) - r(\tau_{n,0})| \leq H$, which implies the population-level preference probability $\mathbb{P}(\tau_{n,1} \succ \tau_{n,0})$ is also bounded, i.e.,

$$\mathbb{P}(\tau_{n,1} \succ \tau_{n,0}) \in [\sigma(-H), \sigma(H)] \subset [\Delta, 1-\Delta].$$

Note that $\bar{o}_n$ may not be within $[\Delta, 1-\Delta]$ due to the finite number of human evaluators. Overall, we may have the following three cases:

1. $\bar{o}_n \in [\Delta, 1-\Delta]$: we have $|\mathrm{trim}[\bar{o}_n|\Delta] - \mathbb{P}(\tau_{n,1} \succ \tau_{n,0})| = |\bar{o}_n - \mathbb{E}[\bar{o}_n]|$.
2. $\bar{o}_n > 1 - \Delta \geq \mathbb{P}(\tau_{n,1} \succ \tau_{n,0})$: we have
$$|\mathrm{trim}[\bar{o}_n|\Delta] - \mathbb{P}(\tau_{n,1} \succ \tau_{n,0})| = 1 - \Delta - \mathbb{P}(\tau_{n,1} \succ \tau_{n,0}) \leq |\bar{o}_n - \mathbb{E}[\bar{o}_n]|.$$
3. $\bar{o}_n < \Delta \leq \mathbb{P}(\tau_{n,1} \succ \tau_{n,0})$: we have
$$|\mathrm{trim}[\bar{o}_n|\Delta] - \mathbb{P}(\tau_{n,1} \succ \tau_{n,0})| = \mathbb{P}(\tau_{n,1} \succ \tau_{n,0}) - \Delta \leq |\bar{o}_n - \mathbb{E}[\bar{o}_n]|.$$

To summarize, we have $|\mathrm{trim}[\bar{o}_n|\Delta] - \mathbb{P}(\tau_{n,1} \succ \tau_{n,0})| \leq |\bar{o}_n - \mathbb{E}[\bar{o}_n]|$. According to Hoeffding's inequality for a Bernoulli random variable, we have:

$$\mathbb{P}\left(|\mathrm{trim}[\bar{o}_n|\Delta] - \mathbb{P}(\tau_{n,1} \succ \tau_{n,0})| > \sqrt{\frac{\log\frac{1}{\delta}}{M}}\right) \leq \mathbb{P}\left(|\bar{o}_n - \mathbb{E}[\bar{o}_n]| > \sqrt{\frac{\log\frac{1}{\delta}}{M}}\right)$$

$$\leq \exp\left(-2M\frac{\log\frac{1}{\delta}}{M}\right) \leq \delta,$$

which concludes the proof. ∎

Next, we are ready to bound the error between the reward difference estimator after plugging in the preference probability estimator $p_{t,n}$ to the inverse link function $\sigma^{-1}$ in Lemma. 1. We first prove equation 5 and then prove equation 6. Let $\mathcal{E}_n$ be the concentration event that is the event in Lemma. 2 holds. By Lemma. 2, when the concentration event does not hold, we can bound the term in the expectation of the left-hand side as:

$$\left| \sigma^{-1}(p_{t,n}) - [r(\tau_{n,1}) - r(\tau_{n,0})] \right| \leq 2H.$$

When the concentration event holds, we first notice that $r(\tau_1) - r(\tau_0) = \sigma^{-1}(\mathbb{P}(\tau_{n,1} \succ \tau_{n,0}))$. By definition, both $p_{t,n}$ and $\mathbb{P}(\tau_{n,1} \succ \tau_{n,0})$ belongs to the interval $[\Delta, 1 - \Delta]$. This allows us to use assumption 2 and we have:

$$\begin{aligned} \left| \sigma^{-1}(p_{t,n}) - [r(\tau_{n,1}) - r(\tau_{n,0})] \right| &= \left| \sigma^{-1}(p_{t,n}) - \sigma^{-1}(\mathbb{P}(\tau_{n,1} \succ \tau_{n,0})) \right| \\ &\leq L \left| p_{t,n} - \mathbb{P}(\tau_{n,1} \succ \tau_{n,0})) \right| \\ &\leq L \sqrt{\frac{\log \frac{1}{\delta}}{M}}, \end{aligned}$$

where the last inequality uses the smoothness of the inverse link function, i.e., Lemma. 2. Combining both cases and let $\delta = 1/M^2$, we have:

$$\mathbb{E} \left| \sigma^{-1}(p_{t,n}) - [r(\tau_{n,1}) - r(\tau_{n,0})] \right| \leq \mathbb{P}(\mathcal{E}_n^{\complement}) L \sqrt{\frac{\log \frac{1}{\delta}}{M}} + \mathbb{P}(\mathcal{E}_n) 2H \leq L \sqrt{\frac{2 \log M}{M}} + \frac{2H}{M^2},$$

which concludes the proof of the first inequality. For the second inequality, we apply a similar procedure. Suppose the concentration event $\mathcal{E}_n$ does not hold, then we can bound the term as:

$$\left| \sigma^{-1}(p_{t,n}) - [r(\tau_{n,1}) - r(\tau_{n,0})] \right|^2 \leq 4H^2.$$

This is because the inverse link function is bounded after applying the trim operator to the preference probability estimation. When the concentration holds, we also employ the continuity of the inverse link function as:

$$\left| \sigma^{-1}(p_{t,n}) - [r(\tau_{n,1}) - r(\tau_{n,0})] \right|^2 \leq \left( L \sqrt{\frac{\log \frac{1}{\delta}}{M}} \right)^2 = \frac{L^2 \log \frac{1}{\delta}}{M},$$

Combining both cases and let $\delta = 1/M^2$, we have:

$$\mathbb{E} \left| \sigma^{-1}(p_{t,n}) - [r(\tau_{n,1}) - r(\tau_{n,0})] \right|^2 \leq \mathbb{P}(\mathcal{E}_n^{\complement}) \frac{2L^2 \log M}{M} + \mathbb{P}(\mathcal{E}_n) 4H^2 \leq \frac{2L^2 \log M}{M} + \frac{4H^2}{M^2},$$

which concludes the proof of both inequalities.

## F    PROOF OF THEOREM. 1

Recall that we choose the smoothed value function $V_\mu(\pi_{\boldsymbol{\theta}})$ as the Lyapunov function, i.e.,

$$V_\mu(\pi_{\boldsymbol{\theta}}) = \mathbb{E}_{\boldsymbol{v}'} \left[ V(\pi_{\boldsymbol{\theta} + \mu \boldsymbol{v}'}) \right],$$

where the vector $\boldsymbol{v}'$ is sampled from a $d$-dimensional uniform distribution over a unit ball. We first provide properties of the smoothed function, which is proved in Liu et al. (2018b, Lemma. 1).

**Lemma 3** *Suppose $\boldsymbol{v}$ is sampled from a uniform distribution over a unit sphere and $\boldsymbol{v}'$ is sampled from a uniform distribution over a unit ball both in $d$-dimensional space, and Assumption. 3 holds. Then the smoothed value function $V_\mu(\pi_{\boldsymbol{\theta}})$ defined above satisfies:*

*1. $V_\mu(\pi_{\boldsymbol{\theta}})$ is $L$-smooth and satisfies:*

$$\nabla_{\boldsymbol{\theta}} V_\mu(\pi_{\boldsymbol{\theta}}) = \mathbb{E}_{\boldsymbol{v}} \left[ \frac{d}{\mu} \left( V(\pi_{\boldsymbol{\theta} + \mu \boldsymbol{v}}) - V(\pi_{\boldsymbol{\theta}}) \right) \boldsymbol{v} \right]. \tag{7}$$

2. *For any $\boldsymbol{\theta} \in \mathbb{R}^d$, the function value difference satisfies:*

$$|V_\mu(\pi_{\boldsymbol{\theta}}) - V(\pi_{\boldsymbol{\theta}})| \leq \frac{L\mu^2}{2}. \tag{8}$$

3. *For any $\boldsymbol{\theta} \in \mathbb{R}^d$, the gradient difference satisfies:*

$$\|\nabla_{\boldsymbol{\theta}} V_\mu(\pi_{\boldsymbol{\theta}}) - \nabla_{\boldsymbol{\theta}} V(\pi_{\boldsymbol{\theta}})\|_2 \leq \frac{\mu L d}{2}. \tag{9}$$

4. *For any $\boldsymbol{\theta} \in \mathbb{R}^d$, the gradient noise satisfies:*

$$\mathbb{E}_{\boldsymbol{v}} \left[ \left\| \frac{d}{\mu} \left( V(\pi_{\boldsymbol{\theta}+\mu\boldsymbol{v}}) - V(\pi_{\boldsymbol{\theta}}) \right) \boldsymbol{v} \right\|_2^2 \right] \leq 2d \|\nabla_{\boldsymbol{\theta}} V(\pi_{\boldsymbol{\theta}})\|_2^2 + \frac{\mu^2 L^2 d^2}{2}. \tag{10}$$

Now, we are ready to prove Theorem. 1. We choose $V_\mu(\pi_{\boldsymbol{\theta}})$ to be the Lyapunov function, where $\boldsymbol{v}$ is sampled from a unit ball. We first analyze its drift, from the smoothness of $V_\mu(\pi_{\boldsymbol{\theta}})$ by Lemma. 3, we have the following upper bound:

$$
\begin{aligned}
V_\mu(\pi_{\boldsymbol{\theta}_t}) - V_\mu(\pi_{\boldsymbol{\theta}_{t+1}}) \leq & \langle -\nabla_{\boldsymbol{\theta}} V_\mu(\pi_{\boldsymbol{\theta}_t}), \boldsymbol{\theta}_{t+1} - \boldsymbol{\theta}_t \rangle + \frac{L}{2} \|\boldsymbol{\theta}_{t+1} - \boldsymbol{\theta}_t\|_2^2 \\
= & -\alpha \langle \nabla_{\boldsymbol{\theta}} V_\mu(\pi_{\boldsymbol{\theta}_t}), \hat{\boldsymbol{g}}_t \rangle + \frac{L\alpha^2}{2} \|\hat{\boldsymbol{g}}_t\|_2^2 \\
= & -\alpha \|\nabla_{\boldsymbol{\theta}} V_\mu(\pi_{\boldsymbol{\theta}_t})\|_2^2 + \alpha \langle \nabla_{\boldsymbol{\theta}} V_\mu(\pi_{\boldsymbol{\theta}_t}), \nabla_{\boldsymbol{\theta}} V_\mu(\pi_{\boldsymbol{\theta}_t}) - \hat{\boldsymbol{g}}_t \rangle + \frac{L\alpha^2}{2} \|\hat{\boldsymbol{g}}_t\|_2^2.
\end{aligned}
$$

We bound the three terms above one by one, the first term can be bounded using equation 7 from Lemma. 3. We first have:

$$
\begin{aligned}
\|\nabla_{\boldsymbol{\theta}} V(\pi_{\boldsymbol{\theta}_t})\|_2^2 = & \|\nabla_{\boldsymbol{\theta}} V(\pi_{\boldsymbol{\theta}_t}) - \nabla_{\boldsymbol{\theta}} V_\mu(\pi_{\boldsymbol{\theta}_t}) + \nabla_{\boldsymbol{\theta}} V_\mu(\pi_{\boldsymbol{\theta}_t})\|_2^2 \\
\leq & 2\|\nabla_{\boldsymbol{\theta}} V_\mu(\pi_{\boldsymbol{\theta}_t}) - \nabla_{\boldsymbol{\theta}} V(\pi_{\boldsymbol{\theta}_t})\|_2^2 + 2\|\nabla_{\boldsymbol{\theta}} V_\mu(\pi_{\boldsymbol{\theta}_t})\|_2^2 \\
\leq & 2\|\nabla_{\boldsymbol{\theta}} V_\mu(\pi_{\boldsymbol{\theta}_t})\|_2^2 + \frac{\mu^2 L^2 d^2}{2},
\end{aligned}
$$

where the first inequality uses $(a+b)^2 \leq 2a^2 + 2b^2$ and the second inequality uses equation 9. This implies:

$$\|\nabla_{\boldsymbol{\theta}} V_\mu(\pi_{\boldsymbol{\theta}_t})\|_2^2 \geq \frac{1}{2} \|\nabla_{\boldsymbol{\theta}} V(\pi_{\boldsymbol{\theta}_t})\|_2^2 - \frac{\mu^2 L^2 d^2}{4}.$$

And we have a bound on the negative drift term as:

$$-\alpha \|\nabla_{\boldsymbol{\theta}} V_\mu(\pi_{\boldsymbol{\theta}_t})\|_2^2 \leq -\frac{\alpha}{2} \|\nabla_{\boldsymbol{\theta}} V(\pi_{\boldsymbol{\theta}_t})\|_2^2 + \frac{\alpha\mu^2 L^2 d^2}{4}.$$

We take a conditional expectation of the drift over the natural filtration $\mathcal{F}_t$ of time $t$ and obtain:

$$
\begin{aligned}
\mathbb{E}\left[V_\mu(\pi_{\boldsymbol{\theta}_t}) - V_\mu(\pi_{\boldsymbol{\theta}_{t+1}})|\mathcal{F}_t\right] \leq & -\frac{\alpha}{2} \|\nabla_{\boldsymbol{\theta}} V(\pi_{\boldsymbol{\theta}_t})\|_2^2 + \frac{\alpha\mu^2 L^2 d^2}{4} + \frac{L\alpha^2}{2} \underbrace{\mathbb{E}\left[\|\hat{\boldsymbol{g}}_t\|_2^2 \big| \mathcal{F}_t\right]}_{\mathsf{Var}_t} \\
& + \alpha \underbrace{\langle \nabla_{\boldsymbol{\theta}} V_\mu(\pi_{\boldsymbol{\theta}_t}), \nabla_{\boldsymbol{\theta}} V_\mu(\pi_{\boldsymbol{\theta}_t}) - \mathbb{E}\left[\hat{\boldsymbol{g}}_t | \mathcal{F}_t\right] \rangle}_{\mathsf{Bias}_t}.
\end{aligned} \tag{11}
$$

Then, we analyze the two positive drift terms $\mathsf{Bias}_t$ for the first-order gradient bias and $\mathsf{Var}_t$ for the gradient noise separately and then construct a tight upper bound of the Lyapunov drift. The results are summarized in the following two lemmas, where the proofs are deferred to Sec. F.1 and Sec. F.2:

**Lemma 4** *For ZPG and let $M \geq 8(H/L)^{\frac{2}{3}}$, conditioned on the information filtration $\mathcal{F}_t$ of any time $t$, the gradient bias can be upper bounded as follows:*

$$
\begin{aligned}
\mathsf{Bias}_t = & \langle \nabla_{\boldsymbol{\theta}} V_\mu(\pi_{\boldsymbol{\theta}_t}), \nabla_{\boldsymbol{\theta}} V_\mu(\pi_{\boldsymbol{\theta}_t}) - \mathbb{E}\left[\hat{\boldsymbol{g}}_t | \mathcal{F}_t\right] \rangle \\
\leq & \frac{2dL}{\mu} \sqrt{\frac{\log M}{M}} \|\nabla_{\boldsymbol{\theta}} V(\pi_{\boldsymbol{\theta}_t})\|_2 + d^2 L^2 \sqrt{\frac{\log M}{M}}.
\end{aligned}
$$

**Lemma 5** *For ZPG and let $M \geq 4(H/L)^2$, conditioned on the information filtration $\mathcal{F}_t$ of any time $t$, the gradient bias can be upper bounded as follows:*

$$\mathsf{Var}_t = \mathbb{E}\left[\left\|\hat{\boldsymbol{g}}_t\right\|_2^2 \middle| \mathcal{F}_t\right] \leq 6d\left\|\nabla_{\boldsymbol{\theta}}V(\pi_{\boldsymbol{\theta}_t})\right\|_2^2 + \frac{3\mu^2 L^2 d^2}{2} + \frac{9d^2 L^2 \log M}{\mu^2 M} + \frac{12d^2 H^2}{\mu^2 N}.$$

Now we use Lemma. 4 and Lemma. 5 and combine the drift bound in equation 11 and obtain:

$$\mathbb{E}\left[V_\mu(\pi_{\boldsymbol{\theta}_t}) - V_\mu(\pi_{\boldsymbol{\theta}_{t+1}})|\mathcal{F}_t\right]$$
$$\leq -\frac{\alpha}{2}\|\nabla_{\boldsymbol{\theta}}V(\pi_{\boldsymbol{\theta}_t})\|_2^2 + \frac{\alpha\mu^2 L^2 d^2}{4} + \alpha\mathsf{Bias}_t + \frac{L\alpha^2}{2}\mathsf{Var}_t$$
$$\leq -\frac{\alpha}{2}\|\nabla_{\boldsymbol{\theta}}V(\pi_{\boldsymbol{\theta}_t})\|_2^2 + \frac{\alpha\mu^2 L^2 d^2}{4} + \frac{2\alpha dL}{\mu}\sqrt{\frac{\log M}{M}}\|\nabla_{\boldsymbol{\theta}}V(\pi_{\boldsymbol{\theta}_t})\|_2 + \alpha d^2 L^2\sqrt{\frac{\log M}{M}}$$
$$+ 3\alpha^2 dL\|\nabla_{\boldsymbol{\theta}}V(\pi_{\boldsymbol{\theta}_t})\|_2^2 + \alpha^2\mu^2 L^3 d^2 + \frac{5\alpha^2 d^2 L^3 \log M}{\mu^2 M} + \frac{6\alpha^2 d^2 H^2 L}{\mu^2 N}.$$

Let $\alpha \leq (12dL)^{-1}$, we can further simplify the drift bound as:

$$\mathbb{E}\left[V_\mu(\pi_{\boldsymbol{\theta}_t}) - V_\mu(\pi_{\boldsymbol{\theta}_{t+1}})|\mathcal{F}_t\right]$$
$$\leq -\frac{\alpha}{2}\|\nabla_{\boldsymbol{\theta}}V(\pi_{\boldsymbol{\theta}_t})\|_2^2 + \frac{\alpha\mu^2 L^2 d^2}{4} + \frac{2\alpha dL}{\mu}\sqrt{\frac{\log M}{M}}\|\nabla_{\boldsymbol{\theta}}V_\mu(\pi_{\boldsymbol{\theta}_t})\|_2$$
$$+ \alpha d^2 L^2\sqrt{\frac{\log M}{M}} + \frac{1}{4}\alpha\|\nabla_{\boldsymbol{\theta}}V(\pi_{\boldsymbol{\theta}_t})\|_2^2 + \alpha\mu^2 L^2 d + \frac{5\alpha^2 d^2 L^3 \log M}{\mu^2 M} + \frac{6\alpha^2 d^2 H^2 L}{\mu^2 N}$$
$$\leq -\frac{\alpha}{4}\|\nabla_{\boldsymbol{\theta}}V(\pi_{\boldsymbol{\theta}_t})\|_2^2 + \frac{\alpha\mu^2 L^2 d^2}{2} + \frac{2\alpha dL}{\mu}\sqrt{\frac{\log M}{M}}\|\nabla_{\boldsymbol{\theta}}V_\mu(\pi_{\boldsymbol{\theta}_t})\|_2 + \alpha d^2 L^2\sqrt{\frac{\log M}{M}}$$
$$+ \frac{5\alpha^2 d^2 L^3 \log M}{\mu^2 M} + \frac{6\alpha^2 d^2 H^2 L}{\mu^2 N},$$

where the last inequality merges both positive and negative drifts on the square of the gradient together and assumes $d \geq 4$. When $\|\nabla_{\boldsymbol{\theta}}V_\mu(\pi_{\boldsymbol{\theta}_t})\|_2$ is large, typically, when we have:

$$\|\nabla_{\boldsymbol{\theta}}V_\mu(\pi_{\boldsymbol{\theta}_t})\|_2 \geq \frac{16dL}{\mu}\sqrt{\frac{\log M}{M}},$$

we have the positive first-order drift regarding the gradient norm is bounded by:

$$\frac{2\alpha dL}{\mu}\sqrt{\frac{\log M}{M}}\|\nabla_{\boldsymbol{\theta}}V_\mu(\pi_{\boldsymbol{\theta}_t})\|_2 \leq \frac{\alpha}{8}\|\nabla_{\boldsymbol{\theta}}V_\mu(\pi_{\boldsymbol{\theta}_t})\|_2^2,$$

and thus can be merged into the negative drift as follows:

$$\mathbb{E}\left[V_\mu(\pi_{\boldsymbol{\theta}_t}) - V_\mu(\pi_{\boldsymbol{\theta}_{t+1}})|\mathcal{F}_t\right]$$
$$\leq -\frac{\alpha}{8}\|\nabla_{\boldsymbol{\theta}}V(\pi_{\boldsymbol{\theta}_t})\|_2^2 + \frac{\alpha\mu^2 L^2 d^2}{2} + \alpha d^2 L^2\sqrt{\frac{\log M}{M}} + \frac{5\alpha^2 d^2 L^3 \log M}{\mu^2 M} + \frac{6\alpha^2 d^2 H^2 L}{\mu^2 N}.$$

If on the other hand, the gradient is small, i.e., we have:

$$\|\nabla_{\boldsymbol{\theta}}V_\mu(\pi_{\boldsymbol{\theta}_t})\|_2 \leq \frac{16dL}{\mu}\sqrt{\frac{\log M}{M}},$$

and then we can upper bound the first-order drift regarding gradient norm in another way:

$$\frac{2\alpha dL}{\mu}\sqrt{\frac{\log M}{M}}\|\nabla_{\boldsymbol{\theta}}V_\mu(\pi_{\boldsymbol{\theta}_t})\|_2 \leq \frac{32\alpha d^2 L^2 \log M}{\mu^2 M}.$$

So in this case, the total drift can be upper bounded as:

$$\mathbb{E}\left[V_\mu(\pi_{\boldsymbol{\theta}_t}) - V_\mu(\pi_{\boldsymbol{\theta}_{t+1}})|\mathcal{F}_t\right]$$

$$\leq -\frac{\alpha}{4}\|\nabla_{\boldsymbol{\theta}}V(\pi_{\boldsymbol{\theta}_t})\|_2^2 + \frac{\alpha\mu^2 L^2 d^2}{2} + \frac{32\alpha d^2 L^2 \log M}{\mu^2 M} + \alpha d^2 L^2 \sqrt{\frac{\log M}{M}}$$

$$+ \frac{5\alpha^2 d^2 L^3 \log M}{\mu^2 M} + \frac{6\alpha^2 d^2 H^2 L}{\mu^2 N}$$

$$\leq -\frac{\alpha}{4}\|\nabla_{\boldsymbol{\theta}}V(\pi_{\boldsymbol{\theta}_t})\|_2^2 + \frac{\alpha\mu^2 L^2 d^2}{2} + \frac{33\alpha d^2 L^2 \log M}{\mu^2 M} + \alpha d^2 L^2 \sqrt{\frac{\log M}{M}} + \frac{6\alpha^2 d^2 H^2 L}{\mu^2 N}.$$

The last inequality uses $\alpha \leq (12dL)^{-1}$ and $d \geq 4$. So by taking a maximum of both bounds and taking an expectation, we have:

$$\mathbb{E}\left[V_\mu(\pi_{\boldsymbol{\theta}_t})\right] - \mathbb{E}\left[V_\mu(\pi_{\boldsymbol{\theta}_{t+1}})\right]$$

$$\leq -\frac{\alpha}{8}\mathbb{E}\left[\|\nabla_{\boldsymbol{\theta}}V(\pi_{\boldsymbol{\theta}_t})\|_2^2\right] + \frac{33\alpha d^2 L^2 \log M}{\mu^2 M} + \frac{\alpha\mu^2 L^2 d^2}{2} + \alpha d^2 L^2 \sqrt{\frac{\log M}{M}} + \frac{6\alpha^2 d^2 H^2 L}{\mu^2 N}.$$

We then take a telescoping sum, which results in:

$$\mathbb{E}\left[V_\mu(\pi_{\boldsymbol{\theta}_0})\right] - \mathbb{E}\left[V_\mu(\pi_{\boldsymbol{\theta}_T})\right] \leq -\frac{\alpha}{8}\sum_{t=0}^{T-1}\mathbb{E}\left[\|\nabla_{\boldsymbol{\theta}}V(\pi_{\boldsymbol{\theta}_t})\|_2^2\right]$$

$$+ \left(\frac{33\alpha d^2 L^2 \log M}{\mu^2 M} + \frac{\alpha\mu^2 L^2 d^2}{2} + \alpha d^2 L^2 \sqrt{\frac{\log M}{M}} + \frac{6\alpha^2 d^2 H^2 L}{\mu^2 N}\right)T.$$

We choose $\mu$ to balance the terms inside the parentheses. Specifically, let

$$\mu^2 = \max\left\{9\sqrt{\frac{\log M}{M}}, \frac{4H}{L\sqrt{dN}}\right\}, \quad \alpha = \frac{1}{12dL}.$$

Then, we have the following inequality on the positive drift:

$$\frac{\alpha\mu^2 L^2 d^2}{2} \geq \frac{33\alpha d^2 L^2 \log M}{\mu^2 M}, \quad \frac{\alpha\mu^2 L^2 d^2}{2} \geq \frac{6\alpha^2 d^2 H^2 L}{\mu^2 N}.$$

So we have the final results in Theorem. 1 as:

$$\frac{1}{T}\sum_{t=0}^{T-1}\mathbb{E}\left[\|\nabla_{\boldsymbol{\theta}}V(\pi_{\boldsymbol{\theta}_t})\|_2^2\right]$$

$$\leq \frac{8\left(\mathbb{E}\left[V_\mu(\pi_{\boldsymbol{\theta}_0})\right] - \mathbb{E}\left[V_\mu(\pi_{\boldsymbol{\theta}_T})\right]\right)}{T\alpha} + 12\mu^2 L^2 d^2 + 8d^2 L^2 \sqrt{\frac{\log M}{M}}$$

$$\leq \frac{8\left(\mathbb{E}\left[V(\pi_{\boldsymbol{\theta}_0})\right] - \mathbb{E}\left[V(\pi_{\boldsymbol{\theta}_T})\right]\right)}{T\alpha} + \frac{8\mu^2 L}{T\alpha} + 12\mu^2 L^2 d^2 + 8d^2 L^2 \sqrt{\frac{\log M}{M}}$$

$$= \mathcal{O}\left(\frac{HLd}{T} + \frac{d^2 L^2 \sqrt{\log M}}{\sqrt{M}} + \frac{HLd\sqrt{d}}{\sqrt{N}}\right).$$

## F.1 PROOF OF LEMMA. 4

We analyze the first-order gradient bias term $\mathsf{Bias}_t$ in the drift of equation 11. Notice that conditioned over $\mathcal{F}_t$, the randomness only comes from sampling the perturbation direction, sampling trajectories, and obtaining human feedback. By equation 7 and the definition of $\hat{\boldsymbol{g}}_t$, we have:

$$\mathbb{E}\left[\nabla_{\boldsymbol{\theta}}V_\mu(\pi_{\boldsymbol{\theta}_t}) - \hat{\boldsymbol{g}}_t|\mathcal{F}_t\right]$$

$$= \mathbb{E}\left[\frac{d\left(V(\pi_{\boldsymbol{\theta}_t + \mu\boldsymbol{v}_t}) - V(\pi_{\boldsymbol{\theta}_t})\right)}{\mu}\boldsymbol{v}_t\bigg|\mathcal{F}_t\right] - \mathbb{E}\left[\frac{d\sum_{n=1}^N \sigma^{-1}(p_{t,n})}{\mu N}\boldsymbol{v}_t\bigg|\mathcal{F}_t\right].$$

Since each trajectory pair $(\tau_{n,0}, \tau_{n,1})$ is generated independently from other trajectories, we have:

$$
\begin{aligned}
\mathbb{E}\left[\left.\frac{d}{\mu}\frac{\sum_{n=1}^{N}\sigma^{-1}(p_{t,n})}{N}\boldsymbol{v}_t\right|\mathcal{F}_t\right] =&\frac{d}{\mu}\mathbb{E}\left[\left.\mathbb{E}\left[\left.\frac{\sum_{n=1}^{N}\sigma^{-1}(p_{t,n})}{N}\right|\boldsymbol{v}_t\right]\boldsymbol{v}_t\right|\mathcal{F}_t\right]\\
=&\frac{d}{\mu}\mathbb{E}\left[\mathbb{E}\left[\left.\sigma^{-1}(p_{t,n})\right|\boldsymbol{v}_t\right]\boldsymbol{v}_t\right|\mathcal{F}_t\right]\\
=&\frac{d}{\mu}\mathbb{E}\left[\mathbb{E}\left[\left.(r(\tau_{n,1})-r(\tau_{n,0}))\right|\boldsymbol{v}_t\right]\boldsymbol{v}_t\right|\mathcal{F}_t\right]\\
&-\frac{d}{\mu}\mathbb{E}\left[\mathbb{E}\left[\left.\left(\sigma^{-1}(p_{t,n})-[r(\tau_{n,1})-r(\tau_{n,0})]\right)\right|\boldsymbol{v}_t\right]\boldsymbol{v}_t\right|\mathcal{F}_t\right],
\end{aligned}
$$

where the first equality uses the law of total expectation. The second equality uses the independent nature of the trajectories generated by the same policy pair. In the last inequality, we aim to bound the difference between the reward difference estimate from human preference and the ground-truth reward difference, which can be constructed through concentration inequalities. Substitute it back into the bias, we have:

$$
\begin{aligned}
\mathbb{E}\left[\left.\nabla_{\boldsymbol{\theta}}V_\mu(\pi_{\boldsymbol{\theta}_t})-\hat{\boldsymbol{g}}_t\right|\mathcal{F}_t\right] =&\frac{d}{\mu}\mathbb{E}\left[\mathbb{E}\left[\left.(V(\pi_{\boldsymbol{\theta}_t+\mu\boldsymbol{v}_t})-V(\pi_{\boldsymbol{\theta}_t})-r(\tau_{n,1})+r(\tau_{n,0}))\right|\boldsymbol{v}_t\right]\boldsymbol{v}_t\right|\mathcal{F}_t\right]\\
&+\frac{d}{\mu}\mathbb{E}\left[\mathbb{E}\left[\left.\left(\sigma^{-1}(p_{t,n})-[r(\tau_{n,1})-r(\tau_{n,0})]\right)\right|\boldsymbol{v}_t\right]\boldsymbol{v}_t\right|\mathcal{F}_t\right],
\end{aligned}
$$

Notice that in the inner expectation of the first term, the randomness comes from sampling trajectories given both policies $\pi_{\boldsymbol{\theta}_t+\mu\boldsymbol{v}_t}$ and $\pi_{\boldsymbol{\theta}_t}$, we have:

$$
\begin{aligned}
&\mathbb{E}\left[\left.(V(\pi_{\boldsymbol{\theta}_t+\mu\boldsymbol{v}_t})-V(\pi_{\boldsymbol{\theta}_t})-r(\tau_{n,1})+r(\tau_{n,0}))\right|\boldsymbol{v}_t\right]\\
=&V(\pi_{\boldsymbol{\theta}_t+\mu\boldsymbol{v}_t})-\mathbb{E}\left[r(\tau_{n,1})|\tau_{n,1}\sim\pi_{\boldsymbol{\theta}_t+\mu\boldsymbol{v}_t}\right]+\mathbb{E}\left[r(\tau_{n,0})|\tau_{n,0}\sim\pi_{\boldsymbol{\theta}_t}\right]-V(\pi_{\boldsymbol{\theta}_t})\\
=&0.
\end{aligned}
$$

The last equality is due to the definition of the value function, where trajectory returns are unbiased estimates of the value functions and the fact that $\tau_{n,1}$ and $\tau_{n,0}$ are generated by the perturbed policy and the original policy independently. Then, the bias drift only comes from the preference estimation error, and we will be able to plug it back and have:

$$
\begin{aligned}
\text{Bias}_t =&\langle\nabla_{\boldsymbol{\theta}}V_\mu(\pi_{\boldsymbol{\theta}_t}),\nabla_{\boldsymbol{\theta}}V_\mu(\pi_{\boldsymbol{\theta}_t})-\mathbb{E}\left[\left.\hat{\boldsymbol{g}}_t\right|\mathcal{F}_t\right]\rangle\\
=&\frac{d}{\mu}\mathbb{E}\left[\left.\left(\sigma^{-1}(p_{t,n})-[r(\tau_{n,1})-r(\tau_{n,0})]\right)\langle\nabla_{\boldsymbol{\theta}}V_\mu(\pi_{\boldsymbol{\theta}_t}),\boldsymbol{v}_t\rangle\right|\mathcal{F}_t\right]\\
\leq&\frac{d}{\mu}\|\nabla_{\boldsymbol{\theta}}V_\mu(\pi_{\boldsymbol{\theta}_t})\|_2\,\mathbb{E}\left[\left.\mathbb{E}\left[\left.\left|\sigma^{-1}(p_{t,n})-[r(\tau_{n,1})-r(\tau_{n,0})]\right|\right|\tau_{n,1},\tau_{n,0}\right]\right|\mathcal{F}_t\right]\\
\leq&\frac{d}{\mu}\|\nabla_{\boldsymbol{\theta}}V_\mu(\pi_{\boldsymbol{\theta}_t})\|_2\left(L\sqrt{\frac{2\log M}{M}}+\frac{2H}{M^2}\right)\\
\leq&\frac{2dL}{\mu}\|\nabla_{\boldsymbol{\theta}}V_\mu(\pi_{\boldsymbol{\theta}_t})\|_2\sqrt{\frac{\log M}{M}}\\
\leq&\frac{2dL}{\mu}\sqrt{\frac{\log M}{M}}\|\nabla_{\boldsymbol{\theta}}V(\pi_{\boldsymbol{\theta}_t})\|_2+d^2L^2\sqrt{\frac{\log M}{M}},
\end{aligned}
$$

where the first inequality uses Cauchy-Schwarz inequality and the second inequality uses Lemma. 1. The second last inequality is obtained by choosing $M\geq 8(H/L)^{\frac{2}{3}}$. The final inequality is obtained by applying the third property of Lemma. 3.

## F.2 PROOF OF LEMMA. 5

We aim for the variance term $\mathsf{Var}_t$ in the Lyapunov drift in equation 11. We first have an upper bound variance as follows:

$$
\begin{aligned}
\mathsf{Var}_t =& \mathbb{E}\left[\left\|\frac{d}{\mu}\frac{\sum_{n=1}^{N}\sigma^{-1}(p_{t,n})}{N}\boldsymbol{v}_t\right\|_2^2 \middle| \mathcal{F}_t\right]\\
\leq& 3\,\mathbb{E}\underbrace{\left[\left\|\frac{d\sum_{n=1}^{N}\left(\sigma^{-1}(p_{t,n}) - [r(\tau_{n,1}) - r(\tau_{n,0})]\right)}{\mu N}\boldsymbol{v}_t\right\|_2^2 \middle| \mathcal{F}_t\right]}_{\mathsf{Var}_{t,1}}\\
&+ 3\,\mathbb{E}\underbrace{\left[\left\|\frac{d}{\mu}\left(\frac{\sum_{n=1}^{N}\left(r(\tau_{n,1}) - r(\tau_{n,0})\right)}{N} - \left(V(\pi_{\boldsymbol{\theta}_t+\mu\boldsymbol{v}_t}) - V(\pi_{\boldsymbol{\theta}_t})\right)\right)\boldsymbol{v}_t\right\|_2^2 \middle| \mathcal{F}_t\right]}_{\mathsf{Var}_{t,2}}\\
&+ 3\,\mathbb{E}\underbrace{\left[\left\|\frac{d}{\mu}\left(V(\pi_{\boldsymbol{\theta}_t+\mu\boldsymbol{v}_t}) - V(\pi_{\boldsymbol{\theta}_t})\right)\boldsymbol{v}_t\right\|_2^2 \middle| \mathcal{F}_t\right]}_{\mathsf{Var}_{t,3}}.
\end{aligned}
$$

The first term comes from the human preference estimation error, which can be bounded using equation 6 from Lemma. 1 as follows:

$$
\begin{aligned}
\mathsf{Var}_{t,1} =& \mathbb{E}\left[\left\|\frac{d\sum_{n=1}^{N}\left(\sigma^{-1}(p_{t,n}) - [r(\tau_{n,1}) - r(\tau_{n,0})]\right)}{\mu N}\boldsymbol{v}_t\right\|_2^2 \middle| \mathcal{F}_t\right]\\
=& \frac{d^2}{\mu^2 N^2}\mathbb{E}\left[\left|\sum_{n=1}^{N}\left(\sigma^{-1}(p_{t,n}) - [r(\tau_{n,1}) - r(\tau_{n,0})]\right)\right|^2\right]\\
\leq& \frac{d^2}{\mu^2 N}\sum_{n=1}^{N}\mathbb{E}\left[\left|\sigma^{-1}(p_{t,n}) - [r(\tau_{n,1}) - r(\tau_{n,0})]\right|^2\right]\\
\leq& \frac{d^2}{\mu^2}\left(\frac{2L^2\log M}{M} + \frac{4H^2}{M^2}\right)\\
\leq& \frac{3d^2 L^2\log M}{\mu^2 M},
\end{aligned}
$$

where the first inequality uses Cauchy-Schwarz inequality, and the second inequality uses Lemma. 1. The final inequality is true by choosing $M \geq 4(H/L)^2$. The second term comes from using the empirical average reward to estimate the value function, similar to REINFORCE. So we have:

$$
\begin{aligned}
\mathsf{Var}_{t,2} =& \mathbb{E}\left[\left\|\frac{d}{\mu}\left(\frac{\sum_{n=1}^{N}\left(r(\tau_{n,1}) - r(\tau_{n,0})\right)}{N} - \left(V(\pi_{\boldsymbol{\theta}_t+\mu\boldsymbol{v}_t}) - V(\pi_{\boldsymbol{\theta}_t})\right)\right)\boldsymbol{v}_t\right\|_2^2 \middle| \mathcal{F}_t\right]\\
=& \frac{d^2}{\mu^2 N^2}\mathbb{E}\left[\left|\sum_{n=1}^{N}\underbrace{\left((r(\tau_{n,1}) - r(\tau_{n,0})) - (V(\pi_{\boldsymbol{\theta}_t+\mu\boldsymbol{v}_t}) - V(\pi_{\boldsymbol{\theta}_t}))\right)}_{\mathsf{E}_{t,n}}\right|_2^2 \|\boldsymbol{v}_t\|_2^2 \middle| \mathcal{F}_t\right]\\
=& \frac{d^2}{\mu^2 N^2}\mathbb{E}\left[\left|\sum_{n=1}^{N}\mathsf{E}_{t,n}\right|_2^2 \middle| \mathcal{F}_t\right],
\end{aligned}
$$

where the last equality is because $\boldsymbol{v}_t$ is sampled from a unit ball. We open up the square as:

$$
\mathbb{E}\left[\left|\left|\sum_{n=1}^{N}\mathsf{E}_{t,n}\right|\right|_2^2\Bigg|\,\mathcal{F}_t\right] = \sum_{n=1}^{N}\mathbb{E}\left[\left|\mathsf{E}_{t,n}\right|_2^2\Big|\,\mathcal{F}_t\right] + \sum_{i\neq j}\mathbb{E}\left[\mathbb{E}[\mathsf{E}_{t,i}\mathsf{E}_{t,j}|\boldsymbol{v}_t]|\,\mathcal{F}_t\right]
$$

$$
= \sum_{n=1}^{N}\mathbb{E}\left[\left|\mathsf{E}_{t,n}\right|_2^2\Big|\,\mathcal{F}_t\right] + \sum_{i\neq j}\mathbb{E}\left[\mathbb{E}[\mathsf{E}_{t,i}|\boldsymbol{v}_t]|\,\mathcal{F}_t\right]\mathbb{E}\left[\mathbb{E}[\mathsf{E}_{t,j}|\boldsymbol{v}_t]|\,\mathcal{F}_t\right]
$$

$$
= \sum_{n=1}^{N}\mathbb{E}\left[\left|\mathsf{E}_{t,n}\right|_2^2\Big|\,\mathcal{F}_t\right].
$$

where the second inequality uses the independence between trajectories generated at the same step, and the last equality is due to $\mathbb{E}[(r(\tau_{n,1}) - r(\tau_{n,0})) - (V(\pi_{\boldsymbol{\theta}_t + \mu\boldsymbol{v}_t}) - V(\pi_{\boldsymbol{\theta}_t}))] = 0$ for any fixed perturbation direction $\boldsymbol{v}_t$ from the definition of value function. So we have:

$$
\mathsf{Var}_{t,2} = \frac{d^2}{\mu^2 N^2}\sum_{n=1}^{N}\mathbb{E}\left[\left|\mathsf{E}_{t,n}\right|_2^2\Big|\,\mathcal{F}_t\right]
$$

$$
= \frac{d^2}{\mu^2 N^2}\sum_{n=1}^{N}\mathbb{E}\left[\left|(r(\tau_{n,1}) - r(\tau_{n,0})) - (V(\pi_{\boldsymbol{\theta}_t + \mu\boldsymbol{v}_t}) - V(\pi_{\boldsymbol{\theta}_t}))\right|_2^2\Big|\,\mathcal{F}_t\right]
$$

$$
\leq \frac{4d^2 H^2}{\mu^2 N},
$$

where the last inequality uses the fact that both the difference of reward and the difference of value function are within $[-H, H]$. The last term to bound $\mathsf{Var}_t$ can be obtained from the property of the smoothed function in Lemma. 3 as:

$$
\mathsf{Var}_{t,3} = \mathbb{E}\left[\left|\left|\frac{d}{\mu}\left(V(\pi_{\boldsymbol{\theta}_t + \mu\boldsymbol{v}_t}) - V(\pi_{\boldsymbol{\theta}_t})\right)\boldsymbol{v}_t\right|\right|_2^2\Bigg|\,\mathcal{F}_t\right] \leq 2d\,\|\nabla_{\boldsymbol{\theta}}V(\pi_{\boldsymbol{\theta}_t})\|_2^2 + \frac{\mu^2 L^2 d^2}{2}.
$$

So, combining three terms, the variance drift can be bounded as:

$$
\mathsf{Var}_t \leq 6d\,\|\nabla_{\boldsymbol{\theta}}V(\pi_{\boldsymbol{\theta}_t})\|_2^2 + \frac{3\mu^2 L^2 d^2}{2} + \frac{9d^2 L^2\log M}{\mu^2 M} + \frac{12d^2 H^2}{\mu^2 N}.
$$

## G  PROOF OF THEOREM. 2

Due to the choice of the generation mechanism of the perturbation vector $\boldsymbol{v}_t$, we cannot construct an explicit expression for the smoothed value function $V_\mu(\pi_{\boldsymbol{\theta}})$ in this setting, whose gradient is somewhat "unbiased" if the correct reward function is known. In this case, we use the original value function as the Lyapunov function and perform drift analysis as follows:

$$
V(\pi_{\boldsymbol{\theta}_t}) - V(\pi_{\boldsymbol{\theta}_{t+1}}) \leq \langle -\nabla_{\boldsymbol{\theta}}V(\pi_{\boldsymbol{\theta}_t}), \boldsymbol{\theta}_{t+1} - \boldsymbol{\theta}_t\rangle + \frac{L}{2}\|\boldsymbol{\theta}_{t+1} - \boldsymbol{\theta}_t\|_2^2
$$

$$
= -\alpha\langle\nabla_{\boldsymbol{\theta}}V(\pi_{\boldsymbol{\theta}_t}), \hat{\boldsymbol{g}}_t\rangle + \frac{L\alpha^2}{2}\|\hat{\boldsymbol{g}}_t\|_2^2
$$

$$
= -\alpha\|\nabla_{\boldsymbol{\theta}}V(\pi_{\boldsymbol{\theta}_t})\|_2^2 + \alpha\langle\nabla_{\boldsymbol{\theta}}V(\pi_{\boldsymbol{\theta}_t}), \nabla_{\boldsymbol{\theta}}V(\pi_{\boldsymbol{\theta}_t}) - \hat{\boldsymbol{g}}_t\rangle + \frac{L\alpha^2}{2}\|\hat{\boldsymbol{g}}_t\|_2^2.
$$

We also take a conditional expectation of the drift over the natural filtration $\mathcal{F}_t$ of time $t$ and obtain:

$$
\mathbb{E}\left[V(\pi_{\boldsymbol{\theta}_t}) - V(\pi_{\boldsymbol{\theta}_{t+1}})|\mathcal{F}_t\right]
$$

$$
\leq -\frac{\alpha}{2}\|\nabla_{\boldsymbol{\theta}}V(\pi_{\boldsymbol{\theta}_t})\|_2^2 + \frac{L\alpha^2}{2}\underbrace{\mathbb{E}\left[\|\hat{\boldsymbol{g}}_t\|_2^2\big|\,\mathcal{F}_t\right]}_{\mathsf{Var}_t} + \alpha\underbrace{\langle\nabla_{\boldsymbol{\theta}}V(\pi_{\boldsymbol{\theta}_t}), \nabla_{\boldsymbol{\theta}}V(\pi_{\boldsymbol{\theta}_t}) - \mathbb{E}\left[\hat{\boldsymbol{g}}_t|\,\mathcal{F}_t\right]\rangle}_{\mathsf{Bias}_t}. \quad (12)
$$

Then, we analyze the two positive drift terms $\mathsf{Bias}_t$ for the first-order gradient bias and $\mathsf{Var}_t$ for the gradient noise separately and then construct a tight upper bound of the Lyapunov drift. The results are summarized in the following two lemmas, where the proofs are deferred to Sec. G.1 and Sec. G.2:

**Lemma 6** *For ZBCPG and let $M \geq 8(H/L)^{\frac{2}{3}}$, conditioned on the information filtration $\mathcal{F}_t$ of any time $t$, the gradient bias can be upper bounded as follows:*

$$\mathsf{Bias}_t = \langle \nabla_{\boldsymbol{\theta}} V(\pi_{\boldsymbol{\theta}_t}), \nabla_{\boldsymbol{\theta}} V(\pi_{\boldsymbol{\theta}_t}) - \mathbb{E}\left[\hat{\boldsymbol{g}}_t | \mathcal{F}_t\right]\rangle \leq \left(\frac{\mu L d}{2} + \frac{2dL}{\mu}\sqrt{\frac{\log M}{M}}\right) \|\nabla_{\boldsymbol{\theta}} V(\pi_{\boldsymbol{\theta}_t})\|_2.$$

**Lemma 7** *For ZBCPG and let $M \geq 4(H/L)^2$, conditioned on the information filtration $\mathcal{F}_t$ of any time $t$, the gradient bias can be upper bounded as follows:*

$$\mathsf{Var}_t = \mathbb{E}\left[\|\hat{\boldsymbol{g}}_t\|_2^2 \big| \mathcal{F}_t\right] \leq 6d\|\nabla_{\boldsymbol{\theta}} V(\pi_{\boldsymbol{\theta}_t})\|_2^2 + \frac{3\mu^2 L^2 d^2}{2} + \frac{9d^2 L^2 \log M}{\mu^2 M} + \frac{12d^2 H^2}{\mu^2 N}.$$

Now we use Lemma. 6 and Lemma. 7 and combine the drift bound in equation 12 and obtain:

$$\mathbb{E}\left[V(\pi_{\boldsymbol{\theta}_t}) - V(\pi_{\boldsymbol{\theta}_{t+1}})|\mathcal{F}_t\right]$$
$$\leq -\frac{\alpha}{2}\|\nabla_{\boldsymbol{\theta}} V(\pi_{\boldsymbol{\theta}_t})\|_2^2 + \alpha\mathsf{Bias}_t + \frac{L\alpha^2}{2}\mathsf{Var}_t$$
$$\leq -\frac{\alpha}{2}\|\nabla_{\boldsymbol{\theta}} V(\pi_{\boldsymbol{\theta}_t})\|_2^2 + \alpha\left(\frac{\mu L d}{2} + \frac{2dL}{\mu}\sqrt{\frac{\log M}{M}}\right)\|\nabla_{\boldsymbol{\theta}} V(\pi_{\boldsymbol{\theta}_t})\|_2$$
$$+ 3\alpha^2 dL\|\nabla_{\boldsymbol{\theta}} V(\pi_{\boldsymbol{\theta}_t})\|_2^2 + \alpha^2\mu^2 L^3 d^2 + \frac{5\alpha^2 d^2 L^3 \log M}{\mu^2 M} + \frac{6\alpha^2 d^2 H^2 L}{\mu^2 N}.$$

Let $\alpha \leq (12dL)^{-1}$, we can further simplify the drift bound as:

$$\mathbb{E}\left[V(\pi_{\boldsymbol{\theta}_t}) - V(\pi_{\boldsymbol{\theta}_{t+1}})|\mathcal{F}_t\right]$$
$$\leq -\frac{\alpha}{2}\|\nabla_{\boldsymbol{\theta}} V(\pi_{\boldsymbol{\theta}_t})\|_2^2 + \alpha\left(\frac{\mu L d}{2} + \frac{2dL}{\mu}\sqrt{\frac{\log M}{M}}\right)\|\nabla_{\boldsymbol{\theta}} V(\pi_{\boldsymbol{\theta}_t})\|_2$$
$$+ \frac{1}{4}\alpha\|\nabla_{\boldsymbol{\theta}} V(\pi_{\boldsymbol{\theta}_t})\|_2^2 + \alpha^2\mu^2 L^3 d^2 + \frac{5\alpha^2 d^2 L^3 \log M}{\mu^2 M} + \frac{6\alpha^2 d^2 H^2 L}{\mu^2 N}$$
$$\leq -\frac{\alpha}{4}\|\nabla_{\boldsymbol{\theta}} V(\pi_{\boldsymbol{\theta}_t})\|_2^2 + \alpha\left(\frac{\mu L d}{2} + \frac{2dL}{\mu}\sqrt{\frac{\log M}{M}}\right)\|\nabla_{\boldsymbol{\theta}} V(\pi_{\boldsymbol{\theta}_t})\|_2$$
$$+ \alpha^2\mu^2 L^3 d^2 + \frac{5\alpha^2 d^2 L^3 \log M}{\mu^2 M} + \frac{6\alpha^2 d^2 H^2 L}{\mu^2 N},$$

where the last inequality merges both positive and negative drifts on the square of the gradient together and assumes $d \geq 4$. When $\|\nabla_{\boldsymbol{\theta}} V(\pi_{\boldsymbol{\theta}_t})\|_2$ is large, typically, when we have:

$$\|\nabla_{\boldsymbol{\theta}} V(\pi_{\boldsymbol{\theta}_t})\|_2 \geq 8\left(\frac{\mu L d}{2} + \frac{2dL}{\mu}\sqrt{\frac{\log M}{M}}\right),$$

we have the positive first-order drift, regarding the gradient norm is bounded by:

$$\left(\frac{\mu L d}{2} + \frac{2dL}{\mu}\sqrt{\frac{\log M}{M}}\right)\|\nabla_{\boldsymbol{\theta}} V(\pi_{\boldsymbol{\theta}_t})\|_2 \leq \frac{1}{8}\|\nabla_{\boldsymbol{\theta}} V(\pi_{\boldsymbol{\theta}_t})\|_2^2,$$

and thus can be merged into the negative drift as follows:

$$\mathbb{E}\left[V(\pi_{\boldsymbol{\theta}_t}) - V(\pi_{\boldsymbol{\theta}_{t+1}})|\mathcal{F}_t\right]$$
$$\leq -\frac{\alpha}{8}\|\nabla_{\boldsymbol{\theta}} V(\pi_{\boldsymbol{\theta}_t})\|_2^2 + \alpha^2\mu^2 L^3 d^2 + \frac{5\alpha^2 d^2 L^3 \log M}{\mu^2 M} + \frac{6\alpha^2 d^2 H^2 L}{\mu^2 N}.$$

If on the other hand, the gradient is small, i.e., when we have:

$$\|\nabla_{\boldsymbol{\theta}} V(\pi_{\boldsymbol{\theta}_t})\|_2 \leq 8\left(\frac{\mu L d}{2} + \frac{2dL}{\mu}\sqrt{\frac{\log M}{M}}\right),$$

and then we can upper bound the first-order drift regarding the gradient norm in another way:

$$\left(\frac{\mu L d}{2} + \frac{2dL}{\mu}\sqrt{\frac{\log M}{M}}\right)\|\nabla_{\boldsymbol{\theta}}V(\pi_{\boldsymbol{\theta}_t})\|_2 \leq 8\left(\frac{\mu L d}{2} + \frac{2dL}{\mu}\sqrt{\frac{\log M}{M}}\right)^2$$

$$\leq 4\mu^2 L^2 d^2 + \frac{64 d^2 L^2 \log M}{\mu^2 M}.$$

So in this case, the total drift can be upper bounded as:

$$\mathbb{E}\left[V(\pi_{\boldsymbol{\theta}_t}) - V(\pi_{\boldsymbol{\theta}_{t+1}})|\mathcal{F}_t\right]$$

$$\leq -\frac{\alpha}{4}\|\nabla_{\boldsymbol{\theta}}V(\pi_{\boldsymbol{\theta}_t})\|_2^2 + 4\alpha\mu^2 L^2 d^2 + \alpha^2\mu^2 L^3 d^2 + \frac{64\alpha d^2 L^2 \log M}{\mu^2 M}$$

$$+ \frac{5\alpha^2 d^2 L^3 \log M}{\mu^2 M} + \frac{6\alpha^2 d^2 H^2 L}{\mu^2 N}$$

$$\leq -\frac{\alpha}{4}\|\nabla_{\boldsymbol{\theta}}V(\pi_{\boldsymbol{\theta}_t})\|_2^2 + 5\alpha\mu^2 L^2 d^2 + \frac{64\alpha d^2 L^2 \log M}{\mu^2 M} + \frac{5\alpha^2 d^2 L^3 \log M}{\mu^2 M} + \frac{6\alpha^2 d^2 H^2 L}{\mu^2 N}.$$

The last inequality uses $\alpha \leq (12dL)^{-1}$ and $d \geq 4$. So, combining both bounds and taking an expectation, we have:

$$\mathbb{E}\left[V(\pi_{\boldsymbol{\theta}_t}) - V(\pi_{\boldsymbol{\theta}_{t+1}})|\mathcal{F}_t\right]$$

$$\leq -\frac{\alpha}{8}\mathbb{E}\left[\|\nabla_{\boldsymbol{\theta}}V(\pi_{\boldsymbol{\theta}_t})\|_2^2\right] + 5\alpha\mu^2 L^2 d^2 + \frac{5\alpha^2 d^2 L^3 \log M}{\mu^2 M} + \frac{64\alpha d^2 L^2 \log M}{\mu^2 M} + \frac{6\alpha^2 d^2 H^2 L}{\mu^2 N}$$

$$\leq -\frac{\alpha}{8}\mathbb{E}\left[\|\nabla_{\boldsymbol{\theta}}V(\pi_{\boldsymbol{\theta}_t})\|_2^2\right] + 5\alpha\mu^2 L^2 d^2 + \frac{65\alpha d^2 L^2 \log M}{\mu^2 M} + \frac{6\alpha^2 d^2 H^2 L}{\mu^2 N},$$

where we use $\alpha \leq (12dL)^{-1}$ again the in the last inequality. We then take a telescoping sum, which results in the following:

$$\mathbb{E}\left[V(\pi_{\boldsymbol{\theta}_0})\right] - \mathbb{E}\left[V(\pi_{\boldsymbol{\theta}_T})\right]$$

$$\leq -\frac{\alpha}{8}\sum_{t=0}^{T-1}\mathbb{E}\left[\|\nabla_{\boldsymbol{\theta}}V(\pi_{\boldsymbol{\theta}_t})\|_2^2\right] + \left(5\alpha\mu^2 L^2 d^2 + \frac{65\alpha d^2 L^2 \log M}{\mu^2 M} + \frac{6\alpha^2 d^2 H^2 L}{\mu^2 N}\right)T.$$

We choose $\mu$ to balance the terms inside the parentheses. Specifically, let

$$\mu^2 = \max\left\{4\sqrt{\frac{\log M}{M}}, \frac{H}{3L\sqrt{dN}}\right\}, \quad \alpha = \frac{1}{12dL}.$$

Then, we have the following inequality on the positive drift:

$$5\alpha\mu^2 L^2 d^2 \geq \frac{65\alpha d^2 L^2 \log M}{\mu^2 M}, \quad 5\alpha\mu^2 L^2 d^2 \geq \frac{6\alpha^2 d^2 H^2 L}{\mu^2 N}.$$

So we have the final results in Theorem. 2 as:

$$\frac{1}{T}\sum_{t=0}^{T-1}\mathbb{E}\left[\|\nabla_{\boldsymbol{\theta}}V(\pi_{\boldsymbol{\theta}_t})\|_2^2\right] \leq \frac{8\left(\mathbb{E}\left[V(\pi_{\boldsymbol{\theta}_0})\right] - \mathbb{E}\left[V(\pi_{\boldsymbol{\theta}_T})\right]\right)}{T\alpha} + 120\mu^2 L^2 d^2$$

$$= \mathcal{O}\left(\frac{HLd}{T} + \frac{d^2 L^2 \sqrt{\log M}}{\sqrt{M}} + \frac{HLd\sqrt{d}}{\sqrt{N}}\right).$$

### G.1 Proof of Lemma. 6

We analyze the first-order gradient bias term $\text{Bias}_t$ in the drift of equation 12. Notice that conditioned over $\mathcal{F}_t$, the randomness only comes from sampling the perturbation coordinate set $\boldsymbol{i}_t$, the

perturbation direction $\boldsymbol{\lambda}_t$, sampling trajectories, and obtaining human feedback. By equation 7 and the definition of $\hat{\boldsymbol{g}}_t$, we have:

$$
\begin{aligned}
&\nabla_{\boldsymbol{\theta}} V(\pi_{\boldsymbol{\theta}_t}) - \mathbb{E}\left[\hat{\boldsymbol{g}}_t \middle| \mathcal{F}_t\right] \\
=&\nabla_{\boldsymbol{\theta}} V(\pi_{\boldsymbol{\theta}_t}) - \mathbb{E}\left[\mathbb{E}\left[\frac{d}{\mu}\frac{\sum_{n=1}^{N}\sigma^{-1}(p_{t,n})}{N}\middle| \boldsymbol{v}_t\right]\boldsymbol{v}_t \middle| \mathcal{F}_t\right] \\
=&\nabla_{\boldsymbol{\theta}} V(\pi_{\boldsymbol{\theta}_t}) - \frac{d}{\mu}\mathbb{E}\left[\mathbb{E}\left[\sigma^{-1}(p_{t,n})\middle| \boldsymbol{v}_t\right]\boldsymbol{v}_t \middle| \mathcal{F}_t\right] \\
=&\underbrace{\nabla_{\boldsymbol{\theta}} V(\pi_{\boldsymbol{\theta}_t}) - \mathbb{E}\left[\frac{d}{\mu}\left(V(\pi_{\boldsymbol{\theta}_t+\mu\boldsymbol{v}_t}) - V(\pi_{\boldsymbol{\theta}_t})\right)\boldsymbol{v}_t \middle| \mathcal{F}_t\right]}_{\text{Bias}_{t,1}} \\
&+\frac{d}{\mu}\mathbb{E}\left[\underbrace{\left[\left(V(\pi_{\boldsymbol{\theta}_t+\mu\boldsymbol{v}_t}) - V(\pi_{\boldsymbol{\theta}_t})\right) - \mathbb{E}\left[r(\tau_{n,1}) - r(\tau_{n,0})\middle| \boldsymbol{v}_t\right]\right]}_{=0}\boldsymbol{v}_t \middle| \mathcal{F}_t\right] \\
&+\underbrace{\frac{d}{\mu}\mathbb{E}\left[\mathbb{E}\left[\left(\sigma^{-1}(p_{t,n}) - [r(\tau_{n,1}) - r(\tau_{n,0})]\right)\middle| \tau_{n,1}, \tau_{n,0}\right]\boldsymbol{v}_t \middle| \mathcal{F}_t\right]}_{\text{Bias}_{t,2}},
\end{aligned}
$$

where the second inequality uses the fact that conditioned on $\boldsymbol{v}_t$, each trajectory pair $(\tau_{n,1}, \tau_{n,0})$ is independently generated from policies $(\pi_{\boldsymbol{\theta}_t+\mu\boldsymbol{v}_t}, \pi_{\boldsymbol{\theta}_t})$. The last inequality is since conditioned on the policy $(\pi_{\boldsymbol{\theta}_t+\mu\boldsymbol{v}_t}, \pi_{\boldsymbol{\theta}_t})$, the expectation of each reward trajectory is the value function of the policy since $(\tau_{n,1}, \tau_{n,0})$ is independently generated from policies $(\pi_{\boldsymbol{\theta}_t+\mu\boldsymbol{v}_t}, \pi_{\boldsymbol{\theta}_t})$. Therefore, the $\text{Bias}_t$ term is written as follows:

$$
\text{Bias}_{\text{t}} = \langle\nabla_{\boldsymbol{\theta}} V(\pi_{\boldsymbol{\theta}_t}), \text{Bias}_{t,1}\rangle + \langle\nabla_{\boldsymbol{\theta}} V(\pi_{\boldsymbol{\theta}_t}), \text{Bias}_{t,2}\rangle.
$$

We first analyze the second term, which comes from preference estimation. Similar to the proof of ZPG, we have:

$$
\begin{aligned}
&\langle\nabla_{\boldsymbol{\theta}} V(\pi_{\boldsymbol{\theta}_t}), \text{Bias}_{t,2}\rangle \\
=&\frac{d}{\mu}\mathbb{E}\left[\mathbb{E}\left[\left(\sigma^{-1}(p_{t,n}) - [r(\tau_{n,1}) - r(\tau_{n,0})]\right)\middle| \tau_{n,1}, \tau_{n,0}\right]\langle\nabla_{\boldsymbol{\theta}} V(\pi_{\boldsymbol{\theta}_t}), \boldsymbol{v}_t\rangle \middle| \mathcal{F}_t\right] \\
\leq&\frac{d}{\mu}\|\nabla_{\boldsymbol{\theta}} V(\pi_{\boldsymbol{\theta}_t})\|_2\mathbb{E}\left[\mathbb{E}\left[\left|\sigma^{-1}(p_{t,n}) - [r(\tau_{n,1}) - r(\tau_{n,0})]\right|\middle| \tau_{n,1}, \tau_{n,0}\right]\middle| \mathcal{F}_t\right] \\
\leq&\frac{d}{\mu}\|\nabla_{\boldsymbol{\theta}} V(\pi_{\boldsymbol{\theta}_t})\|_2\left(L\sqrt{\frac{2\log M}{M}} + \frac{2H}{M^2}\right) \\
\leq&\frac{2dL}{\mu}\sqrt{\frac{\log M}{M}}\|\nabla_{\boldsymbol{\theta}} V(\pi_{\boldsymbol{\theta}_t})\|_2,
\end{aligned}
$$

where the first inequality uses Cauchy-Schwarz inequality and $\|\boldsymbol{v}_t\|_2 = 1$ and the second inequality uses Lemma. 1. The last inequality is obtained by choosing $M \geq 8(H/L)^{\frac{2}{3}}$. Then, we need to deal with the first-term inner product. We first make the following claim:

**Lemma 8** *For any iteration $t$, the following is true:*

$$
\frac{1}{d}\nabla_{\boldsymbol{\theta}} V(\pi_{\boldsymbol{\theta}_t}) = \mathbb{E}_{\boldsymbol{i}_t, \boldsymbol{\lambda}_t}\left[\langle\nabla_{\boldsymbol{\theta}} V(\pi_{\boldsymbol{\theta}_t}), \boldsymbol{v}_t\rangle\boldsymbol{v}_t\right].
$$

*where the subscript indicates the randomness only comes from $\boldsymbol{i}_t$ and $\boldsymbol{\lambda}_t$.*

**Proof.** To prove this equality, we start from the right-hand side:

$$\mathbb{E}_{\boldsymbol{i}_t,\boldsymbol{\lambda}_t}\left[\langle\nabla_{\boldsymbol{\theta}}V(\pi_{\boldsymbol{\theta}_t}),\boldsymbol{v}_t\rangle\boldsymbol{v}_t\right]=\frac{1}{K}\mathbb{E}_{\boldsymbol{i}_t,\boldsymbol{\lambda}_t}\left[\left\langle\nabla_{\boldsymbol{\theta}}V(\pi_{\boldsymbol{\theta}_t}),\sum_{j=1}^K\lambda_{i_{t,j}}\boldsymbol{e}_{i_{t,j}}\right\rangle\sum_{l=1}^K\lambda_{i_{t,l}}\boldsymbol{e}_{i_{t,l}}\right]$$

$$=\frac{1}{K}\mathbb{E}_{\boldsymbol{i}_t,\boldsymbol{\lambda}_t}\left[\sum_{j=1}^K\sum_{l=1}^K\langle\nabla_{\boldsymbol{\theta}}V(\pi_{\boldsymbol{\theta}_t}),\lambda_{i_{t,j}}\boldsymbol{e}_{i_{t,j}}\rangle\lambda_{i_{t,l}}\boldsymbol{e}_{i_{t,l}}\right]$$

$$=\frac{1}{K}\mathbb{E}_{\boldsymbol{i}_t}\left[\sum_{j=1}^K\sum_{l=1}^K\langle\nabla_{\boldsymbol{\theta}}V(\pi_{\boldsymbol{\theta}_t}),\boldsymbol{e}_{i_{t,j}}\rangle\boldsymbol{e}_{i_{t,l}}\mathbb{E}_{\boldsymbol{\lambda}_t}[\lambda_{i_{t,j}}\lambda_{i_{t,l}}]\right],$$

where the first equality is by the definition of $\boldsymbol{v}_t$, and the third equality is by the law of total expectation. For $j\neq l$, we can use the independence between different coordinates of $\boldsymbol{\lambda}_t$. Since the expectation of $\boldsymbol{\lambda}_t$ is zero, we have:

$$\mathbb{E}_{\boldsymbol{\lambda}_t}[\lambda_{i_{t,j}}\lambda_{i_{t,l}}]=\mathbb{E}_{\boldsymbol{\lambda}_t}[\lambda_{i_{t,j}}]\mathbb{E}_{\boldsymbol{\lambda}_t}[\lambda_{i_{t,l}}]=0.$$

On the other hand, if $j=l$, since $\lambda_{i_{t,j}}$ is sampled from $\{-1,1\}$, we have $\lambda_{i_{t,j}}^2=1$. Therefore, the right-hand side can be expressed as:

$$\mathbb{E}_{\boldsymbol{i}_t,\boldsymbol{\lambda}_t}\left[\langle\nabla_{\boldsymbol{\theta}}V(\pi_{\boldsymbol{\theta}_t}),\boldsymbol{v}_t\rangle\boldsymbol{v}_t\right]=\frac{1}{K}\mathbb{E}_{\boldsymbol{i}_t}\left[\sum_{j=1}^K\langle\nabla_{\boldsymbol{\theta}}V(\pi_{\boldsymbol{\theta}_t}),\boldsymbol{e}_{i_{t,j}}\rangle\boldsymbol{e}_{i_{t,j}}\right]$$

$$=\frac{1}{K}\mathbb{E}_{\boldsymbol{i}_t}\left[\sum_{i=1}^d\mathbb{1}_{i\in\boldsymbol{i}_t}\langle\nabla_{\boldsymbol{\theta}}V(\pi_{\boldsymbol{\theta}_t}),\boldsymbol{e}_i\rangle\boldsymbol{e}_i\right]$$

$$=\frac{1}{K}\sum_{i=1}^d\mathbb{P}(i\in\boldsymbol{i}_t)\langle\nabla_{\boldsymbol{\theta}}V(\pi_{\boldsymbol{\theta}_t}),\boldsymbol{e}_i\rangle\boldsymbol{e}_i$$

$$=\frac{1}{d}\sum_{i=1}^d\langle\nabla_{\boldsymbol{\theta}}V(\pi_{\boldsymbol{\theta}_t}),\boldsymbol{e}_i\rangle\boldsymbol{e}_i$$

$$=\frac{1}{d}\nabla_{\boldsymbol{\theta}}V(\pi_{\boldsymbol{\theta}_t}),$$

where the second last equality is because the probability of each coordinate being chosen by $\boldsymbol{i}_t$ is exactly $K/d$. The proof is hence concluded. ∎

With the help of the Lemma above, we can express the $\text{Bias}_{t,1}$ term as follows:

$$\langle\nabla_{\boldsymbol{\theta}}V(\pi_{\boldsymbol{\theta}_t}),\text{Bias}_{t,1}\rangle$$

$$=\left\langle\nabla_{\boldsymbol{\theta}}V(\pi_{\boldsymbol{\theta}_t}),\nabla_{\boldsymbol{\theta}}V(\pi_{\boldsymbol{\theta}_t})-\frac{d}{\mu}\mathbb{E}\left[\left(V(\pi_{\boldsymbol{\theta}_t+\mu\boldsymbol{v}_t})-V(\pi_{\boldsymbol{\theta}_t})\right)\boldsymbol{v}_t|\mathcal{F}_t\right]\right\rangle$$

$$=\frac{d}{\mu}\left\langle\nabla_{\boldsymbol{\theta}}V(\pi_{\boldsymbol{\theta}_t}),\mathbb{E}\left[\frac{\mu}{d}\nabla_{\boldsymbol{\theta}}V(\pi_{\boldsymbol{\theta}_t})-\left(V(\pi_{\boldsymbol{\theta}_t+\mu\boldsymbol{v}_t})-V(\pi_{\boldsymbol{\theta}_t})\right)\boldsymbol{v}_t|\mathcal{F}_t\right]\right\rangle$$

$$=-\frac{d}{\mu}\mathbb{E}\left[\left(V(\pi_{\boldsymbol{\theta}_t+\mu\boldsymbol{v}_t})-V(\pi_{\boldsymbol{\theta}_t})-\langle\nabla_{\boldsymbol{\theta}}V(\pi_{\boldsymbol{\theta}_t}),\mu\boldsymbol{v}_t\rangle\right)\langle\nabla_{\boldsymbol{\theta}}V(\pi_{\boldsymbol{\theta}_t}),\boldsymbol{v}_t\rangle|\mathcal{F}_t\right]$$

$$\leq\frac{d}{\mu}\mathbb{E}\left[|V(\pi_{\boldsymbol{\theta}_t+\mu\boldsymbol{v}_t})-V(\pi_{\boldsymbol{\theta}_t})-\langle\nabla_{\boldsymbol{\theta}}V(\pi_{\boldsymbol{\theta}_t}),\mu\boldsymbol{v}_t\rangle|\|\nabla_{\boldsymbol{\theta}}V(\pi_{\boldsymbol{\theta}_t})\|_2\|\boldsymbol{v}_t\|_2|\mathcal{F}_t\right].$$

By the smoothness of the value function $V$, we have:

$$|V(\pi_{\boldsymbol{\theta}_t+\mu\boldsymbol{v}_t})-V(\pi_{\boldsymbol{\theta}_t})-\langle\nabla_{\boldsymbol{\theta}}V(\pi_{\boldsymbol{\theta}_t}),\mu\boldsymbol{v}_t\rangle|\leq\frac{\mu^2L}{2}\|\boldsymbol{v}_t\|_2=\frac{\mu^2L}{2}.$$

So we can substitute into the bias term and obtain:

$$\langle\nabla_{\boldsymbol{\theta}}V(\pi_{\boldsymbol{\theta}_t}),\text{Bias}_{t,1}\rangle\leq\frac{d}{\mu}\mathbb{E}\left[\frac{\mu^2L}{2}\|\nabla_{\boldsymbol{\theta}}V(\pi_{\boldsymbol{\theta}_t})\|_2\Bigg|\mathcal{F}_t\right]$$

$$=\frac{\mu Ld}{2}\|\nabla_{\boldsymbol{\theta}}V(\pi_{\boldsymbol{\theta}_t})\|_2.$$

Then, we obtain the upper bound of the gradient bias as follows:

$$\mathsf{Bias_t} = \langle \nabla_{\boldsymbol{\theta}} V(\pi_{\boldsymbol{\theta}_t}), \mathsf{Bias}_{t,1} + \mathsf{Bias}_{t,2} \rangle \leq \left( \frac{\mu L d}{2} + \frac{2dL}{\mu} \sqrt{\frac{\log M}{M}} \right) \|\nabla_{\boldsymbol{\theta}} V(\pi_{\boldsymbol{\theta}_t})\|_2.$$

## G.2 PROOF OF LEMMA. 7

We aim for the variance term $\mathsf{Var}_t$ in the Lyapunov drift in equation 12. We first have an upper bound variance as follows:

$$\begin{aligned}
\mathsf{Var}_t =& \mathbb{E}\left[ \left\| \frac{d}{\mu} \frac{\sum_{n=1}^{N} \sigma^{-1}(p_{t,n})}{N} \boldsymbol{v}_t \right\|_2^2 \middle| \mathcal{F}_t \right] \\
\leq& 3\,\mathbb{E}\left[ \underbrace{\left\| \frac{d \sum_{n=1}^{N} \left( \sigma^{-1}(p_{t,n}) - [r(\tau_{n,1}) - r(\tau_{n,0})] \right)}{\mu N} \boldsymbol{v}_t \right\|_2^2 \middle| \mathcal{F}_t}_{\mathsf{Var}_{t,1}} \right] \\
& + 3\,\mathbb{E}\left[ \underbrace{\left\| \frac{d}{\mu} \left( \frac{\sum_{n=1}^{N} (r(\tau_{n,1}) - r(\tau_{n,0}))}{N} - (V(\pi_{\boldsymbol{\theta}_t + \mu \boldsymbol{v}_t}) - V(\pi_{\boldsymbol{\theta}_t})) \right) \boldsymbol{v}_t \right\|_2^2 \middle| \mathcal{F}_t}_{\mathsf{Var}_{t,2}} \right] \\
& + 3\,\mathbb{E}\left[ \underbrace{\left\| \frac{d}{\mu} (V(\pi_{\boldsymbol{\theta}_t + \mu \boldsymbol{v}_t}) - V(\pi_{\boldsymbol{\theta}_t})) \boldsymbol{v}_t \right\|_2^2 \middle| \mathcal{F}_t}_{\mathsf{Var}_{t,3}} \right].
\end{aligned}$$

The first term comes from the human preference estimation error, which can be bounded using equation 6 from Lemma. 1 as follows:

$$\begin{aligned}
\mathsf{Var}_{t,1} =& \mathbb{E}\left[ \left\| \frac{d \sum_{n=1}^{N} \left( \sigma^{-1}(p_{t,n}) - [r(\tau_{n,1}) - r(\tau_{n,0})] \right)}{\mu N} \boldsymbol{v}_t \right\|_2^2 \middle| \mathcal{F}_t \right] \\
=& \frac{d^2}{\mu^2 N^2} \mathbb{E}\left[ \left| \sum_{n=1}^{N} \left( \sigma^{-1}(p_{t,n}) - [r(\tau_{n,1}) - r(\tau_{n,0})] \right) \right|^2 \|\boldsymbol{v}_t\|_2^2 \middle| \mathcal{F}_t \right] \\
\leq& \frac{d^2}{\mu^2 N} \sum_{n=1}^{N} \mathbb{E}\left[ \left| \sigma^{-1}(p_{t,n}) - [r(\tau_{n,1}) - r(\tau_{n,0})] \right|^2 \right] \\
\leq& \frac{d^2}{\mu^2} \left( \frac{2L^2 \log M}{M} + \frac{4H^2}{M^2} \right) \\
\leq& \frac{3d^2 L^2 \log M}{\mu^2 M},
\end{aligned}$$

where the first inequality uses Cauchy-Schwarz inequality and the fact that $\|\boldsymbol{v}_t\|_2^2 = 1$, and the second inequality uses Lemma. 1. The final inequality is true by choosing $M \geq 4(H/L)^2$. The second term comes from using the empirical average reward to estimate the value function, similar

to REINFORCE. So we have:

$$\mathsf{Var}_{t,2} = \mathbb{E}\left[\left\|\frac{d}{\mu}\left(\frac{\sum_{n=1}^{N}\left(r(\tau_{n,1})-r(\tau_{n,0})\right)}{N} - \left(V(\pi_{\boldsymbol{\theta}_t+\mu\boldsymbol{v}_t})-V(\pi_{\boldsymbol{\theta}_t})\right)\right)\boldsymbol{v}_t\right\|_2^2\,\middle|\,\mathcal{F}_t\right]$$

$$= \frac{d^2}{\mu^2 N^2}\mathbb{E}\left[\left|\sum_{n=1}^{N}\left(\underbrace{\left(r(\tau_{n,1})-r(\tau_{n,0})\right)-\left(V(\pi_{\boldsymbol{\theta}_t+\mu\boldsymbol{v}_t})-V(\pi_{\boldsymbol{\theta}_t})\right)}_{\mathsf{E}_{t,n}}\right)\right|_2^2\|\boldsymbol{v}_t\|_2^2\,\middle|\,\mathcal{F}_t\right]$$

$$= \frac{d^2}{\mu^2 N^2}\mathbb{E}\left[\left|\sum_{n=1}^{N}\mathsf{E}_{t,n}\right|_2^2\,\middle|\,\mathcal{F}_t\right],$$

where the last equality is because $\|\boldsymbol{v}_t\|_2^2 = 1$. We open up the square as:

$$\mathbb{E}\left[\left|\sum_{n=1}^{N}\mathsf{E}_{t,n}\right|_2^2\,\middle|\,\mathcal{F}_t\right] = \sum_{n=1}^{N}\mathbb{E}\left[\left|\mathsf{E}_{t,n}\right|_2^2\,\middle|\,\mathcal{F}_t\right] + \sum_{i\neq j}\mathbb{E}\left[\mathbb{E}[\mathsf{E}_{t,i}\mathsf{E}_{t,j}|\boldsymbol{v}_t]|\,\mathcal{F}_t\right]$$

$$= \sum_{n=1}^{N}\mathbb{E}\left[\left|\mathsf{E}_{t,n}\right|_2^2\,\middle|\,\mathcal{F}_t\right] + \sum_{i\neq j}\mathbb{E}\left[\mathbb{E}[\mathsf{E}_{t,i}|\boldsymbol{v}_t]|\,\mathcal{F}_t\right]\mathbb{E}\left[\mathbb{E}[\mathsf{E}_{t,j}|\boldsymbol{v}_t]|\,\mathcal{F}_t\right]$$

$$= \sum_{n=1}^{N}\mathbb{E}\left[\left|\mathsf{E}_{t,n}\right|_2^2\,\middle|\,\mathcal{F}_t\right].$$

where the second inequality uses the independence between trajectories generated at the same step, and the last equality is due to $\mathbb{E}[(r(\tau_{n,1})-r(\tau_{n,0}))-(V(\pi_{\boldsymbol{\theta}_t+\mu\boldsymbol{v}_t})-V(\pi_{\boldsymbol{\theta}_t}))]=0$ for any fixed perturbation direction $\boldsymbol{v}_t$ from the definition of value function. So we have:

$$\mathsf{Var}_{t,2} = \frac{d^2}{\mu^2 N^2}\sum_{n=1}^{N}\mathbb{E}\left[\left|\mathsf{E}_{t,n}\right|_2^2\,\middle|\,\mathcal{F}_t\right]$$

$$= \frac{d^2}{\mu^2 N^2}\sum_{n=1}^{N}\mathbb{E}\left[\left|(r(\tau_{n,1})-r(\tau_{n,0}))-(V(\pi_{\boldsymbol{\theta}_t+\mu\boldsymbol{v}_t})-V(\pi_{\boldsymbol{\theta}_t}))\right|_2^2\,\middle|\,\mathcal{F}_t\right]$$

$$\leq \frac{4d^2 H^2}{\mu^2 N},$$

where the last inequality uses the fact that both the difference of reward and the difference of value function are within $[-H, H]$. Now we analyze the last term:

$$\mathsf{Var}_{t,3} = \mathbb{E}\left[\left\|\frac{d}{\mu}\left(V(\pi_{\boldsymbol{\theta}_t+\mu\boldsymbol{v}_t})-V(\pi_{\boldsymbol{\theta}_t})\right)\boldsymbol{v}_t\right\|_2^2\,\middle|\,\mathcal{F}_t\right]$$

$$= \frac{d^2}{\mu^2}\mathbb{E}\left[\left\|\left(V(\pi_{\boldsymbol{\theta}_t+\mu\boldsymbol{v}_t})-V(\pi_{\boldsymbol{\theta}_t})-\langle\nabla_{\boldsymbol{\theta}}V(\pi_{\boldsymbol{\theta}_t}),\mu\boldsymbol{v}_t\rangle\right)\boldsymbol{v}_t+\langle\nabla_{\boldsymbol{\theta}}V(\pi_{\boldsymbol{\theta}_t}),\mu\boldsymbol{v}_t\rangle\boldsymbol{v}_t\right\|_2^2\,\middle|\,\mathcal{F}_t\right]$$

$$\leq \frac{2d^2}{\mu^2}\mathbb{E}\left[\left\|\left(V(\pi_{\boldsymbol{\theta}_t+\mu\boldsymbol{v}_t})-V(\pi_{\boldsymbol{\theta}_t})-\langle\nabla_{\boldsymbol{\theta}}V(\pi_{\boldsymbol{\theta}_t}),\mu\boldsymbol{v}_t\rangle\right)\boldsymbol{v}_t\right\|_2^2\,\middle|\,\mathcal{F}_t\right]$$

$$+ \frac{2d^2}{\mu^2}\mathbb{E}\left[\left\|\langle\nabla_{\boldsymbol{\theta}}V(\pi_{\boldsymbol{\theta}_t}),\mu\boldsymbol{v}_t\rangle\boldsymbol{v}_t\right\|_2^2\,\middle|\,\mathcal{F}_t\right].$$

The first term can be bounded using the smoothness of the value function as follows:

$$\mathbb{E}\left[\left\|\left(V(\pi_{\boldsymbol{\theta}_t+\mu\boldsymbol{v}_t})-V(\pi_{\boldsymbol{\theta}_t})-\langle\nabla_{\boldsymbol{\theta}}V(\pi_{\boldsymbol{\theta}_t}),\mu\boldsymbol{v}_t\rangle\right)\boldsymbol{v}_t\right\|_2^2\,\middle|\,\mathcal{F}_t\right]$$

$$= \mathbb{E}\left[\left|\left(V(\pi_{\boldsymbol{\theta}_t+\mu\boldsymbol{v}_t})-V(\pi_{\boldsymbol{\theta}_t})-\langle\nabla_{\boldsymbol{\theta}}V(\pi_{\boldsymbol{\theta}_t}),\mu\boldsymbol{v}_t\rangle\right)\right|_2^2\|\boldsymbol{v}_t\|_2^2\,\middle|\,\mathcal{F}_t\right]$$

$$\leq \mathbb{E}\left[\frac{L^2\mu^4}{4}\|\boldsymbol{v}_t\|_2^4\,\middle|\,\mathcal{F}_t\right]$$

$$= \frac{L^2\mu^4}{4},$$

where the last equality uses $\|\boldsymbol{v}_t\|_2^2 = K$. The second term can be simplified as follows:

$$
\mathbb{E}\left[\left\|\langle\nabla_{\boldsymbol{\theta}}V(\pi_{\boldsymbol{\theta}_t}), \mu\boldsymbol{v}_t\rangle\boldsymbol{v}_t\right\|_2^2 \Big| \mathcal{F}_t\right] = \mu^2\mathbb{E}\left[\nabla_{\boldsymbol{\theta}}V(\pi_{\boldsymbol{\theta}_t})^\top \boldsymbol{v}_t\boldsymbol{v}_t^\top \nabla_{\boldsymbol{\theta}}V(\pi_{\boldsymbol{\theta}_t})\|\boldsymbol{v}_t\|_2^2 \Big| \mathcal{F}_t\right]
$$
$$
= \mu^2\nabla_{\boldsymbol{\theta}}V(\pi_{\boldsymbol{\theta}_t})^\top \mathbb{E}\left[\boldsymbol{v}_t\boldsymbol{v}_t^\top \Big| \mathcal{F}_t\right]\nabla_{\boldsymbol{\theta}}V(\pi_{\boldsymbol{\theta}_t}).
$$

Notice that $\boldsymbol{v}_t$ is a vector such that on the coordinates chosen by $\boldsymbol{i}_t$, the element is either $-1/\sqrt{K}$ or $1/\sqrt{K}$, and on other coordinates, it is zero. So $\boldsymbol{v}_t\boldsymbol{v}_t^\top$ will result in a diagonal matrix where on position $(i_{t,1}, i_{t,1}), \cdots, (i_{t,K}, i_{t,K})$ the element is $1/K$ and 0 otherwise. Since each coordinate $i$ will be chosen in each iteration in $\boldsymbol{i}_t$ with probability $K/d$, taking an expectation over $\boldsymbol{v}_t\boldsymbol{v}_t^\top$ will result in a diagonal matrix where the diagonal elements are $1/d$. Therefore, we have:

$$
\mathbb{E}\left[\left\|\langle\nabla_{\boldsymbol{\theta}}V(\pi_{\boldsymbol{\theta}_t}), \mu\boldsymbol{v}_t\rangle\boldsymbol{v}_t\right\|_2^2 \Big| \mathcal{F}_t\right] = \frac{\mu^2}{d}\nabla_{\boldsymbol{\theta}}V(\pi_{\boldsymbol{\theta}_t})^\top \nabla_{\boldsymbol{\theta}}V(\pi_{\boldsymbol{\theta}_t}) = \frac{\mu^2}{d}\|\nabla_{\boldsymbol{\theta}}V(\pi_{\boldsymbol{\theta}_t})\|_2^2.
$$

So Let us combine both terms and we can bound $\mathsf{Var}_{t,3}$ as follows:

$$
\mathsf{Var}_{t,3} \leq \frac{\mu^2 L^2 d^2}{2} + 2d\|\nabla_{\boldsymbol{\theta}}V(\pi_{\boldsymbol{\theta}_t})\|_2^2.
$$

So, combining three terms, the variance drift can be bounded as:

$$
\mathsf{Var}_t \leq 6d\|\nabla_{\boldsymbol{\theta}}V(\pi_{\boldsymbol{\theta}_t})\|_2^2 + \frac{3\mu^2 L^2 d^2}{2} + \frac{9d^2 L^2 \log M}{\mu^2 M} + \frac{12d^2 H^2}{\mu^2 N}.
$$