# OpenReview forum: "Zeroth-Order Policy Gradient for Reinforcement Learning from Human Feedback without Reward Inference"
_ICLR.cc/2025/Conference — ICLR 2025 Poster_

### Official Review · Reviewer_9HsE · 2024-10-27

**Soundness:** 3
**Presentation:** 3
**Contribution:** 3
**Rating:** 8
**Confidence:** 4

**Summary:**

This paper introduces two new algorithms for RLHF that do not require learning a reward function. The core algorithm is a variation of reinforce, using a zero-order gradient approximator (the two proposed algorithms differing in the choice of the perturbation direction). To avoid learning a reward function, the authors assume that the preference model is Bradley-Terry-like (considering a more general sigmoid link function, instead of just the logistic function), that $M$ human raters (with $M$ large enough) are available for ranking $N$ pairs of completions (current policy - perturbation), and then use the  inverse of the link function to estimate the reward difference from the empirical preference probability, this reward difference being used for the zero-order gradient estimate. The authors show a convergence result for each algorithm, with rate of convergence (the same), for reaching a stationary policy (not necessarily the optimal one, but one with zero gradient). They also provide some explanations about the technical difficulties of their proof (full proofs being provided in the appendix). There is no empirical study.

**Strengths:**

* The proposed algorithms are interesting, at least from a theoretical viewpoint
* The paper is overall clearly and well written
* There is a nice discussion of the technical difficulties for proving their results, which is quite helpful
* The provided theoretical results sound plausible (some questions about the proofs of Thm1 though, and Thm 2 has not been double-checked)

**Weaknesses:**

* There are overclaims and/or lack of discussion of the limitations of the proposed approach. Drawbacks of DPO or classic RLHF are discussed at length, suggesting that the proposed approach alleviate them (which is only very partly true), but not the limitations of the proposed approach. The strongest issue is that it is assumed that $M$ humans are available for rating $N$ pairs of trajectories for each gradient step (and this must be done online, because the proposed approach is purely on-policy). So, it is hard to see any setting where this could be applicable or reasonable, and it is a strong drawback compared to more practical RLHF approaches. It may be fine to have an algorithm with theoretical guarantees that is not fully applicable, but this should be discussed.
*  The proposed algorithms seem to be new, and they are explicitly positioned as addressing shortcomings of classic RLHF or DPO, which would call for some empirical results, at least on bandits or toy MDPs (possibly comparing with classic RLHF, DPO, but also things like online DPO, eg see « Direct language model alignment from online ai feedback » by Guo et al). There is no such empirical study, and it would really strengthen the paper by providing some empirical evidences on some claims done in the paper.
* There are parts of the paper and of the proofs that are not totally clear, see questions.

**Questions:**

## Lack of discussion
* Can you please discuss more thoroughly the limitations of the proposed approach (like the need to have on-policy online human raters), and in more details what are the pros and cons compared to DPO and classic RLHF?
* One advantage claimed compared to DPO is the ability to handle stochastic transitions. However, the focus is also mostly done on LLMS. Can you explain in which case we would have stochastic transitions in this context (and how it is handled by your framework)?
* There is no KL regularization, compared to all other works for doing RLHF (and not without reason). Can you expand on this, is it no longer required (from a practical perspective)?
* The paper claims going « beyond the BT model », but the generalization is rather weak, could it handle more general preference models such as those considered for example in « Nash learning from human feedback » by Munos et al?

## No empirical study
* Given that the analyzed algorithm is not representative of practical approaches, and that the second proposed algorithm is motivated by computational aspects (memory, distributed aspect), would it make sense to at least provide some minimal empirical study (with the proposed algorithms, but also DPO, classic RLHF, online DPO)? Or to provide a clear discussion about why it would not bring interesting evidences?

## Aspects to clarify and/or correct
* It is claimed (l.163) that the considered setting encompasses things like convex MDP. Can the authors expend on this (from the setting, they do not make the assumption that the return is sum-decomposable, but why would this tell that it can be framed as a convex function of the occupancy measure)?
* Would it be possible to group or recall the properties and assumptions of the link function, as it is a quite central object? For example, should we not assume also that $\sigma(x) = 1 - \sigma(-x)$, or is it a consequence of what is already written? Could you also give some examples of the Lipschitz constants $L$ (Assm 2), for example for the logistic function used in Bradley-Terry?
* There is a typo l. 305 (as written, the gradient is always zero, issue in the second value index)
* Assm 2 and 3, the same notation $L$ is used for the Lipschitz constant of $\sigma^{-1}$ and for $L$-smoothness of $V$. There is not reason for these two values to be the same. They are also confused in the proofs. Would it be possible to restate the results by making them distinct (or at least by stating that it’s for some upper-bound of both terms, if that's enough to make proofs work)?
* In Thm 1 and 2, $M$ is a function of $D$, quantity which has never been defined. Even checking the proofs it is not clear what $D$ is. Please clarify (and this is especially important as it is the number of human raters).
* Corollary 1 doesn’t seem to be proven, even in the appendix, can you provide a proof?
* l. 435, how does $\theta^*$ appear with a telescoping argument (plus, the corresponding equation in the appendix involves $\theta_T$, which seems more logical)?
* Lemma 2, would be helpful to explain why the trim operator is introduced (as the result would hold without the trim according to the proof)
* l. 977, typo (parenthesis)
* l. 1014, is it the same $L$ as in Assm 3?
* the end of proof of Thm 1 is pretty unclear (line 1104 to the end of the proof), which is a pitty, as it is the clever part of the proof discussed l.455-465. Why do we have Eq 1106, and why does it lead (and how) to the equation line 1110? It is claimed to be for the case of large gradient, but line 1119 it is again for the case of large gradient. How is the Eq line 1122 obtained, and how does it relates to the one in line 1107? Do these both equations covers all possible gradient cases, and why? L. 1134, how are combined the bounds (max of them, summing them, something else)? Overall, could you reexplain this overall important part?
* l. 1214, there is a missing term (or the line should be removed) * l. 1230-1241, why the 3 factor? (not wrong, but why)

---

> ### Author Response · Authors · 2024-11-22
> **Response to Reviewer 9HsE (1/2)**
>
> We thank the reviewers for their precious time in reviewing our paper carefully, and for their valuable comments. We apologize that the typos in our original manuscript caused confusion to the reviewer and have fixed all of them. Below, we discuss the points raised by the reviewer.
>
> #  Human Labeling Complexity and Limitations
> It is correct that our algorithm suffers from online human labeling complexity. We thank the reviewer for bringing this up, and we have added a paragraph discussing our limitations. Generally, for stochastic MDPs, DPO-based algorithms have bad performances since the loss is specifically designed under deterministic MDPs. The classic RLHF method with a reward model may be applied but are subject to the error of inaccurate reward model. In our experiment, neither baselines have a performance comparable to our approach. Our methods don't require a reward model but need multiple online human evaluations per iteration. One possible way to alleviate this limitation is to replace human feedback with AI feedback as suggested by online DPO. Another possible solution is to apply variance reduction methods such as momentum in gradient descent so we will require fewer samples and fewer human annotation tasks for each iteration.
>
> # Emprical Results
> Please refer to the experiment section of our general response. It is shown that both ZPG and ZBCPG have better performance than DPO-based algorithms since the DPO loss is specific to deterministic MDPs. They also perform better than classic RLHF with a reward model, since the reward models may be inaccurate.
>
> # Stochastic Transition
> Our paper does not focus on applying RLHF in LLMs but on more general RL problems. For example, in robotic learning, agents operating in natural environments may encounter stochastic disturbances such as wind or other natural events. Our paper shows the potential of developing RLHF algorithms and improving the performance of agents in such scenarios.
>
> # KL Regularization
> The KL regularization used in other RLHF papers served two purposes. First, in deterministic MDPs, the KL regularization ensures an analytical solution of the best policy which is essential in obtaining the DPO loss. Second, it ensures the fine-tuning policy does not deviate much from the initial point, which exhibits stability and preserves good properties of the initial point (usually a pre-trained model). However, if your pre-trained model is not good enough, the KL regularization also limits the possibility of converging to the best policy defined as the policy maximizing the value function. Our paper studies the general RL training process so we focus on finding the best policy and do not restrict ourselves to KL-constrained problems.
>
> # Preference Model Compared to Nash Learning
> The preference model studied in our paper does not cover the preference in Nash Learning, nor does the preference in Nash Learning cover our preference model. Both models are generalizations of the BT model with trajectory feedback. Our preference model admits transitivity but does not require antisymmetry, on the other hand, the preference studied in Nash Learning requires antisymmetry but does not require transitivity. Moreover, the definition and performance of the best policies are also different, so it is hard to compare from a theoretical perspective directly. A more detailed discussion of our studied preference model alongside its value can be found in our general response. Specifically, the Weibull model allows us to resolve preferences that are not antisymmetric.

---

> ### Author Response · Authors · 2024-11-22
> **Response to Reviewer 9HsE (2/2)**
>
> # Clarifications and Corrections
> We sincerely thank the reviewer for going through our paper in detail. We apologize for the confusion caused by typos and have already fixed them in our revision.
>
> **Link function**: We will revise the paper to state the properties of link function and recall them when needed. We do not require antisymmetry of the link function, and the result still holds as long as the link function is known, e.g. a example could be Weibull link function. The smooth parameter $L$ for BT model can be upper bounded by $4H$ in our setting.
>
> **Convex MDP**: We apologize for the mistake and have removed the claim regarding convex MDP from the paper. We intend to say that the return in our model can be a convex function of per-step reward.
>
> **Smooth parameter $L$**: to not introduce additional notations for constants, the constant $L$ in our paper should satisfy both Assumption 2 and 3 simultaneously, i.e., $L$ is the maximum of constant in Assumption 2 and constant in Assumption 3. In addition, this specific $L$ will satisfy Lemma 3. We have remarked in the paper that an upper bound would suffice for all theoretical results.
>
> **Theorem 1 and Theorem 2**: We apologize for the typo that $D$ should be $H$ in this context. Specifically, if $M=H^2$, we have the result as stated, if $M$ is smaller, an additional term $d^2 H/M$ will appear in both Theorem 1 and 2, so this is not a strict requirement to obtain theoretical results.
>
> **Corollary 1**: the proof is simply to require all three terms in Theorem 1 and 2 less than $\epsilon/3$. And it will lead to the human query complexity bound in Corollary 1. We omitted the proof in the paper because it seems straightforward from the Theorems, we will add it in our appendix if we are mistaken.
>
> $\theta^*$ **in Proof Outline**: Neglecting the difference between the value function $V$ and its smoothed version $V_\mu$, this is by replacing $V(\theta_T)$ with $V(\theta^*)$ since $V(\theta_T) \leq V(\theta^*)$ by definition of the best parameter. We have revised the main body to be $\theta_T$ to avoid confusion.
>
> **Trim operator**: with finite human evaluation, all $M$ feedbacks for the same trajectory pair may be the same, say the feedbacks are all 1, then the inverse link function will possibly output infinity (say under the BT model), which does not make sense since the reward difference is bounded in [-H, H], the trim operator ensures this will not happen and thus the inverse link function provides meaningful reward difference estimation.
>
> **Proof of Theorem 1**: we apologize for the typos that confuse the reviewer, and we have now fixed all typos and made it clear. Based on line 1182 of the current version, we can upper bound the constants before gradient norm $\|\nabla V(\pi_{\theta_t})\|$ (at LHS) by the gradient norm $\|\nabla V(\pi_{\theta_t})\|$ times $\alpha/8$, and this gives RHS of line 1187. Line 1193 should be for a small gradient norm where the inequality of line 1182 is reversed, i.e., the gradient norm $\|\nabla V(\pi_{\theta_t})\|$ has an upper bound. Then, we can use this upper bound to substitute the gradient norm at the LHS of line 1199 to get the RHS of line 1199. This covers all gradient cases. In line 1212, the bounds are combined by taking a maximum.
>
> **Line 1290 missing term**: This is not a missing term, it is by merging the $H/M^2$ term into the $1/\sqrt{M}$ term when $M$ is large enough, i.e., $M \geq 8 (H/L)^{\frac{2}{3}}$. In this case, the $H/M^2$ term is less than the latter term times a constant. So the constant in front is changed from $\sqrt{2}$ to 2.
>
> **Why 3 factor**: To avoid more discussion on the correlation amongst the three terms, we just bound it by $(a + b + c)^2 \leq 3a^2 + 3b^2 + 3c^2$, so there comes the 3 factor.
>
> ***
> We hope our response can address the reviewer's concerns and help the reviewer understand our contributions more clearly. We would very much appreciate it if the reviewer could consider re-evaluating our paper.

---

> > ### Author Response · Authors · 2024-11-25
> >
> > Dear Reviewer 9HsE:
> >
> > We want to follow up to see whether our response addresses your concerns. If so, we would very much appreciate it if the reviewer would consider re-evaluating their score based on our response. Please don't hesitate to let us know if you have any other questions/comments. Thanks!

---

> > > ### Comment · Reviewer_9HsE · 2024-11-26
> > > **Rebuttal clarified many aspects**
> > >
> > > Thank you for the responses, clarifications and experimental results, these clarify a number of things, notably regarding the preference model or the proofs. I update my score accordingly, assuming the following additional suggestions will also be taken into consideration:
> > > * (minor) for the KL aspect, I do agree with your point. On the other hand, if the pretrained model is not good enough (or the initial policy in a robotic context), there may be severe exploration issues, not really tackled by the proposed approach (or most of policy-gradient-based approaches, especially when the learning signal is sparse). This is not a blocker, and I think that your analysis should extend to this case (because a regularized MDP is not very different from a classic MDP, but not asking to do so), but given the emphasis put on LLMs this would be worth a short discussion in the paper.
> > > * (less minor) the experimental results are a nice addition, but I would recommend providing more empirical details, it is currently not reproducible. Notably, what is the policy parameterization? Is it a (nonlinear) neural policy (to further support the theoretical analysis) or a simpler tabular policy? How is the policy initialized? How much data is used to train the RM or DPO, and how are these data collected? There are other missing details, like the optimizer or the batch size (if batches are considered). It would also be great to provide more details on the contribution specific aspects. For example, what is the value of $M$, the number of raters? Is it set to $M=H^2 = 400$? This is a quantity that would be interesting to ablate (bounds are often conservative, could work with much less). Another interesting aspect to showcase empirically is the effect of the preference model, beyond Bradley-Terry, as it is an important point of the proposed approach. For example, what would happen if different preference models are used, how would this bias the results of DPO or PPO compared to the proposed approach (as DPO relies on the BT assumption, and training the RM with the classic BT loss for PPO)

---

> > > > ### Author Response · Authors · 2024-11-28
> > > >
> > > > We sincerely thank the reviewer for re-evaluating our paper and significantly raising the score. We are happy that our responses address the reviewer's initial concerns, and we enjoyed discussing with the reviewer to improve our manuscript. We have already considered the reviewer's additional suggestions and revised our manuscript.
> > > >
> > > > ## Experimental Detail
> > > > We have added a detailed description of our experiment setup in Appendix F.  All codes that we used for numerical experiments will be publicly available in the near future.
> > > >
> > > > As a snapshot, all policies and reward models use a tabular parameterization, where parameters are uniformly initialized in [0,1]. The number of trajectories used to train RM and DPO is in the order of $10^6$, collected from the randomly initialized policy. We ensure that all algorithms use the same number of samples and human queries. The optimizers are SGD for all algorithms to be consistent with theory, and the batch size is $1000$ for all algorithms between two consecutive policy updates. Currently, the level of $M$ is $1000$. For more detail, we encourage the reviewer to refer to Appendix F of the new revision.
> > > >
> > > > The reviewer has raised very interesting experimental setups. We are currently working on incorporating less number of human evaluators $M$, and a non-BT model (the non-symmetric Weibull model). Due to the rebuttal time limit, we are unable to provide these results and will include them in the final revision if the paper is accepted.
> > > >
> > > > ## KL Regularization
> > > > We thank the reviewer for bringing this up for discussion, we have provided a detailed discussion on the pros and cons of KL regularization in Appendix B in our updated revision. Indeed, exploring MDPs with sparse rewards is extremely hard for policy gradient based algorithms, as the exploration relies on the parameterized action probability. For unregularized MDPs, since the optimal policy is deterministic, the agent's policy is likely to converge to a sub-optimal deterministic policy that lacks exploration. On the other hand, the best policy for KL regularized MDPs is likely to still be stochastic, which ensures a certain level of exploration. We agree with the reviewer on the value of KL regularization in exploring MDPs, so we also commented on how to modify our proposed algorithms for regularized MDPs. In addition to estimating the value function difference to estimate the zeroth-order gradient, we also need to estimate the difference of KL regularizers between the perturbed policy and the current policy, concerning the reference policy. So we need to maintain the reference policy and evaluate its action probability on the trajectories generated by both policies to construct estimations of both KL regularizers, and then take the difference. Using the difference of the value function estimate, and the KL regularizer estimate, we will construct the zeroth-order gradient for the regularized MDP and proceed with gradient descent.

---

### Official Review · Reviewer_BWEU · 2024-11-03

**Soundness:** 4
**Presentation:** 3
**Contribution:** 3
**Rating:** 8
**Confidence:** 3

**Summary:**

This paper proposes the use of zero-th order algorithms to align a policy to preference data. They propose ZPG, based on random spherical perturbations, and ZBCPG, based on subblockwise perturbations, and prove sample complexity and convergence rates. These algorithms generalize from a Bradley-Terry preference model to general preference model with a specified link function, as well as generalize from deterministic to stochastic domains.

**Strengths:**

- The methods ZPG and ZBCPG work for a general class of link functions (beyond the BT-model) and stochastic dynamics as well.
- Theoretical sample complexity and convergence rates that give an idea of the dependence on the number of human preferences needed, and prove convergence exists. I have not checked the proofs in detail.

**Weaknesses:**

- ZPG and ZBCPG do not work directly from an offline dataset like DPO can; they require online human preferences for the current samples. - Furthermore, even comparing ZPG to online DPO, it seems that ZPG requires more trajectory samples (i.e. more forward passes) because you need multiple perturbations, which is multiple times more costly than online DPO.
- Zero-th order methods may be slower than gradient-based methods, especially for very large models like LLMs.
- No toy experiments. While LLM experiments would seem daunting in addition to the theoretical contribution, toy experiments in simple, stochastic domains to illustrate how DPO would fail but ZPG would work, would be a great addition to further motivate the need for a DPO alternative. I would be willing to increase my score if toy experiments are added to the paper.

**Questions:**

- You mention that classic RLHF and DPO use human preferences to construct a global model, while your method uses the preferences only to locally estimate the policy gradient. While that’s true in theory, however in the actual DPO algorithm in practice, it is also using the expected difference in preference between two generations to construct a policy gradient. Like you mention, because of function approximation, the global BT model assumed by DPO no longer holds, so the empirical DPO also ends up only using local information to estimate the policy gradient. Especially if DPO is used online instead of from a fixed offline dataset. So in practice, both DPO and ZPG are using local information to estimate a policy gradient, and the global-local divide isn’t really clearly there anymore. Could you update the discussion to reflect this?
- You mention the link function generalizes the BT-model. Could you give one or  more motivating examples of non-BT models and link functions that could be applicable (no need for experiments or deep analysis, just give some intuition on when non-BT models could be useful)? I think this would strengthen the contribution of the more general link function.

-- After Author Response --
Thank you for adding the additional numerical experiments and non-BT explanations. I have increased my score.

---

> ### Author Response · Authors · 2024-11-22
>
> We thank the reviewers for their precious time in reviewing our paper and for their valuable comments which help us improve our manuscript. Below, we discuss the points raised by the reviewer.
>
> # Online Human Preference Query
> Compared to vanilla DPO, both ZPG and ZBCPG are online algorithms and require to collection of human feedback in an online fashion. This weakness is not unique to our algorithms but to all online RLHF algorithms such as online DPO. However, the strength of online RLHF is they usually provide better performances. We can also potentially replace human feedback with AI feedback to reduce the labeling burden.
>
> # Computational Cost Compared to Online DPO:
> Compared to online DPO, at each iteration, we indeed require multiple samples to estimate the value function difference, which is more costly per iteration than online DPO. However, the value of our algorithm lies in its applicability to general RL problems with stochastic transition, while DPO is for deterministic MDPs and has a poor performance in stochastic MDPs, as shown in our experiment. It is also shown in our numerical experiment that our proposed algorithms, even though require more samples each iteration, need less number of iterations to converge, so the overall computation time is still comparable. In fact, the DPO baselines took longer to run in our experiments overall. The batched sample-query-update workflow used in our algorithms may also more suitable for implementation compared to having a human in the loop to label streaming data as suggested by online DPO, and may also require fewer circles.
>
> # Zeroth-Order Method is Slower than Gradient Method:
> For large models like LLMs, gradient-based methods are usually memory inefficient, and sometimes not implementable due to the requirement to store the interim gradient results during backpropagation. It has also been shown that zeroth-order methods are a good alternative in this setting[1]. The main aim of this paper is to design provable and efficient RLHF algorithms without a reward model for general RL tasks beyond LLMs, where there are few efficient gradient-based algorithms. Our proposed algorithms serve to fill this gap.
>
> [1] Malladi, Sadhika, et al. "Fine-tuning language models with just forward passes." Advances in Neural Information Processing Systems 36 (2023): 53038-53075.
>
> # Toy Numerical Example
> Please refer to the experiment section of our general response. It is shown that both ZPG and ZBCPG have better performance than DPO-based algorithms since the DPO loss is specific to deterministic MDPs. They also perform better than classic RLHF with a reward model, since the reward models may be inaccurate.
>
> # Local Information Used in Online DPO
> We thank the reviewer for this discussion, and we agree that online DPO also uses local information for gradient estimation since it obtains new trajectories and preferences, and then evaluates the gradient based on the current model and the new samples. The global versus local divide is more about comparing to classic RLHF with reward models. We also want to remark that the relation between the optimal policy in deterministic MDP and the reward function is global, and is the core for designing DPO loss, so there is still differences regarding what local information is used. Again, we want to emphasize that **DPO is for deterministic MDPs, not for general stochastic MDPs** so our proposed algorithms can be viewed as extending this local estimation insight to general MDPs. We have added and updated a this discussion in our revision.
>
> # Non-Bradley-Terry Models
> Please refer to the Non-Bradley-Terry Model section in our general response. We also added a paragraph in our rebuttal revision discussing a general view on how to model human choices and preferences in the context of random utility theory.
>
> ***
> We hope our response can address the reviewer's concerns. Given that we indeed provided a numerical experiment which demonstrated the better performance of our approach compared to baselines, we would very very appreciate it if the reviewer could increase their score.

---

> > ### Author Response · Authors · 2024-11-25
> >
> > Dear Reviewer BWEU:
> >
> > We want to follow up to see whether our response addresses your concerns. If so, we would very much appreciate it if the reviewer could consider re-evaluating their score based on our response. Please don't hesitate to let us know if you have any other questions/comments. Thanks!

---

> > > ### Comment · Reviewer_BWEU · 2024-11-25
> > > **Thank you for the additional results**
> > >
> > > The additional experimental results and explanations are a great addition to the paper and so I have increased my score.

---

> > > > ### Author Response · Authors · 2024-11-26
> > > >
> > > > We sincerely thank the reviewer for re-evaluating and acknowledging the value of our paper. We are very happy that the additional experiment can address the reviewer's concern. We value the discussion raised by the reviewer, which helps to improve our manuscript. We thank the reviewer again for their precious time in evaluating our paper.

---

### Official Review · Reviewer_HH3A · 2024-11-03

**Soundness:** 2
**Presentation:** 2
**Contribution:** 2
**Rating:** 3
**Confidence:** 4

**Summary:**

The paper proposes a zero-order optimization algorithm for RLHF. The idea is to carry out random perturbation in the space of parameters directly and construct gradient updates based on the random samples. The paper presents some theoretical results that characterize the bias and convergence property of the algorithm.

**Strengths:**

The paper presents theoretical guarantees for zero-order optimization algorithm for RLHF.

**Weaknesses:**

The idea of applying zero-order optimization is not novel. Similar ideas have been extensively studied in the deep RL literature. Much literature reference is missing from discussion. A theoretical paper like this is not very valuable to the RLHF or deep RL community.

**Questions:**

### === Zero-order method in deep RL ===

The idea of applying random perturbation to model parameters and construct gradient accordingly, has been studied extensively in the deep RL literature as replacement for REINFORCE or PPO like algorithms. It seems that much discussion is missing from the paper such as [1] which basically started the trend of zero-order based method for deep RL.

[1] Salisman et al, 2017, Evolution Strategies as a Scalable Alternative to Reinforcement Learning

### === Relevance to RLHF ===

It's not clear if the discussion here is specific to RLHF at all. Do RLHF allow us to assume additional structure on the rewards or the parameter space? No such discussion is furthered in the paper - in regular RL or deep RL setups, the same algorithmic discussions apply. And as a result, the paper is not really a RLHF specific algorithm, the algorithm itself has been extensively studied in the RL literature as well.

### === Empirical ===

There is no empirical backup of the results at all, not even tabular experiments. I think a theoretical paper in its current form like this would not offer too much additional value to the field of RLHF or deep RL. Certain amount of experimentation is warranted, and in the case of RLHF, it is also interesting to see if the algorithm can work at all and how it compares with REINFORCE / PPO.

---

> ### Author Response · Authors · 2024-11-22
>
> We thank the reviewers for the time they spent reviewing our paper. Below, we discuss the points raised by the reviewer.
>
> # Novelty of ZPG Compared to Zeroth Order Methods in Classic RL
> Indeed zeroth-order methods have been empirically studied in classic RL in the form of evolutionary strategies. However, the theoretical guarantees of such algorithms are largely under-explored. We want to point out a major difference between classic RL and RLHF: **classic RL problems have well-formulated loss functions that can be queried, but RLHF does not**. The loss function in classic RL could be value function or KL-constrained value function. It is straightforward to follow the classic zeroth-order optimization framework to design RL algorithms by querying the loss functions, such as evolutionary strategies. However, in RLHF without a reward model, the **value function objective studied in this paper cannot be directly sampled, approximated, and queried, since the reward is unobservable**. What we can observe is only the trajectory preference feedback. One of the main messages this paper conveys is that human preference feedback can be viewed as a natural source of the zeroth-order gradient since it is more likely to inform the better trajectory. **The novelty of this paper lies in designing human query strategies and then extracting zeroth order gradients from human feedback**. We want to emphasize that this step is non-trivial and more complex compared to constructing zeroth order gradient in classic RL, since estimating the zeroth-order gradient from human feedback will result in additional bias, as stated in Sec. 4.3. The new bias from human feedback adds more difficulty to the algorithm design since we need to choose the learning rate and perturbation distance much more carefully to balance the bias and gradient noise. To the best of our knowledge, there is no RLHF algorithm for general RL problems with policy function approximation that has a provable theoretical guarantee and does not require a reward approximation. We believe ZPG points out the internal relation between human feedback and the zeroth order gradient, which will motivate more RLHF algorithms in general RL problems beyond LLMs.
>
> # Relevance to RLHF
> We argue that the design and discussion of our proposed algorithms, i.e., ZPG and ZBCPG, are specific to RLHF, in particular for RL problems beyond LLM. Compared to classic RL, RLHF reveals much less feedback information to the agent, i.e., the trajectory reward is unobservable but only the preference feedback is provided. Therefore, without a reward model, the loss functions used in classic RL algorithms, such as the PPO loss or vanilla value function, are difficult to approximate. It is unclear how to sample the loss function to construct a gradient (either first-order or zeroth-order), which prohibits directly using the zeroth order methods such as the evolutionary strategy to RLHF. The contribution of our proposed algorithms lies in building a relation between human preference and zeroth-order gradient and then constructing zeroth-order information from human feedback. This characteristic makes both algorithms an RLHF-specific algorithm and has essential differences compared to zeroth-order algorithms in classic RL such as evolutionary strategies. We thank the reviewer for mentioning the literature on evolutionary strategies and have added a discussion in our literature review comparing the similarities and differences in the revision.
>
> # Empirical Results
> Please refer to the experiment section of our general response. It is shown that both ZPG and ZBCPG have better performance than DPO-based algorithms since the DPO loss is specific to deterministic MDPs. They also perform better than classic RLHF with a reward model, since the reward models may be inaccurate.
>
> ***
> We hope our response can address the reviewer's concerns and help the reviewer understand the difference between zeroth-order methods in classic RL and in RLHF, as well as our contributions. We would very much appreciate it if the reviewer could consider re-evaluating our paper.

---

> > ### Author Response · Authors · 2024-11-25
> >
> > Dear Reviewer HH3A:
> >
> > We want to follow up to see whether our response addresses your concerns. If so, we would very much appreciate it if the reviewer would consider re-evaluating their score based on our response. Please don't hesitate to let us know if you have any other questions/comments. Thanks!

---

> > ### Comment · Reviewer_HH3A · 2024-11-27
> > **reply**
> >
> > Thank you to the response of the authors. I will adjust my scores after the discussion phase. A few follow-up thoughts
> >
> > === *classic RL problems have well-formulated loss functions that can be queried, but RLHF does not* ===
> >
> > Indeed for classic RL problems rewards are easier to obtain. I think there are two problems here: query the reward, and optimize the policy. What I meant to say is, the algorithm proposed in this work is very much alike the zero-order optimization algorithms that have been investigated in prior RL literature, and I feel the algorithmic contribution is not technically novel (despite having more theoretical guarantees adapted to concrete settings).
> >
> > The reward query part feels to me to be a separate issue - all REINFORCE based algorithms can be plugged into the reward query setting designed in this work if I understand right? Instead of learning a reward function, we can directly query reward from the human annotators using eg policy gradient algorithms. I don't think the reward query design is a fundamentally novel aspect of the algorithmic setup.
> >
> > Concretely, in algorithm 1, the algorithmic procedure effectively constructs random-perturbation based gradient estimate, using $M$ queries from human annotators as the oracle reward. I would not think it is fundamentally different from the algorithmic setup in [1], except for replacing a regular reward query by asking $M$ humans for an average rating?
> >
> > [1] Evolutionary strategy as a scalable alternative to reinforcement learning, salisman et al, 2017
> >
> > === *The novelty of this paper lies in designing human query strategies and then extracting zeroth order gradients from human feedback* ===
> >
> > If I understand right, the "design" here corresponds to asking for $M$ human feedback during a gradient estimate procedure as described in algorithm 1? But this would be a fairly straightforward way to query human annotators directly for optimization signals, as one would do when plugging in any policy optimization algorithm directly against human annotators?
> >
> > I understand that a technical difference here is that since the reward $r$ is not observed you would estimate the win rate per gradient estimate and effectively hill climb on the win rate $p$. By the transitivity entailed by the bradley-terry assumption, this will bear equivalence to maximizing the reward.
> >
> > === *They also perform better than classic RLHF with a reward model, since the reward models may be inaccurate* ===
> >
> > In the experiment here, RLHF uses a reward model but can RLHF be adapted to directly using the pairwise query protocol as proposed in algorithm 1? Since we are hill climbing on the win rate in algorithm 1, I'd feel that RLHF policy gradient estimators can be adjusted to such a setting as well.
> >
> > I guess what I meant to say is a baseline is missing - RLHF uses a learned reward, but it should be able to adapt to directly using human queried reward too - this will narrow down the comparison between the zero-order gradient estimate vs regular PG estimate.

---

> > > ### Author Response · Authors · 2024-11-30
> > >
> > > We thank the reviewer for the follow-up discussion. We remark that our RLHF algorithms are indeed based on the zeroth-order optimization as stated in our title, which was studied as early as [1]. In this paper, we investigated and built up the direct relationship between human feedback and the zeroth-order gradient estimate in the context of RLHF. As the reviewer stated, we also established theoretical guarantees in this setting.
> > >
> > > ## Reward Query as a Separate Issue and All REINFORCE-Based Algorithms Can Be Plugged in
> > > We want to point out that our algorithms don't recover per-step reward functions and only estimate the value-function difference between two policies.  Our algorithms query human preferences to recover the value-function difference between the current policy and its perturbed version, which is not a reward query. REINFORCE-based algorithms would need the per-step reward function or the value function of the current policy, which is not available under our algorithms. So, our algorithms do not have the required information to be "plugged in" to the traditional REINFORCE-based algorithms.
> > >
> > > ## Hill Climb the Win Rate
> > > Regarding the reviewer's comment: "Since the reward is not observed you would estimate the win rate per gradient estimate and effectively hill climb on the win rate," we want to clarify that even under the transitivity of the Bradley-Terry model, directly maximizing the average win rate of trajectory pairs generated by two policies is **not** equivalent to maximizing the expected reward. This is because the logistic link function of BT is non-linear, so you cannot push the expectation inside, i.e., $$ \arg\max \text{win rate} =\arg\max E[\sigma(r(\tau_1) - r(\tau_2))] \neq  \arg\max \sigma( E[r(\tau_1) - r(\tau_2)]) = \arg\max E[r(\tau_1)] - E[r(\tau_2)]$$
> > > Therefore, the win rate, or human 'average rating' as referred to by the reviewer, does not have the same role as the reward in classic RL, and hill climb on the win rate may not align with the gradient direction. It is necessary to transform the average win rate back to the true reward to estimate the value function, which results in additional bias that significantly complicates the problem and analysis. Again, we want to emphasize we don't query the reward function and value function of a given policy, but the value-function difference between two policies based on human preferences.
> > >
> > >
> > > ## Algorithm not Fundamentally Different
> > >
> > > We respectively disagree with the opinion that the paper lacks novelty and simply **"replaces a regular reward query by asking humans for an average rating"** in zeroth-order optimization. Note that zeroth order optimization [1] replaces the exact gradient with a noisy estimate (the function difference), which is from the definition of the gradient. However, everyone would agree that zeroth-order optimization is an important direction in optimization with many novel results on why it works and how to make it work better. Similarly, stochastic gradient methods replace gradients with noisy versions but have become one of the most studied problems in ML. It is not trivial to "replace a regular reward query by asking humans for an average rating", e.g., we cannot view the win rate directly as a reward. We transform preferences to the value function difference, not rewards, and we need to control the gradient bias extremely carefully due to its bias, which results in fundamental differences compared to zeroth-order optimization, such as the principle of choosing perturbation distances.
> > >
> > >
> > > ## Human Query to Gradient Estimate Design is Straightforward
> > >
> > > We also want to clarify that "design" here should also involve using the link function to transform the average win rate back to reward and then using multiple trajectories to estimate the value function. It should also involve how to choose parameters such as the learning rate $\alpha$ and perturbation distance $\mu$. Even though this may still sound natural, there is no guarantee that a "natural" method will always work. Indeed it does not directly work, since the estimation error in the win rate will lead to a biased gradient estimate, instead of an unbiased gradient estimate like in the existing zeroth-order optimization. This bias could lead to divergence and break the algorithm if not controlled carefully. Much effort is spent on designing the number of human queries $M$, number of trajectories $N$, the learning rate $\alpha$, and perturbation distance $\mu$, to effectively control the gradient bias and variance, to minimize the human query complexity. This is highly non-trivial and is different from the classic zeroth-order results [1], which for example suggests that $\mu$ should typically be as small as possible. In our case, small $\mu$ would significantly amplify the gradient bias, and we should choose a moderate perturbation as stated in the paper's main Theorems.

---

> > > > ### Author Response · Authors · 2024-11-30
> > > >
> > > > ## Adapt RLHF to Directly Using Human Queried Reward
> > > >
> > > > We are not sure how to adapt RLHF algorithms based on per-step reward functions **without** inferring a reward model. Note that in the setting we consider, we cannot query the reward function or value function directly. Furthermore, we are not optimizing the win rate, but inferring the trajectory-based value function difference from it. It is hard for us to see a way to use the "regular PG estimate" without a per-step reward function or value function. We would love to get the reviewer's further comment on this.
> > > >
> > > > [1] Ghadimi, Saeed, and Guanghui Lan. "Stochastic first-and zeroth-order methods for nonconvex stochastic programming." SIAM journal on optimization 23.4 (2013): 2341-2368.

---

> ### Author Response · Authors · 2024-12-03
>
> Dear Reviewer HH3A,
>
> Thank you again for your time in reviewing our paper and for asking many valuable questions. Please don't hesitate to let us know if you have any other questions/comments before the discussion deadline today. We would be happy to answer. Thanks!

---

### Official Review · Reviewer_6Mwn · 2024-11-04

**Soundness:** 4
**Presentation:** 3
**Contribution:** 2
**Rating:** 8
**Confidence:** 4

**Summary:**

This is a novel result that establishes local convergence for an RLHF setup utilizing a novel method to estimate the difference in value functions. However, given the current literature which has established global convergence with a better sample complexity. I do not think this work has sufficient novelty for acceptance.

**Strengths:**

The paper utilizes a novel method for estimation of the difference in the value functions using the inverse of the preference functions. It is able to prove local convergence for a non-deterministic MDP. The paper is well laid out and easy to read.

**Weaknesses:**

The paper can only establish local convergence for the RLHF setup with a sample complexity of $\epsilon^{-5}$. In contrast Xie et al. (2024) established global convergence with sample complexity of $\epsilon^{-2}$.  Even though that case is for a finite state and action space, given the improved sample complexity and global convergence Xie et al. (2024) seems to be the more relevant result.

I am open to improving my score if the authors can give an argument as to why their work improves upon the cited work.



References:


Tengyang Xie, Dylan J. Foster, Akshay Krishnamurthy, Corby Rosset, Ahmed Awadallah, and Alexander Rakhlin. Exploratory preference
 optimization: Harnessing implicit q*-approximation for sample-efficient rlhf, 2024.

**Questions:**

Is there a way to go from local to global convergence? Perhaps is global convergence can be achieved, the work may be accepted for publication.

---

> ### Author Response · Authors · 2024-11-22
>
> We thank the reviewers for their precious time in reviewing our paper and for their valuable comments. Below, we discuss the points raised by the reviewer.
>
> # Comparison to Xie et al. (2024)
> We want to point out that the setting and problem studied in our paper have several fundamental differences from XPO in Xie et al, which make it difficult to directly compare the theoretical results. We outline these differences below.
>
> **Stochastic versus Deterministic MDPs:** One major difference is that Xie et al. only study deterministic MDPs while we study stochastic MDPs, where deterministic MDPs are a special class of MDPs where the transitions are deterministic. The XPO algorithm in Xie et al. is based on the DPO loss which is only meaningful under deterministic MDPs, and the global convergence result is stated only for deterministic MDPs. In contrast, we consider stochastic MDPs which model general RL problems and are much harder to solve. In fact, **in Remark 3.2 in Xie et al., the authors state that their results cannot be generalized to stochastic MDPs. Therefore, the global convergence of XPO does not apply to our setting**. Our conducted experiment also confirms their statement since DPO has a poor performance in stochastic MDPs. This phenomenon is likely to apply to all algorithms based on the DPO loss, including XPO.
>
> **Unconstrained versus KL-Constrained Objective:** The definition of optimal policy in Xie et al. is based on the KL-regularized objective, while our definition is based on the vanilla value function. Their results and algorithm cannot cover the case without KL-regularization, i.e., setting $\beta = 0$ is not meaningful in both their algorithm and their convergence result. This again shows that it is not very meaningful to directly compare the two results.
>
> **Policy Parameterization:** The results in Xie et al. are stated for finite function class with realizability, which is important in deriving their global convergence result, e.g., they assume the optimization problem in Line 6 of Algorithm 1 of each iteration can be exactly solved. Our paper studies general policy parameterization without these assumptions. Even though they can extend the results to infinite function classes such as the parameterization in our paper, exactly solving the optimization problem at each iteration becomes problematic and inevitably requires certain types of gradient descent, which is likely to result in sub-optimality.
>
> **Additional Assumptions:** The results of Xie et al. also depend on the trajectory density ratio and coverability, which may be large since some trajectories may have zero occupancy measures. We don't require such assumptions, but we require smoothness of the policy parameterization for theoretical guarantees.
>
> Overall, our paper studies a more general MDP with stochastic transition and no KL-regularization, with a more general policy parameterization. The algorithm developed by Xie et al. is not suitable for this setting and new algorithm designs (such as our proposed approaches) are required. The contributions of both papers should be viewed as complementary instead of a direct comparison.
>
> # Global Convergence
> Given the generality of the RLHF setting studied in this paper, i.e., stochastic MDPs with policy function approximation, it is extremely difficult to establish global convergence even for standard RL, except for specific parameterization or specific algorithms such as tabular softmax parameterization with natural policy gradient. **It is unlikely that any algorithm, including XPO in Xie et al., will achieve global optimality in this setting without further assumptions** such as convexity or PL condition over the function parameterization. Again, these assumptions are often questionable in practice. Considering the difficulty and generality of the setting, we think local convergence is the best result that can be obtained, and a meaningful result showing the effectiveness of the proposed algorithm to a wider range of RL problems beyond LLMs. This is also confirmed by our conducted experiment which shows a close-to-optimal empirical performance for our proposed algorithm.
>
> ***
> We hope our response can address the reviewer's concerns and help the reviewer understand difference in the cited work and our paper, as well as our contributions. We would very much appreciate it if the reviewer could consider re-evaluating our paper.

---

> > ### Comment · Reviewer_6Mwn · 2024-11-25
> > **Reply To Authors**
> >
> > Thank you for your detailed reply. Based on your response I have adjusted my score accordingly.

---

> > > ### Author Response · Authors · 2024-11-25
> > >
> > > We sincerely thank the reviewer for re-evaluating our paper and significantly increasing the score. We are very glad that our response addresses the reviewer's concern. The discussion raised by the reviewer is valuable and insightful. We thank the reviewer again for their precious time on our paper. If the reviewer has more questions, we are happy to answer.

---

### Author Response · Authors · 2024-11-22
**General Response to Reviewers**

We thank all the reviewers for their time reviewing our paper, and we answer common questions raised by the reviewers in the general response.

# Numerical Experiments
We conducted a numerical experiment and added a section to our rebuttal revision to discuss the empirical performance of ZPG and ZBCPG. We tested our proposed both algorithms on a GridWorld environment with **stochastic transitions**, where the stochasticity comes from imperfect control of external disturbances that prevent that agent from performing the action that it chooses. We compare to popular RLHF baselines: DPO, PPO with reward model, and online DPO. The complete results and discussions can be found in Sec. 5 of our revised paper, e.g., Fig. 2. Here, we outline the average trajectory return (value function) of the final policy as follows:

| Algorithm | Average Return of Final Policy |
| ----------- | ----------- |
| ZPG (ours) | 2.01 |
| ZBCPG (ours) | 1.96 |
| PPO+RM| 1.67|
|Online DPO| 1.68|
|DPO| -4.22|

With the same number of trajectories and the same number of human queries, it is shown that **both our algorithms perform better than all three baselines**. The reason partially lies in that PPO suffers from inaccurate reward models, and both online and offline versions of DPO suffer from the error of DPO loss, which is only valid and meaningful in deterministic MDPs, not stochastic ones studied in our paper, and the offline DPO is constrained to the neighborhood of a sub-optimal reference policy. These observations coincide with our discussions of the limitations of RLHF baselines in our paper and also coincide with our motivation to develop RLHF algorithms without reward inference for general RL problems. We believe the numerical experiments serve to validate the consistency of our theoretical results. It also shows the potential of applying our proposed algorithms to practical problems in the field of general RL beyond LLMs.

We also compare ZPG and ZBCPG under distributed implementations to demonstrate the advantage of block coordinate gradient descent, i.e., for every parameter block, we estimate their gradient in parallel after partitioning the human panel. Results are provided in Fig. 2(b). It also shows that as we divide the parameters into more blocks, the faster ZBCPG converges to a stationary policy. However, fewer human evaluators for each block will also introduce bias which harms the overall optimality of the converged policy.

# Non-Bradley-Terry Model
In this paper, we consider preference models assuming latent utility function, which naturally implies transitivity. Non-Bradley-Terry models under this category have been studied extensively in the discrete social choice theory (or random utility model), e.g., a comprehensive study can be found in [1][2]. Besides the BT model (also referred to as logit model), the probit model which uses Gaussian CDF as the link function is also one of the most used utility models. It is the first studied utility model due to the naturalism that the individual utility of different decision-makers is normally distributed [3]. The model correctness is validated with several datasets in [4]. The BT model has a much heavier tail probability than the probit model so is more suitable to analyze comparisons from crowdsourced human individuals since they are less trained and the variation of individual preference is large among populations. The probit model is more suitable when feedbacks are compared to well-trained panelists since the outcomes from different people are mostly aligned even when the reward difference is small. Other preference models such as the Weibull model is proposed for preferences without antisymmetry, i.e., $1-\sigma(-x) \neq \sigma(x)$ [5], which can also be used in our approach as well. In general, which model fits best depends on the specific application and there is much value in studying non-BT models to formulate RLHF approaches suitable in different domains.

[1] Train, Kenneth E. Discrete choice methods with simulation. Cambridge university press, 2009.

[2] Azari, Hossein, David Parks, and Lirong Xia. "Random utility theory for social choice." Advances in Neural Information Processing Systems 25 (2012).

[3] Thurstone, Louis L. "A law of comparative judgment." Scaling. Routledge, 2017. 81-92.

[4] Gulliksen, Harold, and John W. Tukey. "Reliability for the law of comparative judgment." Psychometrika 23 (1958): 95-110.

[5] Greene, William H. "Econometric analysis 4th edition." International edition, New Jersey: Prentice Hall (2000): 201-215.

***
We hope our response can address the reviewers' concerns, and help the reviewers understand further the contribution of our paper.

---

### Meta-Review · Area_Chair_nykj · 2024-12-22

**Metareview:**

This paper is primarily theoretical. The authors propose zeroth order optimization methods for RLHF, prove sample complexity and convergence rates.
The algorithm is quite interesting from a theoretical perspective.
The method is new and seems to alleviate some of the limitations of current algorithm for RLHF.

The main limitation, although not extensively highlighted by the reviewers, is the lack of a comprehensive empirical study.

**Additional Comments On Reviewer Discussion:**

None of significance.

---

### Decision · Program_Chairs · 2025-01-22

Accept (Poster)